# Geometric Inductive Biases of Deep Networks: The Role of Data and Architecture

**Sajad Movahedi**[1,2,3]**, Antonio Orvieto**[1,2,3] **& Seyed-Mohsen Moosavi-Dezfooli**[4]*

[1]ELLIS Institute Tübingen, [2]MPI for Intelligent Systems, [3]Tübingen AI Center, [4]Apple

{sajad.movahedi, antonio}@tue.ellis.eu, smoosavi@apple.com

## Abstract

In this paper, we propose the *geometric invariance hypothesis (GIH)*, which argues that the input space curvature of a neural network remains invariant under transformation in certain architecture-dependent directions during training. We investigate a simple, non-linear binary classification problem residing on a plane in a high dimensional space and observe that—unlike MLPs—ResNets fail to generalize depending on the orientation of the plane. Motivated by this example, we define a neural network's **average geometry** and **average geometry evolution** as compact *architecture-dependent* summaries of the model's input-output geometry and its evolution during training. By investigating the average geometry evolution at initialization, we discover that the geometry of a neural network evolves according to the data covariance projected onto its average geometry. This means that the geometry only changes in a subset of the input space when the average geometry is low-rank, such as in ResNets. This causes an architecture-dependent invariance property in the input space curvature, which we dub GIH. Finally, we present extensive experimental results to observe the consequences of GIH and how it relates to generalization in neural networks. The code for this paper is available at `https://github.com/dr-faustus/GIH`.

## 1 Introduction

Ever since the advent of deep learning, the importance of architecture and its impact on performance, especially through introducing inductive biases, have been well-known and established (Battaglia et al., 2018; Tay et al., 2023a;b; White et al., 2023). However, the modern machine learning toolbox still lacks a unified tool that can provide an interpretable and tangible connection between architecture, data, and inductive biases. This has fragmented the approaches to architecture design into heuristic-based modifications (He et al., 2016; Ioffe & Szegedy, 2015; Hochreiter & Schmidhuber, 1997), biology-inspired design (Bahdanau et al., 2015; Vaswani et al., 2017; LeCun et al., 2015), or efficiency-guided decisions (Dosovitskiy et al., 2021; Gu & Dao, 2023), among other things. Many consequential characteristics of a deep neural network such as decision boundaries, smoothness, and the types of features the model relies on for prediction can be determined from the input space. We propose the geometry of a model in the input space as a new tool to relate neural architectures and their inductive biases.

Consider a simple $D$-dimensional non-linear binary classification problem, where the discriminative features of the two classes are two 2-dimensional concentric circles (Figure 1) occupying a plane determined by two orthogonal directions $(\mathbf{u}_1, \mathbf{u}_2) \in \mathbb{R}^D$. The other $D-2$ dimensions are unstructured, meaning the variation in those dimensions is caused by i.i.d. and zero mean Gaussian noise. Solving such a problem with a nonlinear decision boundary was one of the motivations behind the exodus of the machine learning community from linear models (logistic regression, SVMs, etc.) to kernel methods and then neural networks (Bishop, 2007; Goodfellow et al., 2016). Conventional wisdom tells us that no matter on which directions the circles reside on, a modern neural network is complex enough to learn this decision boundary with ease (Zhang et al., 2017; 2021). For an MLP, this is indeed the case: any model that is wide enough and uses non-linearity with at least a single hidden layer can learn this decision boundary for any orthogonal $\mathbf{u}_1, \mathbf{u}_2$. However, a standard ResNet18 (He et al., 2016) is seldom capable of generalizing for randomly selected $\mathbf{u}_1, \mathbf{u}_2$ despite its elaborate architecture design and much larger size. This observation indicates the existence of an architecture dependent *geometric inductive bias* in the input space, which raises the question:

---
*Work done while at Imperial College London.

*How does the geometry of a neural network in the input space evolve during training,*
*and what is the role of architecture and data in it?*[1]

In our quest to answer this question, we discover a phenomenon, which we dub the **Geometric Invariance Hypothesis (GIH)**. GIH claims that for a family of neural networks $\mathcal{F}$, there exists an *architecture-dependent* subspace $\mathbb{S}_{\mathcal{F}} \subseteq \mathbb{R}^D$, where the geometry of neural networks in $\mathcal{F}$ can only change in directions $\mathbf{u} \in \mathbb{S}_{\mathcal{F}}$ during training. In the context of our example, let us assume two datasets $\mathcal{D}_A$ and $\mathcal{D}_B$ sharing x-axis $\mathbf{u}_1 = \mathbf{v} \in \mathbb{S}_{\mathcal{F}}$, but have different y-axes $\mathbf{u}_2 = \mathbf{v}_A \in \mathbb{S}_{\mathcal{F}}$ for $\mathcal{D}_A$ and $\mathbf{u}_2 = \mathbf{v}_B \notin \mathbb{S}_{\mathcal{F}}$ for $\mathcal{D}_B$, with $\mathcal{F}$ corresponding to the ResNet18 architecture. As evident by Figure 1, in this case the neural network is able to achieve generalization on $\mathcal{D}_A$, but not on $\mathcal{D}_B$. More specifically, the decision-boundaries (green curves) show us that in $\mathcal{D}_A$ the model is able to find the optimal, circular-shaped decision boundary regardless of the stochasticity introduced by initialization or mini-batching. However, in $\mathcal{D}_B$, the model is missing the signal on the directions corresponding to $\mathbf{u}_2$, relying on noise on other dimensions $\in \mathbb{S}_{\mathcal{F}}$ in order to reduce the train loss, and thus failing to generalize.

In order to formalize a notion of "geometry in the input space" for a neural architecture and how it "evolves" during training, we define two distinct but related maps for a neural network with a scalar output, which we dub the average geometry ($\mathbf{G}_{\mathcal{F}}^t$

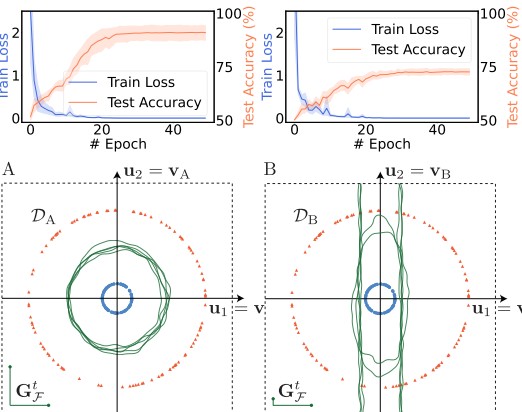

Figure 1: **Geometric Invariance Hypothesis in ResNet.** A simple binary classification problem in $\mathbb{R}^D$. The data is structured as two concentric circles in the plane defined by axes $\mathbf{u}_1, \mathbf{u}_2$ and otherwise uncorrelated. The axis $\mathbf{u}_1$ is aligned with the average geometry, i.e. $\mathbf{u}_1 \mathbf{G}_{\mathcal{F}}^t \mathbf{u}_1 \gg 0$, and shared between both datasets $\mathcal{D}_A$ and $\mathcal{D}_B$. The axis $\mathbf{u}_2$ is aligned with $\mathbf{G}_{\mathcal{F}}^t$ in $\mathcal{D}_A$ on the left and not aligned with $\mathbf{G}_{\mathcal{F}}^t$ in $\mathcal{D}_B$ on the right (see projections on bottom left corners). The green curves demonstrate the decision boundaries of the model for different initializations and mini-batches. Observe that the model can only generalize if both $\mathbf{u}_1$ and $\mathbf{u}_2$ are aligned with the average geometry.

at time $t$ during training and for the family of models $\mathcal{F}$) and the average geometry evolution ($\Delta_{\mathcal{F}}^t$ at time $t$ during training), respectively. These two maps correspond to the covariance of the gradient of a neural network w.r.t. the input and its derivative w.r.t. time according to the gradient flow model (Barrett & Dherin, 2021). We find these functions to be related to decision boundaries through a first-order Taylor approximation, and also to the Hessian of the loss w.r.t. the input. Through investigating $\Delta_{\mathcal{F}}^t$ at initialization ($t = 0$) for a simple MLP and CNN architecture, we observe that the changes in the input geometry of a neural network at initialization correspond to the projection of the train data covariance onto its geometry at initialization. Hence, we hypothesize a similar behavior for the remainder of training, which brings us to our definition of GIH, whereupon the role of $\mathbb{S}_{\mathcal{F}}$ is taken by the subspace spanned by the eigenvectors of $\mathbf{G}_{\mathcal{F}}^0$. We confirm GIH through experimentation on synthetic data and CIFAR-10.

**Related Works.** The inductive bias of neural networks are seen as one of the missing links explaining the generalization ability of deep neural networks (Soudry et al., 2018; Barrett & Dherin, 2021; Smith et al., 2021), while also being an important tool in architecture design (Battaglia et al., 2018; Tay et al., 2023a). The majority of research focused on identifying and explaining the inductive biases of deep neural networks are focused on the behavior of the model in the parameter-space (Jacot et al., 2018; Atanasov et al., 2022; Mahankali et al., 2023; Tu et al., 2024), in the frequency domain (Basri et al., 2019; Teney et al., 2024), or in the function domain (Nye & Saxe, 2018; Pérez et al., 2019; Shah et al., 2020). The closest of such attempts to our work is the neural anisotropy directions (NADs) paper (Ortiz-Jiménez et al., 2020), which provides empirical evidence for an existing geometric inductive bias in deep neural networks in linear classification. Specifically, they empirically show that the eigenvectors and the eigenvalues of the covariance of the gradient of a neural network w.r.t. the input provides a robust method for identifying and ranking directions (called NADs) in which the network prefers to classify the data in a linear classification task. As a follow-up, it was later shown that these directions can be connected to the set eigenfunctions of the NTK of a neural network in a linear task (Ortiz-Jiménez et al., 2021). In this paper, we improve upon these works by providing an explanation for this behavior through analytical and empirical investigation, while also removing the linearity assumption from the results. As a result of this investigation, we are

---

[1]Note that in this paper, we are specifically focusing on geometric inductive biases at initialization.

also able to justify the proposed method in (Ortiz-Jiménez et al., 2020) for identifying the NADs. For a detailed summary of the related works, please refer to App. A.1.

**Contributions.** Our contributions can be summarized as follows:

- We first define a function summarizing the input space geometry of a model and its evolution. We then factorize the contributors to the evolution of geometry during training into a data-dependent term and a model-dependent term. We investigate each element and their interaction theoretically, from which we conjecture a general form. Then, the results are confirmed empirically.

- Following these insights, we propose **The Geometric Invariance Hypothesis (GIH)**, which states that the geometry of deep learning models in the input space becomes invariant in certain directions determined by the architecture.

- We provide extensive experimental results aimed at establishing the validity of GIH and how it relates to generalization in neural networks. Specifically, we show that through GIH, we can determine the location of the decision boundaries a neural network can learn. Furthermore, we observe that GIH can help us detect the features and train samples the model relies on for generalization.

For details on the experimental settings and a discussion on the limitations of our work, please refer to App. A.12 and App. A.13, respectively.

## 2 THE GEOMETRY OF THE INPUT SPACE

In this section, we will first provide the notations we use in our paper. Then, we provide preliminary definitions and results which we will use to formalize and motivate our claims in the rest of the paper.

### 2.1 NOTATIONS

We denote the pre-activation output of the classification layer of the model, which for simplicity we assume to be scalar (i.e., dealing with binary classification problems) as $f_\theta(\mathbf{x})$, parameterized by $\theta$ with $\mathbf{x} \in \mathbb{R}^D$. We assume the model to be a neural network, trained on $\mathcal{D}_T = \{(\mathbf{x}_\mu, y_\mu)\}_{\mu=1}^m$, where $\mathbf{x}_\mu$ is the input and $y_\mu$ is the label. We denote a family of models (i.e., architecture) as $\mathcal{F}$, with $f_\theta(\cdot) \in \mathcal{F}$. We denote the loss function as $\mathcal{L}(\theta) = \sum_{\mu=1}^m \ell(f_\theta(\mathbf{x}_\mu), y_\mu)$, where $\mathcal{L}(.)$ is the sum of squared errors (SSE, refer to App. A.2 for details) in our theoretical analysis, and cross-entropy in our experiments. Furthermore, we denote the gradient w.r.t. the input and the parameters by $\nabla_\mathbf{x} f_\theta(\mathbf{x})$ and $\nabla_\theta f_\theta(\mathbf{x})$, respectively, and the second-order derivative w.r.t. the parameters and data as $\nabla^2_{\mathbf{x},\theta} f_\theta(\mathbf{x})$. For ease of analysis, we rely entirely on the gradient flow model for modeling the training procedure: i.e., we denote $t \in \mathbb{R}_0^+$ as the $t \geq 0$ moment in training and assume the optimization process to be continuous. So to investigate the changes in a particular function $g_\mathcal{F}(\mathbf{x}, \theta)$, which is a function of input $\mathbf{x}$ and parameters $\theta$ for the family of model $\mathcal{F}$, we usually rely on $\frac{\mathrm{d}g_\mathcal{F}}{\mathrm{d}t} = \nabla_\theta g_\mathcal{F}(\mathbf{x}, \theta)^\top \dot{\theta}$, where $\dot{\theta} = \frac{\mathrm{d}\theta}{\mathrm{d}t}$.

We also use a probing distribution over the input to get the average behavior of a particular function $g_\mathcal{F}(\mathbf{x}, \theta)$, which we denote as $\mathbf{x} \sim \mathcal{P}$. The probing distribution can be any distribution of choice that covers the area of interest in the input space. For instance, we can use train or validation samples, or a standard Gaussian distribution, which is our choice for $\mathcal{P}$ in this paper. We also define $\mathbf{A} \xrightarrow{m_1, m_2, \ldots, m_k} \mathbf{B}$ as $|\mathbf{Corr}(\mathbf{A}, \mathbf{B})|$ approaching 1 at the limit $\forall i\ m_i \to \infty$, where $\mathbf{A}$ and $\mathbf{B}$ are both matrices of the same dimension, $\mathbf{Corr}(\cdot, \cdot)$ is the Frobenius cosine similarity, $|\cdot|$ is the absolute value operator, and $\mathbf{A}$ is a function of scalars $\{m_i\}_{i=1}^k$. More informally, we use $\mathbf{A} \propto \mathbf{B}$ when we claim $|\mathbf{Corr}(\mathbf{A}, \mathbf{B})|$ for the two matrices $\mathbf{A}$ and $\mathbf{B}$ to be much larger than the magnitude of the cosine similarity between two randomly generated vectors of the same dimension to the point of one being able to approximate the other.

### 2.2 DEFINITIONS

In order to facilitate answering our central question, we will try to define two functions that help us understand the geometry of a neural network architecture in the input space, and how it changes during training. Given that our focus in this paper is on classification tasks, a good starting point would be the gradient of the model w.r.t. the input $\nabla_\mathbf{x} f_\theta(\mathbf{x})$. From robustness literature, we know that $\nabla_\mathbf{x} f_\theta(\mathbf{x})$ can inform us about the decision-boundaries in the immediate vicinity of $\mathbf{x}$ (Madry et al., 2018; Moosavi-Dezfooli et al., 2016). This can be easily shown through a first-order Taylor expansion of the output of the model. Specifically, let $\delta$ be a vector with a small L2 norm. Then we can write

$f_\theta(\mathbf{x} + \delta) - f_\theta(\mathbf{x}) \approx \delta^\top \nabla_\mathbf{x} f_\theta(\mathbf{x})$ using Taylor expansion. Therefore, $\delta$ in the direction of $\nabla_\mathbf{x} f_\theta(\mathbf{x})$ maximizes the change in the output of the model, potentially flipping its prediction. Consequently, by maximizing $\delta^\top \nabla_\mathbf{x} f_\theta(\mathbf{x})$ we obtain valuable information about the local changes in $f_\theta(\mathbf{x})$ through $\delta$.

However, a $\delta$ computed based on $\nabla_\mathbf{x} f_\theta(\mathbf{x})$ for a single value of $\theta$ only informs us about a decision-boundary for a single parameterization of the model, subjecting it to noise caused by stochastic factors such as mini-batching and initialization. Therefore, we propose to average the gradient over a distribution of $\theta$: $\mathbb{E}_\theta[\nabla_\mathbf{x} f_\theta(\mathbf{x})]$. While trying to maximize $\delta^\top \mathbb{E}_\theta[\nabla_\mathbf{x} f_\theta(\mathbf{x})]$ can be a way to find a direction that can potentially flip the prediction of $f_\theta(\mathbf{x})$, it only contains first-order information about how the distribution of $\theta$ affects the local changes in $f_\theta(\cdot)$ at $\mathbf{x}$, causing potential loss of information. For instance, let us assume a case where $\nabla_\mathbf{x} f_\theta(\mathbf{x})$ is zero-mean w.r.t. $\theta$, i.e., $\mathbf{x}$ corresponds to a local extremum. In this case, $\mathbb{E}_\theta[\nabla_\mathbf{x} f_\theta(\mathbf{x})]$ does not contain any information about the local changes in $f_\theta(\mathbf{x})$. On the other hand, one can instead look at the covariance of $\nabla_\mathbf{x} f_\theta(\mathbf{x})$ w.r.t. $\theta$ to obtain information about the directions towards which the decision boundaries have a significant presence. Therefore, a much more informative function about the local changes in $f_\theta(\mathbf{x})$ in the input space would be $\mathbb{E}_\theta\left[\nabla_\mathbf{x} f_\theta(\mathbf{x})\nabla_\mathbf{x} f_\theta(\mathbf{x})^\top\right]$. Finally, since we rarely care about the behavior of a model for a single input, we also use a probing distribution $\mathcal{P}$ to sample $\mathbf{x}$ from. With this motivation, we introduce the following definition for a function informing us about the geometry of a neural network architecture in the input space:

**Definition 2.1.** Let $\mathcal{F}$ be a family of neural networks. We define the **average geometry** of $\mathcal{F}$ at $\mathbf{x}$ and train time $t$ as:
$$\mathbf{G}_\mathcal{F}^t(\mathbf{x}) = \mathbb{E}_{\theta \sim \mathcal{T}_t}\left[\nabla_\mathbf{x} f_\theta(\mathbf{x})\nabla_\mathbf{x} f_\theta(\mathbf{x})^\top\right], \tag{1}$$
where $\mathcal{T}_t$ determines the distribution of the trajectory of $\theta$ during training on the train data $\mathcal{D}_T$ at moment $t$, with the source of stochasticity usually being initialization and mini-batching. Furthermore, we define the average geometry of $\mathcal{F}$ induced by the probing distribution $\mathcal{P}$ as $\mathbf{G}_{\mathcal{F},\mathcal{P}}^t = \mathbb{E}_{x \sim \mathcal{P}}[\mathbf{G}_\mathcal{F}^t(\mathbf{x})]$.

In App. A.3, we report that $\mathbf{G}_\mathcal{F}^t(\mathbf{x})$ is indeed directly related to the curvature of the loss w.r.t. input, making the connection between the function we propose and the geometry of a neural network architecture in the input space clear.

We can capture the geometry of a family of models in the input through $\mathbf{G}_\mathcal{F}^t(\mathbf{x})$ at time $t$ during training. However, this function does not tell us how the model geometry changes at time $t$, thus only providing half of the picture. So we need another quantity that can inform us about the evolution of the geometry in the input space. Given our line of reasoning that resulted in the definition of average geometry, we start with the function $\nabla_\mathbf{x} f_\theta(\mathbf{x})\nabla_\mathbf{x} f_\theta(\mathbf{x})^\top$. Following the gradient-flow model, we can capture the evolution of this function by getting its derivative w.r.t. time, which gives us: $\nabla_{\mathbf{x},\theta}^2 f_\theta(\mathbf{x})\dot{\theta}\nabla_\mathbf{x} f_\theta(\mathbf{x})^\top + \nabla_\mathbf{x} f_\theta(\mathbf{x})\dot{\theta}^\top \nabla_{\mathbf{x},\theta}^2 f_\theta(\mathbf{x})^\top$. With a similar motivation, we get the expectation of this quantity w.r.t. the parameters $\theta$ and $\mathbf{x}$ sampled from a probing distribution, resulting in the following definition:

**Definition 2.2.** Let $\mathcal{F}$ be a family of neural networks. We define the **average geometry evolution** of $\mathcal{F}$ at $\mathbf{x}$ and train time $t$ as:
$$\Delta_\mathcal{F}^t(\mathbf{x}) = \mathbb{E}_{\theta \sim \mathcal{T}_t}\left[\nabla_{\mathbf{x},\theta}^2 f_\theta(\mathbf{x})\dot{\theta}\nabla_\mathbf{x} f_\theta(\mathbf{x})^\top\right] + \mathbb{E}_{\theta \sim \mathcal{T}_t}\left[\nabla_{\mathbf{x},\theta}^2 f_\theta(\mathbf{x})\dot{\theta}\nabla_\mathbf{x} f_\theta(\mathbf{x})^\top\right]^\top, \tag{2}$$
where $\mathcal{T}_t$ determines the distribution of the trajectory of $\theta$ during training on the train data $\mathcal{D}_T$ at moment $t$, with the source of stochasticity usually being initialization and mini-batching. Furthermore, we define the average geometry evolution of $\mathcal{F}$ induced by the probing distribution $\mathcal{P}$ as $\Delta_{\mathcal{F},\mathcal{P}}^t = \mathbb{E}_{x \sim \mathcal{P}}[\Delta_\mathcal{F}^t(\mathbf{x})]$.

In App. A.3, we will see that $\Delta_\mathcal{F}^t(\mathbf{x})$ is indeed directly related to the evolution of the curvature of the loss w.r.t. input, making the connection between the function and the evolution of geometry of a neural network architecture in the input space clear. For the SSE loss we have $\Delta_\mathcal{F}^t(\mathbf{x}) = \mathbf{D}_\mathcal{F}^t(\mathbf{x}) + \mathbf{D}_\mathcal{F}^t(\mathbf{x})^\top$, where:
$$\mathbf{D}_\mathcal{F}^t(\mathbf{x}) = -\sum_{\mu=1}^{m} \mathbb{E}_{\theta_t}\left[\left(\nabla_{\mathbf{x},\theta}^2 f_{\theta_t}(\mathbf{x})\nabla_\theta f_{\theta_t}(\mathbf{x}_\mu)\nabla_\mathbf{x} f_{\theta_t}(\mathbf{x})^\top\right) \cdot (f_{\theta_t}(\mathbf{x}_\mu) - y_\mu))\right]. \tag{3}$$

Following (3), we can argue that the changes in the local geometry are the result of the interaction between two main components: 1) the data, and 2) the current average geometry. The first component affects the geometry through the changes in the parameter ($\dot{\theta}$) by gradient-descent, while the second component affects the geometry through the mixed-derivative $\nabla_{\mathbf{x},\theta}^2 f_\theta(\mathbf{x})$ and the gradient $\nabla_\mathbf{x} f_\theta(\mathbf{x})$. While such an interaction may seem extremely complex at first, we will see that by probing the average geometry and average geometry evolution with an appropriate probing distribution $\mathcal{P}$, we observe (through theoretical and empirical means) that the expected value of this interaction is actually surprisingly simple and akin

to a linear projection, leading to interesting results on the behavior of the average geometry $\mathbf{G}^t_{\mathcal{F},\mathcal{P}}$ and generalization in a neural network.

For simplicity, we refer to both of the average geometry and the average geometry evolution of a standard Gaussian probing distribution and at initialization (i.e., $t=0$) as $\mathbf{G}_{\mathcal{F}}$ and $\Delta_{\mathcal{F}}$, respectively. For $t>0$, we denote these two functions as $\mathbf{G}^t_{\mathcal{F}}$ and $\Delta^t_{\mathcal{F}}$. We name $\mathbf{S}=\sum_{\mu=1}^{m} x_\mu x_\mu^\top$ the unnormalized data covariance.

## 3 CHANGES IN THE AVERAGE GEOMETRY DURING TRAINING

In this section, we study the factors that affect the changes in the geometry of the model in the input space through theoretical and empirical means. More specifically, we will start by theoretically detecting the data-dependent factor in $\Delta_{\mathcal{F}}$ by investigating it in an "isotropic" model. The term "isotropic," as will become clear in this section, indicates a lack of *geometric inductive bias* in the model. We then theoretically look at the interaction between the data-dependent and model-dependent factors in a "non-isotropic" model, which we set to be a convolutional neural network with a pooling layer. Motivated by these theoretical results, we will conjecture that at the initial stage of training, the changes in the average geometry of the model follow a linear dynamic driven by an interaction between the initial geometry ($\mathbf{G}_{\mathcal{F}}$) due to the model and the covariance of the data. In App. A.4, we provide theoretical and empirical results supporting the claim that this dynamic is independent of the labels, and can be considered task-independent.

### 3.1 ISOTROPIC MODEL

We provide theoretical results for the average geometry evolution $\Delta_{\mathcal{F},\mathcal{P}}$ at initialization according to the gradient flow model, over the probing distribution $\mathcal{P}=\mathcal{N}\left(0,\sigma_x^2\mathbf{I}\right)^2$. We do not make assumptions about the data, and instead focus on models that can be considered "isotropic" (Battaglia et al., 2018).

First, let us start with a simple example of a linear regression (i.e., $\mathcal{F}$ corresponding to linear regression models), which we write as $f_\theta(\mathbf{x})=\theta^\top\mathbf{x}$. We assume that the parameters are initialized as zero-mean and i.i.d., as is the common practice. Since $\nabla_\mathbf{x}f_\theta(x)=\theta$, the average geometry at initialization for this model will be $\mathbf{G}_{\mathcal{F}}=\mathbb{E}_{\theta_0}\left[\theta_0\theta_0^\top\right]\propto\mathbf{I}$. This property indicates a "lack of geometric inductive biases" at initialization, which as shown in App. A.5, is also shared with MLPs of any depth with ReLU non-linearities. In Section 3.2, we will elaborate on the significance of this property. Following (3), and given that $\nabla^2_{\mathbf{x},\theta}f_\theta(\mathbf{x})=\mathbf{I}$, we have $\Delta_{\mathcal{F}}=-\sum_{\mu=1}^{m}\mathbb{E}_{\theta_0}\left[\left(\theta^\top\mathbf{x}_\mu-y_\mu\right)\cdot\mathbf{x}_\mu\theta^\top\right]=-\sigma_\theta^2\mathbf{S}$ for an i.i.d. and zero-mean Gaussian initialization of $\theta$ with standard deviation $\sigma_\theta$, following Stein's lemma. Therefore, we can see that in the absence of non-linearity and more layers, the average geometry evolution of an isotropic model at initialization corresponds to the covariance of the data. When introducing these two elements, in the form of an MLP with a single hidden layer and ReLU non-linearity, we have the following theorem that shows a similar behavior:

**Theorem 3.1.** *Let $\mathcal{F}$ be the family of MLPs with a single hidden layer of size $n$ and ReLU non-linearity. Assuming that we use the SSE loss, then as the input dimension $D$ and the model width $n$ become larger, the average geometry at initialization $\Delta_{\mathcal{F}}$ approaches the data covariance $\mathbf{S}$ up to a constant, i.e., $\Delta_{\mathcal{F}}\xrightarrow{n,D}\mathbf{S}$. The convergence rate is $\mathcal{O}\left(\frac{1}{\sqrt{D}\cdot n}\right)$.*

We provide a sketch of the proof and encourage the reader to read the whole proof in App. A.11.2. The average geometry evolution of this model at initialization corresponds to $n^2$ terms of the second and fourth moments of the parameters of the first layer $\phi$ over the space shared by halfspaces defined by the train data and the inputs sampled from the probing distribution, i.e. $\mathbb{E}_\phi\left[\mathbf{1}_{\phi^\top\mathbf{x}>0}\mathbf{1}_{\phi^\top\mathbf{x}_\mu>0}\phi\phi^\top\right]\mathbf{x}_\mu\mathbf{x}_\mu^\top$ and $\mathbb{E}_\phi\left[\mathbf{1}_{\phi^\top\mathbf{x}>0}\mathbf{1}_{\phi^\top\mathbf{x}_\mu>0}\left(\phi^\top\mathbf{x}_\mu\right)^2\phi\phi^\top\right]$. As the input dimension becomes large enough, these two halfspaces will almost surely become orthogonal, resulting in the second and the fourth moments of the first-layer parameters over the intersection of these two halfspaces becoming equal to the second and the fourth moments up to a constant. The second moments yield $n^2-n$ terms corresponding to $\mathbf{S}$, while the fourth moments yield $n^2-n$ terms corresponding to $\mathbf{S}$ and $n$ terms corresponding to $\mathbb{E}_{\mathbf{x}_\mu\in\mathcal{D}_T}\left[\|x_\mu\|_2^2\right]\mathbf{I}$, which vanish in favor of the $\mathbf{S}$ terms as the depth of the first layer becomes large enough.

Theorem 3.1 gives us an interesting insight into our factorization of $\Delta_{\mathcal{F}}$ underscoring that in the absence of geometric inductive biases introduced by the model, the geometry adapts according to the data covariance.

---

[2]Given the homogeneous quality of the architectures used in our theoretical analysis, we will set $\sigma_x=1$ for the probing distribution.

Therefore, the data-dependent component in (3) is equivalent to the empirical covariance of the data estimated on the train samples.

## 3.2 THE LINEAR RELATIONSHIP BETWEEN GEOMETRY, DATA, AND ARCHITECTURE

We now shift our focus to examining the effect of the model on the geometry and the interaction between the data-dependent and the model-dependent components in $\Delta_{\mathcal{F}}$. Specifically, we aim to introduce architecture components, such as convolution operations and pooling layers, that as noted in App. A.5, introduce *geometric inductive biases* to the model. We start our investigation by looking at a neural network with a linear activation function:

**Theorem 3.2.** *Let $\mathcal{F}$ be the family of convolutional neural networks with a single hidden layer, linear activation function, and a global average pooling layer on each channel. Assuming that we use the SSE loss, then as the model width $n$ becomes larger, the average geometry evolution at initialization $\Delta_{\mathcal{F}}(\mathbf{x})$ approaches $\mathbf{G}_{\mathcal{F}}(\mathbf{x})\mathbf{S}\mathbf{G}_{\mathcal{F}}(\mathbf{x})$ up to a constant, i.e., $\Delta_{\mathcal{F}}(\mathbf{x}) \xrightarrow{n} \mathbf{G}_{\mathcal{F}}(\mathbf{x})\mathbf{S}\mathbf{G}_{\mathcal{F}}(\mathbf{x})$. The convergence rate is $\mathcal{O}(1/n)$. For proof, please refer to App. A.11.3.*

From Theorem 3.2, we observe that in the initial stages of training, $\mathbf{G}_{\mathcal{F}}^t$ will change towards the linear projection of $\mathbf{S}$ onto $\mathbf{G}_{\mathcal{F}}$ in a linear convolutional neural network with pooling. Considering the results of Theorem 3.1 and Theorem 3.2, we can now better understand the effect of the two components present in (3) causing the shift in geometry at the beginning of training: the data in the form of covariance, and the architecture in the form of average geometry. We provide empirical confirmation for the two theorems in Figure 7.

Given the results of Theorem 3.1 and Theorem 3.2, we will conjecture about $\Delta_{\mathcal{F}}$ for a neural network without making assumptions about the architecture. From Theorem 3.1, we know that for the gradient-flow model at $t = 0$, the changes in the geometry in the presence of an isotropic model (more explanation in App. A.5) correspond to the covariance of the data. From Theorem 3.2, we understand that the data covariance is linearly projected on the average geometry at initialization $\mathbf{G}_{\mathcal{F}}$ for the non-isotropic model used in the theorem. Then, without any assumptions about the architecture we have the following:

*Conjecture* 1. Let $\mathcal{F}$ be a family of neural networks. Then, assuming we use the SSE loss, the average geometry evolution at initialization $\Delta_{\mathcal{F}}$ is highly correlated with the projection of data covariance onto the average geometry at initialization, i.e., $\Delta_{\mathcal{F}} \propto \mathbf{G}_{\mathcal{F}}\mathbf{S}\mathbf{G}_{\mathcal{F}}$. As a result, the average geometry will change according to this form at the initial stages of training, i.e., $\mathbf{G}_{\mathcal{F}}^t$ approaches $\mathbf{G}_{\mathcal{F}}\mathbf{S}\mathbf{G}_{\mathcal{F}}$ for $t$ close to 0.

So the changes in the geometry are caused by the projection of the covariance of the data onto the initial average geometry. As a result, in the case of the existence of a structure in the initial average geometry $\mathbf{G}_{\mathcal{F}}$, one can fully expect the geometry to be invariant (or numerically, changing extremely slowly) in certain directions with low correlation to the initial average geometry.

Considering that the average geometry evolution and our analysis of $\Delta_{\mathcal{F}}^t$ is based on the gradient-flow model, we cannot confirm Conjecture 1 directly in a discrete training regime. Therefore, in order to present empirical evidence confirming our results so far, we instead rely on looking at the average geometry $\mathbf{G}_{\mathcal{F}}^t$ for the entirety of the training period using gradient descent. In this case, an **increasing correlation** between $\mathbf{G}_{\mathcal{F}}^t$ and $\mathbf{G}_{\mathcal{F}}\mathbf{S}\mathbf{G}_{\mathcal{F}}$ (or $\mathbf{S}$ for an MLP) near the initialization point in training will support our theoretical analysis. We provide the experimental results supporting Conjecture 1 in Figure 2. In this experiment, we plot the correlation between average geometry at time $t$ ($\mathbf{G}_{\mathcal{F}}^t$) and the projection of data onto the initial average geometry of the model, as proposed in Conjecture 1. We will use the normalized Frobenius dot product to measure the correlation between two matrices. For our experiments, we use the CIFAR-10 data. We construct a binary variant of CIFAR-10 in which the task is to distinguish animal from non-animal inputs, which we dub CIFAR-2. For each class of CIFAR-2, we sample 5000 data points randomly[3]. Furthermore, we report the results for an MLP with 2 hidden layers of size 100, LeNet, ResNet18 without batch normalization, and ViT without layer normalization. More details on the implementation can be found in the appendix.

As we can observe in Figure 2, there is clear evidence for the changes in the average geometry being highly correlated with $\mathbf{S}$ and $\mathbf{G}_{\mathcal{F}}\mathbf{S}\mathbf{G}_{\mathcal{F}}$ at the beginning of training, which is what we claim in Conjecture 1.

---

[3]Note that per the results of Corollary A.4, in case of normalization, $\mathbf{G}_{\mathcal{F}}$ will depend on the variance of each individual patch of the input. In our experiments, we found that in practice it is difficult to exactly compute this "normalized" variant of $\mathbf{G}_{\mathcal{F}}$ due to its dependence on - among other things - the kernel size of the convolution operations or the patch size of the ViT. For this reason, and for the sake of simplicity, we omit the normalization layers from the architectures we experiment on for non-synthetic data.

Figure 2: The correlation between $\mathbf{G}_{\mathcal{F}}^t = \mathbf{G}_{\mathcal{F}, \mathcal{N}(0, \mathbf{I})}^t$ and $\mathbf{S}$ and $\mathbf{G}_{\mathcal{F}} \mathbf{S} \mathbf{G}_{\mathcal{F}}$ for the **(a)** MLP, **(b)** LeNet, **(c)** ResNet18 without batch normalization, and **(d)** ViT without layer normalization on CIFAR-2. Note that $D = 32 \times 32 \times 3 = 3072$, which means the expected cosine similarity of two randomly generated vectors in the input space is $\mathcal{O}(1/\sqrt{3072}) \approx 0.02$. Therefore, we consider the correlations significant.

Another interesting observation to note is that the values of $\mathbf{S}$ and $\mathbf{G}_{\mathcal{F}} \mathbf{S} \mathbf{G}_{\mathcal{F}}$ are virtually identical in the case of MLPs, which confirms our view of MLPs being isotropic, or in other words, having no geometric inductive biases, which manifests itself in the form of $\mathbf{G}_{\mathcal{F}} \approx \mathbf{I}$. On the other hand, this is not the case in the other models, with the correlation between $\mathbf{G}_{\mathcal{F}}^t$ and $\mathbf{G}_{\mathcal{F}} \mathbf{S} \mathbf{G}_{\mathcal{F}}$ being significantly larger than MLP during the initial training stage compared to $\mathbf{S}$. This difference in correlation indicates the existence of a structure in $\mathbf{G}_{\mathcal{F}}$ for these models, which we consider to be the *geometric inductive biases* of the model.

# 4 THE GEOMETRIC INVARIANCE HYPOTHESIS

In this section, we will introduce the **Geometric Invariance Hypothesis (GIH)**, providing empirical evidence supporting it, and investigating its implications on the generalization ability of deep networks. Notably, we study the relationship between the complexity of the decision boundary and GIH, establishing a connection between GIH and the simplicity bias hypothesis (SBH) (Pérez et al., 2019).

## 4.1 THE PROJECTION OF DISCRIMINANT FEATURES ONTO THE INITIAL GEOMETRY

In the previous section, we observed an intriguing phenomenon wherein at the initial stages of training, the average geometry of the model changes according to the projection of the covariance of the data $\mathbf{S}$ onto the initial average geometry $\mathbf{G}_{\mathcal{F}}$. Note that as we get further into the training process, we expect the set of features used by the model to narrow down to "discriminant features," which can be viewed as the features upon which the model can discriminate data. Assuming that the changes in the geometry are still caused by the projection of a subspace of the data support (relating to the discriminant features) onto the initial average geometry, and the initial average geometry itself is structured, then the input geometry will remain invariant in the directions with low correlation with $\mathbf{G}_{\mathcal{F}}$[4]. In App. A.5, we consider the choices in the architecture that can result in a structure in $\mathbf{G}_{\mathcal{F}}$. Furthermore, in App. A.6 we will observe the structure in $\mathbf{G}_{\mathcal{F}}$ for the neural networks used in this paper.

This observation indicates the possibility that the initial geometry also plays a role in the later stages of training. More specifically, we want to know whether the model's geometry in the input space will remain invariant during later stages of training in directions with low correlation with $\mathbf{G}_{\mathcal{F}}$ as well. We formalize this concept in the form of the following hypothesis, which we dub the "**Geometric Invariance Hypothesis**" (GIH):

*Conjecture* 2. (**Geometric Invariance Hypothesis**) Let $\mathcal{F}$ be a family of neural networks. Let **Eig**$(\mathbf{A})$ correspond to the subspace defined by the top eigenvalues of the matrix $\mathbf{A}$. We conjecture that **Eig**$(\Delta_{\mathcal{F}}^t) \subseteq$ **Eig**$(\mathbf{G}_{\mathcal{F}} \mathbf{S} \mathbf{G}_{\mathcal{F}})$. In other words, the geometry of the model remains invariant in directions not in **Eig**$(\mathbf{G}_{\mathcal{F}})$ during training.

In order to support Conjecture 2, we first define a measure to monitor the changes in the average geometry, which we call the geometric velocity: $\dot{\mathbf{G}}_t = 1 - \mathbf{Corr}(\mathbf{G}_{\mathcal{F}}^t, \mathbf{G}_{\mathcal{F}}^{t-1})$, where $\mathbf{Corr}(\cdot, \cdot)$ is the normalized Frobenius dot product. For this experiment, we will look at $\dot{\mathbf{G}}_t$ and training accuracy when the data is sampled from the Gaussian distribution $\mathcal{N}(0, \mathbf{G}_{\mathcal{F}} / \|\mathbf{G}_{\mathcal{F}}\|_2)$, where $\|\mathbf{G}_{\mathcal{F}}\|_2$ is the spectral norm of

---

[4]Note that while we use the term invariance to refer to the geometry of the model having small changes in certain directions, in practice this "invariance" comes in the form of a large condition number in $\mathbf{G}_{\mathcal{F}}$. As a result, the invariance happens on a spectrum, and the changes in $\mathbf{G}_{\mathcal{F}}^t$ in a specific direction $\mathbf{u}$ gradually decreases as the correlation between $\mathbf{u}$ and $\mathbf{G}_{\mathcal{F}}$ becomes smaller.

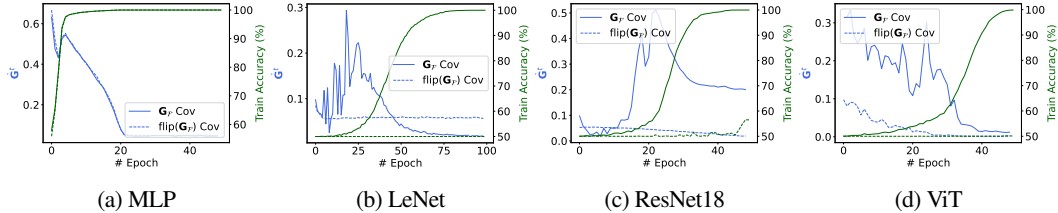

(a) MLP     (b) LeNet     (c) ResNet18     (d) ViT

Figure 3: The train accuracy (green lines) and velocity $\dot{\mathbf{G}}^t(\cdot,\cdot)$ (blue lines) of the **(a)** MLP, **(b)** LeNet, **(c)** ResNet18 without batch normalization, and **(d)** ViT without layer normalization on two synthetic datasets: **$\mathbf{G}_{\mathcal{F}}$ covariance** $\mathbf{x} \sim \mathcal{N}(0,\mathbf{G}_{\mathcal{F}}/\|\mathbf{G}_{\mathcal{F}}\|_2)$ and **flip($\mathbf{G}_{\mathcal{F}}$) covariance** $\mathbf{x} \sim \mathcal{N}(0,\text{flip}(\mathbf{G}_{\mathcal{F}})/\|\text{flip}(\mathbf{G}_{\mathcal{F}})\|_2)$ with random labels. A horizontal line for velocity indicates no change in the geometry.

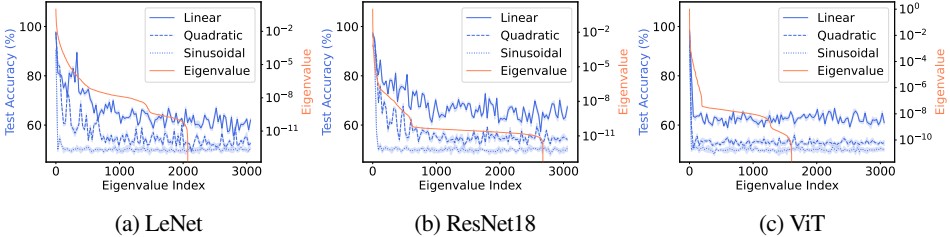

(a) LeNet     (b) ResNet18     (c) ViT

Figure 4: Test accuracy of **(a)** LeNet, **(b)** ResNet18 without batch normalization, and **(c)** ViT without layer normalization on the CIFAR-2 data with for various types of decision boundary. The $i^{th}$ x-axis point corresponds to a dataset wherein the discriminant feature is the $i^{th}$ eigenvalue of $\mathbf{G}_{\mathcal{F}}\mathbf{S}\mathbf{G}_{\mathcal{F}}$ in descending order.

$\mathbf{G}_{\mathcal{F}}$. We compare training accuracy and the velocity of this dataset to another dataset where the data is sampled from another Gaussian distribution $\mathcal{N}(0,\text{flip}(\mathbf{G}_{\mathcal{F}})/\|\text{flip}(\mathbf{G}_{\mathcal{F}})\|_2)$, with flip($\mathbf{G}_{\mathcal{F}}$) being $\mathbf{G}_{\mathcal{F}}$ with its eigenvalues flipped (i.e., in the reverse order). The experimental results are observable in Figure 3.

We observe very little change in the average geometry of the model when trained on the data with flip($\mathbf{G}_{\mathcal{F}}$) covariance compared to the data with $\mathbf{G}_{\mathcal{F}}$ covariance. This larger shift in the geometry supports Conjecture 2, showing that the geometry of the model will change very little in directions with a low correlation with $\mathbf{G}_{\mathcal{F}}$. In Section 4.2, we will see that neural networks trained using gradient descent may fail to generalize when the normal to the decision boundary aligns with directions exhibiting *geometric invariance*, as defined by Conjecture 2.

## 4.2 THE GEOMETRIC INVARIANCE HYPOTHESIS AND THE GENERALIZATION GAP

In this section, we try to investigate the relationship between GIH and generalization. In particular, we define three types of labeling for the CIFAR-10 dataset with varying complexity: linear as $y=\mathbf{sgn}((\mathbf{x}^{\top}\mathbf{u})+\mathbf{b})$, quadratic as $y=\mathbf{sgn}((\mathbf{x}^{\top}\mathbf{u})^2+\mathbf{b})$, and sinusoidal as $y=\mathbf{sgn}(\sin(\bar{\mathbf{x}}^{\top}\mathbf{u})+\mathbf{b})$, where $\bar{\mathbf{x}}$ corresponds to a standard normalized input[5], and $\mathbf{u}$ corresponds to the discriminant feature. We train and evaluate a model on each labeling function for $\mathbf{u}$ set to the eigenvectors of $\mathbf{G}_{\mathcal{F}}\mathbf{S}\mathbf{G}_{\mathcal{F}}$ in descending order, which can be seen in Figure 4. In order to make sure that our observation is not caused by a lack of separability for certain selections of $\mathbf{u}$, we set the learning rate and number of epochs in a way to ensure all models will reach $100\%$ train accuracy. Also, in order to simulate the effect of label noise, we add a small amount of noise to the input when calculating the labels (but not to the input itself during training)[6].

As we can observe in Figure 4, there is a clear correlation between the eigenvalues corresponding to $\mathbf{u}$ and the test accuracy of the model on all labeling functions, with datasets labeled based on a $\mathbf{u}$ with a smaller eigenvalue having larger generalization gap. These results further extend (Ortiz-Jiménez et al., 2020), which mainly focused on linear decision boundaries. Note that as we saw in the previous subsection, for datasets with a low correlation between the discriminant feature and $\mathbf{G}_{\mathcal{F}}$, the model is incapable of finding the actual decision boundary. As a result, it will instead rely on features in other directions to "memorize" the train samples, resulting in a lack of generalization. Consequently, the GIH explains this phenomenon, which was first observed in (Ortiz-Jiménez et al., 2020). Another interesting observation to make is that for more complex

---

[5]We added a normalization component to the sinusoidal labeling in order to control the complexity of the problem by controlling the variance of $\mathbf{x}$, which is set to $1.0$ on each channel separately.

[6]We sample i.i.d. Gaussian noise scaled on each channel separately, setting the variance to $0.2$ times the variance of the channel.

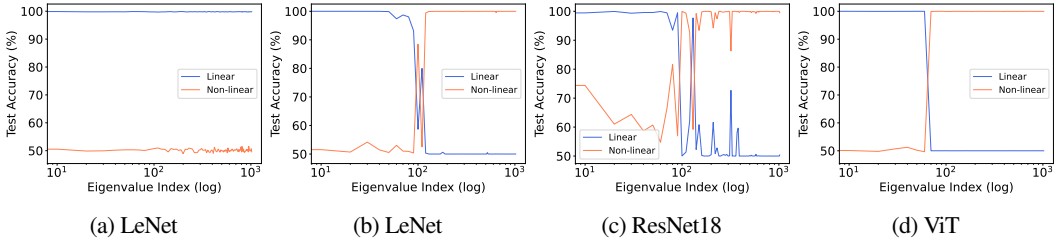

| (a) LeNet | (b) LeNet | (c) ResNet18 | (d) ViT |

Figure 5: The test accuracy of the linear and non-linear components of **(a)** MLP, **(b)** LeNet, **(c)** ResNet18, and **(d)** ViT on the synthetic data with both linear and non-linear components. The x-axis corresponds to the eigenvalue index in descending order.

decision boundaries, GIH will have a "sharper" impact on generalization, meaning that the decision boundary needs to reside in $\mathbf{u}$s with larger eigenvalues in order for the model to detect the decision boundary. This observation appears to be separate from GIH, and best explained by the SBH. However, as we will see in the next subsection, the geometric inductive biases need to be considered when talking about SBH. In App. A.7, you can observe the relationship between GIH and generalization in the presence of "isotropic" data.

## 4.3 RETHINKING THE SIMPLICITY BIAS HYPOTHESIS

The simplicity bias hypothesis (SBH) argues that neural networks prefer to learn features with simpler relationships with the predictive variable. However, as we observed in the previous subsection, depending on the direction of these features, the model may not be capable of learning them at all. So in order to reconcile the SBH with GIH, we design the following experiment involving synthetic data formulated as $\mathbf{x} = \epsilon \mathbf{u}_1 y + \mathbf{z}_y^{\mathbf{u}_2, \mathbf{u}_3} + \omega$, with Gaussian noise $\omega \sim \mathcal{N}\left(0, \sigma_\omega^2 \left(\mathbf{I} - \mathbf{u}_1 \mathbf{u}_1^\top - \mathbf{u}_2 \mathbf{u}_2^\top - \mathbf{u}_3 \mathbf{u}_3^\top\right)\right)$ for orthogonal $\mathbf{u}_1, \mathbf{u}_2, \mathbf{u}_3$. We define $\mathbf{z}_y$ as $\mathbf{z}_y^{\mathbf{u}_2, \mathbf{u}_3} = r_y \cdot \frac{\alpha \cdot \mathbf{u}_2 + \beta \cdot \mathbf{u}_3}{\|\alpha \cdot \mathbf{u}_2 + \beta \cdot \mathbf{u}_3\|_2}$, with $\alpha$ and $\beta$ being standard normal variables, and $r_1 \neq r_{-1}$ for uniformly distributed $y \in \{-1, 1\}$.

Therefore, in this experiment we have a linear discriminant and a non-linear discriminant feature in the form of $\mathbf{u}_1$ and $\mathbf{z}_y$. We select $\mathbf{u}_2$ and $\mathbf{u}_3$ from eigenvectors of $\mathbf{G}_\mathcal{F}$ with large eigenvalues, while changing $\mathbf{u}_1$ according to the eigenvectors of $\mathbf{G}_\mathcal{F}$. Figure 5 reports the test accuracy for the linear and non-linear components. In agreement with GIH, we can see that when $\mathbf{u}_1$ has a high correlation with $\mathbf{G}_\mathcal{F}$, the model always selects the linear component since the linear component has perfect accuracy while the non-linear component does not do better than a random guess. However, as the correlation between $\mathbf{u}_1$ and $\mathbf{G}_\mathcal{F}$ decreases, we observe that the model becomes gradually more reliant on the non-linear component. Therefore, SBH holds true when both components are on the same scale in terms of correlation with $\mathbf{G}_\mathcal{F}$. But when the linear component has a low correlation with $\mathbf{G}_\mathcal{F}$, the model learns the non-linear component contrary to what SBH would predict.

## 5 WHAT CAN GEOMETRY TELL US ABOUT THE SAMPLES

In this section, we provide an application-based approach to the implications of the GIH. Specifically, we focus on the implications of the GIH on the feature space and the distribution of the samples and try to see if the distribution of features and data can be related to the generalization ability of the model through GIH.

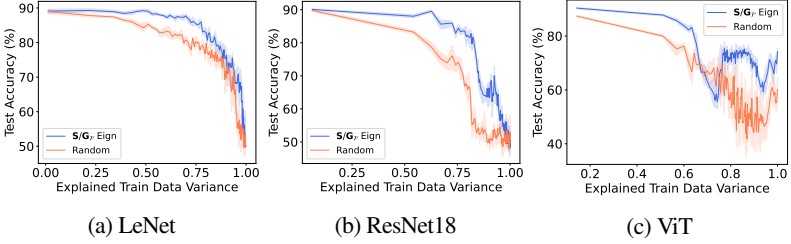

| (a) LeNet | (b) ResNet18 | (c) ViT |

Figure 6: Test accuracy of **(a)** LeNet, **(b)** ResNet18 without batch normalization, and **(c)** ViT without layer normalization for the feature distribution experiment. We report results for CIFAR-2, with features eliminated up to the index number of the generalized eigenvectors of $\mathbf{G}_\mathcal{F}$ and $\mathbf{S}$. For a fair comparison and using random orthogonal directions, we delete features until similar variability is removed from the data support.

**Feature importance.** A natural question that arises from the geometric invariance hypothesis is: *will eliminating features with the lowest possible correlation with $\mathbf{G}_\mathcal{F}$ from the data have an impact on the performance?* In order to answer this question, we design the following experiment. We find the directions with the lowest correlation with $\mathbf{G}_\mathcal{F}$ and the highest correlation with $\mathbf{S}$ by solving the generalized eigenvalue decomposition problem over $\mathbf{G}_\mathcal{F}$ and $\mathbf{S}$. Using the resulting eigenvectors[7], we perform a simple orthogonal projection on the data to eliminate the features residing in those directions. More concretely, for a sample $\mathbf{x}$ and the set of spanning vectors $\mathbb{V}_{\text{GIH}} = \{\mathbf{v}_1^n, \mathbf{v}_2^n, ..., \mathbf{v}_D^n\}$, we set $\mathbf{x}^{\perp\text{GIH}} = \mathbf{Q}_k^{\text{GIH}}\mathbf{x}$, where $\mathbf{Q}_k^{\text{GIH}} = \mathbf{I}_D - \sum_{i=1}^k \mathbf{v}_i \mathbf{v}_i^\top$. We perform the orthogonalization on both the train and the test data, essentially eliminating the features from the data. As a baseline method, we also sample a set of spanning vectors $\mathbb{V}_{\text{Rand}}$ from a random Gaussian distribution orthogonalized using the Gram-Schmidt procedure. However, for a fair comparison, for a given $\mathbf{Q}_k^{\text{GIH}}$, we first find an index $k'$ where the explained variance by $\{\mathbf{v}_1^r, \mathbf{v}_2^r, ..., \mathbf{v}_{k'}^r\}$ on $\mathbf{S}$ becomes no more than the explained variance by $\{\mathbf{v}_1^n, \mathbf{v}_2^n, ..., \mathbf{v}_{k'}^n\}$, and then set $\mathbf{Q}_k^{\text{Rand}} = \mathbf{I}_D - \sum_{i=1}^{k'} \mathbf{v}_i \mathbf{v}_i^\top$. We then train and evaluate the model on the orthogonalized dataset. The results from CIFAR-2 are reported in Figure 6.

From the experiment results we can observe that eliminating features based on their correlation with the geometry at initialization causes a lower drop in performance than eliminating them randomly. Therefore, we can safely claim that it is beneficial to take GIH into account when performing dimensionality reduction on data.

**Sample importance.** In this experiment, we try to understand whether there is a relationship between the contribution of a datapoint to the performance of the model and the amount of variation it has in the invariant directions of the geometry. Specifically, given a datapoint $\mathbf{x}$, we measure its correlation with the initial geometry $\mathbf{G}_\mathcal{F}$ as: $\mathbf{Score}(\mathbf{x}) = (\mathbf{x}/\|\mathbf{x}\|_2)^\top \mathbf{G}_\mathcal{F}(\mathbf{x}/\|\mathbf{x}\|_2)$.

| Model | Percentage of Removed Data (%) | | | |
|---|---|---|---|---|
| | 30 | 50 | 70 | 90 |
| **LeNet-R** | 61.7±0.4 | 56.6±0.3 | 52.4±0.3 | 45.1±0.3 |
| **LeNet-S** | **62.5±0.2** | **58.5±0.4** | **53.5±0.3** | **45.9±0.2** |
| **ResNet18-R** | 75.8±0.6 | 71.8±0.4 | 65.1±0.3 | 52.4±0.4 |
| **ResNet18-S** | **77.0±0.2** | **73.2±0.4** | **66.2±0.3** | **54.4±0.2** |
| **ViT-R** | 57.5±0.1 | 51.5±0.1 | 45.5±0.1 | 35.6±0.1 |
| **ViT-S** | **58.4±0.1** | **52.2±0.2** | **46.5±0.1** | **36.2±0.1** |

Table 1: The test accuracy of LeNet, ResNet18 without batch normalization, and ViT without layer normalization for the data distribution experiment. We report mean performance along with a $68\%$ confidence interval for CIFAR-10, with datapoints eliminated randomly (**R**) or based on their score value (**S**). The best performance is boldface.

Then, we sort the datapoints according to their $\mathbf{Score}(\cdot)$ value and eliminate the ones with the smallest score. We compare the results to a baseline wherein we randomly eliminate the datapoints. You can see the results of this experiment in Table 1. For a more detailed experiment please refer to Table 2 in the appendix.

As evident by the experimental results, there is a clear relationship between the contribution of a datapoint to the performance and its $\mathbf{Score}(\cdot)$ value. While the difference in performance between the random baseline and our geometry-based score is somewhat modest (at most about $2\%$), we note two factors are not considered in our $\mathbf{Score}(\cdot)$: 1) whether the features present in a datapoint with high correlation with the initial geometry are actually discriminant or not, and 2) whether by eliminating a datapoint, we lose a rare feature that is not frequent in the other train samples. Considering these factors may improve the performance further, which is out of the scope of this paper.

## 6 Conclusion and Future Work

In this paper, we have provided theoretical and empirical results that investigate the changes in the input space geometry of a neural network during training. Based on these results, we proposed the Geometric Invariance Hypothesis, which argues that depending on the architecture, the changes in the input geometry of the model can be limited to a small subspace of the input space. After empirically supporting this hypothesis, we provided several experimental pieces of evidence to show the impact of geometric invariance on the generalization of deep neural networks. Considering the practical impacts of GIH, as discussed in Section 5, we expect our paper to provide valuable information to the study of inductive biases in neural networks, the role of data and its conditioning on generalization, and architecture design.

For future work, we point to several possible extensions of our results. Firstly, we note that the theoretical results provided can be applied to more complex architectures. Secondly, other optimization methods and their implications for GIH can be considered. Thirdly, we note the possible impact of GIH on other subfields of machine learning, such as AutoML and neural architecture search, the student-teacher framework, and foundation models as another important issue to consider.

---

[7]We perform an orthogonalization procedure on the resulting eigenvectors to ensure they can span the input space.

ACKNOWLEDGMENT

We would like to thank Hadi Daneshmand, Guillermo Ortiz-Jiménez, and Felix Sarnthein for the helpful discussions and comments. We also thank Mostafa Dehghani for his contributions to this work in its earlier phase. Antonio Orvieto acknowledges the financial support of the Hector Foundation.

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

Table 2: The test accuracy of LeNet, ResNet18 without batch normalization, and ViT without layer normalization for the data distribution experiment. We report results for CIFAR-10, with datapoints eliminated randomly (**R**) or based on their score value (**S**). We report the mean performance along with a $68\%$ confidence interval for each setting. The best performance is boldface.

| Model | Percentage of Removed Data (%) | | | | | | | | |
|---|---|---|---|---|---|---|---|---|---|
| | 10 | 20 | 30 | 40 | 50 | 60 | 70 | 80 | 90 |
| **LeNet-R** | 65.0±0.4 | 63.95±0.3 | 61.7±0.4 | 58.9±0.4 | 56.6±0.3 | 54.8±0.3 | 52.4±0.3 | 49.5±0.2 | 45.1±0.3 |
| **LeNet-S** | **65.9±0.5** | **65.1±0.6** | **62.5±0.2** | **60.9±0.46** | **58.5±0.4** | **56.4±0.1** | **53.5±0.3** | **50.8±0.2** | **45.9±0.2** |
| **ResNet18-R** | 78.6±0.2 | 76.8±0.3 | 75.8±0.6 | 73.9±0.3 | 71.8±0.4 | 68.8±0.1 | 65.1±0.3 | 60.3±0.5 | 52.4±0.4 |
| **ResNet18-S** | **79.6±0.2** | **78.4±0.3** | **77.0±0.2** | **75.9±0.3** | **73.2±0.4** | **70.0±0.3** | **66.2±0.3** | **61.5±0.4** | **54.4±0.2** |
| **ViT-R** | 62.0±0.1 | 60.0±0.2 | 57.5±0.1 | 54.6±0.1 | 51.5±0.1 | 48.7±0.0 | 45.5±0.1 | 41.3±0.1 | 35.6±0.1 |
| **ViT-S** | **62.9±0.1** | **60.6±0.1** | **58.4±0.1** | **55.4±0.1** | **52.2±0.2** | **49.7±0.1** | **46.5±0.1** | **42.1±0.0** | **36.2±0.1** |

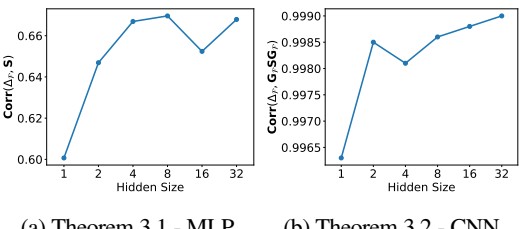

(a) Theorem 3.1 - MLP   (b) Theorem 3.2 - CNN

Figure 7: The correlation between $\Delta_{\mathcal{F}}$ **(a)** for the MLP introduced in Theorem 3.1 and **S**, and **(b)** the CNN introduced in Theorem 3.2 and $\mathbf{G}_{\mathcal{F}}\mathbf{SG}_{\mathcal{F}}$ at initialization. The experiment is performed on CIFAR-2. The x-axis corresponds to the width of the model (i.e., hidden size in MLP and number of channels in CNN), while the y-axis shows the correlation based on Frobenius cosine similarity.

## A   APPENDIX

### A.1   RELATED WORKS

Since the seminal work of (Zhang et al., 2017), there has been a lot of research aimed at reconciling the classical notion of bias-variance hypothesis and the uniform convergence theory with the success of deep neural networks (Neyshabur et al., 2017; Nagarajan & Kolter, 2019; Belkin et al., 2018; Kalimeris et al., 2019; Bachmann et al., 2021; Chiang et al., 2023). Regarding this issue, there are two schools of thoughts that either attribute this observation to an inductive bias introduced by the gradient-based optimization methods used in deep learning (Neyshabur et al., 2017; Barrett & Dherin, 2021; Smith et al., 2021), or the "volume hypothesis" arguing for the inherently larger volume of "good" solutions that generalize well (Chiang et al., 2023). However, there is one certainty with regards to this issue: the existence of "inductive biases" in these networks that result in a large decrease in the hypothesis class of solutions provided by the model, causing generalization in practice.

In search of these inductive biases, some research has been focused on the optimization of neural networks (Neyshabur et al., 2017; Arora et al., 2019; Guille-Escuret et al., 2023), which inevitably deals with the geometry of these models in the parameter space (Arpit et al., 2017). On the other hand, the introduction of neural tangent kernels allowed a link to be established between the rich literature of kernel methods (Bartlett & Mendelson, 2002) and the asymptotic behavior of neural networks at infinite width (Jacot et al., 2018). This also gave rise to some research concerned with inductive biases of deep models in the kernel space, which is also mainly focused on the geometry of parameter space (Jacot et al., 2018; Bietti & Mairal, 2019; Ortiz-Jiménez et al., 2021). However, there are also some research on the spectral inductive biases of neural networks, which provide interesting but intangible results for the type of frequencies used by these models in practice (Basri et al., 2019; Wen & Jacot, 2024).

The limited research on the geometry of deep neural networks in the input space is our main motivation for this work. The closest line of research to our paper are the simplicity bias hypothesis (SBH) (Pérez et al., 2019), and the neural anisotropy directions (Ortiz-Jiménez et al., 2020). The SBH (Arpit et al., 2017; Pérez et al., 2019; Kalimeris et al., 2019) has become a cornerstone of deep learning theory, with implication for generalization and robustness of deep neural networks (Kalimeris et al., 2019; Shah et al., 2020). Specifically, SBH argues that neural networks learn functions of increasing complexity from the

data. Therefore, in the presence of multiple solutions to the problem with various complexities, deep neural networks opt for the simplest one. In this paper, we see that while SBH is still a valid hypothesis for the inductive biases of neural networks, the geometry of the model in the input space - which is determined by its architecture - also plays a role in the "preference" of deep neural networks for simpler functions. Meaning that, depending on the architecture, there are more complex solutions that may be learned by the model before simpler ones. This behavior is due to the geometry of the model remaining invariant in certain directions of the input space. We dub this behavior *geometric invariance*, and our hypothesis that introduces this behavior as a geometric inductive bias of neural networks, the *Geometric Invariance Hypothesis* (GIH).

The neural anisotropy directions (NADs) as defined in (Ortiz-Jiménez et al., 2020) touch on the concept of geometric invariance for training on linearly separable datasets. Specifically, (Ortiz-Jiménez et al., 2020) observe that in the case of linearly separable data in specific direction **u**, depending on the architecture, neural networks are incapable of generalizing when **u** has a low correlation with the covariance of the gradient of the model w.r.t. input at initialization. This observation can be seen as a specific version of GIH limited to the linearly separable data. However, unlike our paper which motivates GIH from the perspective of investigating the geometry of a neural network, (Ortiz-Jiménez et al., 2020) explain their observations regarding GIH through the concept of discriminative dipoles, which corresponds to a pair of data points residing on the opposite sides of a linear decision boundary. Therefore, given that our paper does not make assumptions about the task, GIH can be seen as a generalization of the concept of NADs.

## A.2 SUM OF SQUARED ERROR LOSS AND ITS DYNAMICS

Following related works (Jacot et al., 2018; Lee et al., 2019; Arora et al., 2019) we use the sum of squared error (SSE) loss between the labels and the prediction of the model for the simplicity of analysis. However, as we see in the experiments, in practice our analysis holds for cross-entropy loss as well. The SSE loss can be written as:

$$\mathcal{L}(\theta) = \frac{1}{2}\sum_{\mu=1}^{m}(f_\theta(\mathbf{x}_\mu) - y_\mu)^2. \tag{4}$$

In this case, we can write the dynamics of the parameter as:

$$\dot{\theta} = -\sum_{\mu=1}^{m}\nabla_\theta f_\theta(\mathbf{x}_\mu)(f_\theta(\mathbf{x}_\mu) - y_\mu). \tag{5}$$

## A.3 LOSS CURVATURE IN THE INPUT SPACE

An important source of information about the geometry of a neural network in the input space is the Hessian of the loss w.r.t. the input, which contains information about the curvature. Let $\nabla_\mathbf{x}^2$ be the Hessian operator. Using the chain rule on $\ell(f_\theta(\mathbf{x}), y)$, we can write:

$$\nabla_\mathbf{x}^2\ell(f_\theta(\mathbf{x}), y) = \frac{\partial\ell}{\partial f}(f_\theta(\mathbf{x}), y)\cdot\nabla_\mathbf{x}^2 f_\theta(\mathbf{x}) + \frac{\partial^2\ell}{\partial f^2}(f_\theta(\mathbf{x}), y)\nabla_\mathbf{x}f_\theta(\mathbf{x})\nabla_\mathbf{x}f_\theta(\mathbf{x})^\top.$$

We can factorize the two components determining the curvature of the loss in the input space in two: 1) global model curvature (Hessian, for scalar $f_\theta(\mathbf{x})$), and 2) local model curvature (gradient outer-product). Assuming that we're in the over-parameterized setting with $f_\theta(\cdot)$ belonging to a family of neural networks, we can expect the magnitude of the first derivative of the loss function w.r.t. the model to reach near 0 at some point during the training procedure. Therefore, starting at some point in the training, we continue to have:

$$\nabla_\mathbf{x}^2\ell(f_\theta(\mathbf{x}), y) \approx \frac{\partial^2\ell}{\partial f^2}(f_\theta(\mathbf{x}), y)\nabla_\mathbf{x}f_\theta(\mathbf{x})\nabla_\mathbf{x}f_\theta(\mathbf{x})^\top. \tag{6}$$

So we can look at the gradient outer-product of the model as a low-rank approximation of the geometry of the model in the input space[8]. In order to isolate the effect of *architecture* and other *non-stochastic components*

---

[8]Note that this is very similar to the Gauss-Newton approximation of the loss Hessian w.r.t. the parameters, which is extensively used in optimization (Martens, 2020). However, here we are using this approximation solely to understand the geometry of the model in the input space.

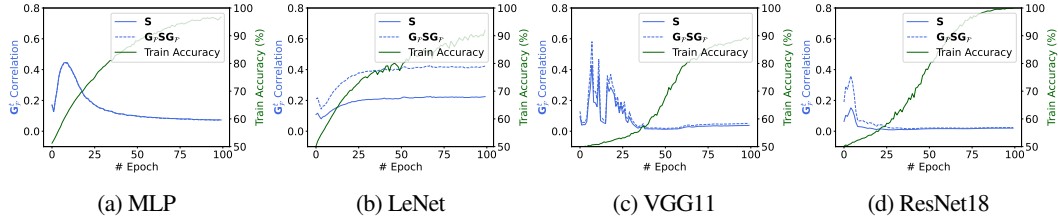

Figure 8: The correlation between $\mathbf{G}_{\mathcal{F}}^t = \mathbf{G}_{\mathcal{F},\mathcal{N}(0,\mathbf{I})}^t$ and $\mathbf{S}$ and $\mathbf{G}_{\mathcal{F}}\mathbf{SG}_{\mathcal{F}}$ for the **(a)** MLP, **(b)** LeNet, **(c)** VGG11 without batch normalization, and **(d)** ResNet18 without batch normalization on CIFAR-2 with random labeling.

in the model (e.g. the optimization method, initialization method, etc.) that influence the "geometric induc­tive biases" of the model, we can marginalize the effect of the parameters out. Assuming that the second-derivative of the loss function w.r.t. the model output has a small variation over $\mathcal{T}_t$[9], then we can write:

$$\mathbb{E}_{\theta \sim \mathcal{T}_t}\left[\nabla_{\mathbf{x}}^2 \ell(f_\theta(\mathbf{x}),y)\right] \propto \mathbf{G}_{\mathcal{F}}^t(x).$$

Therefore, by looking at the average geometry at point $\mathbf{x}$ and time $t$ through $\mathbf{G}_{\mathcal{F}}^t(\mathbf{x})$, we will have an increasingly accurate approximation of the loss curvature in the input at point $\mathbf{x}$ and time $t$ up to a constant. Given a probing distribution $\mathcal{P}$, we can consider $\mathbb{E}_{\mathbf{x}\sim\mathcal{P},\theta\sim\mathcal{T}_t}\left[\nabla_{\mathbf{x}}^2 \ell(f_\theta(\mathbf{x}),y)\right]$ as the **average geometry**, which we estimate by $\mathbf{G}_{\mathcal{F},\mathcal{P}}^t = \mathbb{E}_{x\sim\mathcal{P}}[\mathbf{G}_{\mathcal{F}}^t(\mathbf{x})]$.

Based on this approximation of the geometry in the input space, we will formulate a function informing us about the changes in the geometry as well. Let us start with the derivative of (6) w.r.t. time in order to get the changes in the loss curvature through the gradient flow model:

$$\frac{\mathrm{d}\nabla_{\mathbf{x}}^2 \ell}{\mathrm{d}t}(f_\theta(\mathbf{x}),y) \approx \frac{\partial^3 \ell}{\partial f^3}(f_\theta(\mathbf{x}),y)\cdot\left(\nabla_\theta f_\theta(\mathbf{x})^\top \dot\theta\right)\nabla_{\mathbf{x}}f_\theta(\mathbf{x})\nabla_{\mathbf{x}}f_\theta(\mathbf{x})^\top$$
$$+ \frac{\partial^2 \ell}{\partial f^2}(f_\theta(\mathbf{x}),y)\nabla_{\mathbf{x},\theta}^2 f_\theta(\mathbf{x})\dot\theta\nabla_{\mathbf{x}}f_\theta(\mathbf{x})^\top + \frac{\partial^2 \ell}{\partial f^2}(f_\theta(\mathbf{x}),y)\nabla_{\mathbf{x}}f_\theta(\mathbf{x})\dot\theta^\top\nabla_{\mathbf{x},\theta}f_\theta(\mathbf{x})^\top.$$

Assuming that the third-derivative of the loss function w.r.t. the model output is close to zero[10], and assuming the independence of $\mathcal{T}_t$ and the second derivative of the loss, we can get the expected value of the two sides w.r.t. the parameters to isolate the effect of inductive biases, which gives us:

$$\mathbb{E}_{\theta\sim\mathcal{T}_t}\left[\frac{\mathrm{d}\nabla_{\mathbf{x}}^2 \ell}{\mathrm{d}t}(f_\theta(\mathbf{x}),y)\right] \propto \Delta_{\mathcal{F}}^t(\mathbf{x}).$$

Therefore, based on the loss curvature evolution function and given a probing distribution $\mathcal{P}$, we define the **average geometry evolution** as $\mathbb{E}_{\mathbf{x}\sim\mathcal{P},\theta\sim\mathcal{T}_t}\left[\frac{\mathrm{d}\nabla_{\mathbf{x}}^2 \ell}{\mathrm{d}t}(f_\theta(\mathbf{x}),y)\right]$, which we estimate as $\Delta_{\mathcal{F},\mathcal{P}}^t$ up to a constant.

### A.4 INDEPENDENCE OF $\Delta_{\mathcal{F}}(\mathbf{x})$ FROM LABELS

In this section, we try to investigate the dependence between the average geometry evolution at initialization $\Delta_{\mathcal{F}}$ and the train labels $y_\mu$, i.e., the relationship between the changes in the geometry at initialization and the task. Firstly, we provide the following proposition to show that at the first gradient descent step, this independence can be proved provided that the model has a single-layer perceptron for its classification layer and is trained via SSE loss.

**Proposition A.1.** *Let $\mathcal{F}$ be a family of neural networks with a single layer perceptron initialized i.i.d. from a zero-mean Gaussian as the classification layer, trained via SSE. Then $\Delta_{\mathcal{F}}(\mathbf{x})$ is independent of train labels.*

**Proof.** Let $f_\theta(\mathbf{x})$ be the model defined by the proposition:

$$f_\theta(\mathbf{x}) = \sum_{i=1}^{n}\omega_i h_\phi^i(\mathbf{x}),$$

---

[9]This can happen, for instance, when we have the SSE loss where the second-derivative is constant.

[10]For instance, in the cross entropy loss the third derivative will be equal to the second derivative of the softmax function, which is bounded by $1/8$, while it is equal to zero in the SSE loss.

where $h_\phi(\mathbf{x})$ is the hidden representation in the penultimate layer of the model, $h_\phi^i(\mathbf{x})$ is its $i^{th}$ element, and $\omega$ corresponds to the weights of the classification layer. Therefore, $\theta$ corresponds to the concatenation of $\omega$ and $\phi$. The required derivatives are:

$$\nabla_x f_\theta(\mathbf{x}) = \sum_{i=1}^{n} \omega_i \nabla_x h_\phi^i(\mathbf{x}),$$

$$\frac{\partial f_\theta}{\partial \omega_i}(\mathbf{x}) = h_\phi^i(\mathbf{x}),$$

$$\frac{\partial \nabla_{\mathbf{x}} f_\theta}{\partial \omega_i}(\mathbf{x}) = \nabla_{\mathbf{x}} h_\phi^i(\mathbf{x}),$$

$$\nabla_\phi f_\theta(\mathbf{x}) = \sum_{i=1}^{n} \omega_i \nabla_\phi h_\phi^i(\mathbf{x}),$$

$$\nabla_{\mathbf{x},\phi} f_\theta(\mathbf{x}) = \sum_{i=1}^{n} \omega_i \nabla_{\mathbf{x},\phi} h_\phi^i(\mathbf{x}).$$

So we can write $\Delta_\mathcal{F}(\mathbf{x}) = \mathbf{A}_\mathcal{F}(\mathbf{x}) + \mathbf{A}_\mathcal{F}(\mathbf{x})^\top$, where we have:

$$\mathbf{A}_\mathcal{F}(\mathbf{x}) = -\sum_{\mu=1}^{m}\left(\sum_{i_1=1}^{n}\sum_{i_3=1}^{n}\sum_{i_4=1}^{n}\mathbb{E}_\theta\left[\omega_{i_3}h_\phi^{i_1}(\mathbf{x}_\mu)\left(\omega_{i_4}h_\phi^{i_4}(\mathbf{x}_\mu) - y_\mu\right)\nabla_{\mathbf{x}}h_\phi^{i_1}(\mathbf{x})\nabla_{\mathbf{x}}h_\phi^{i_3}(\mathbf{x})^\top\right]\right. \tag{7}$$
$$\left. + \mathbb{E}_\theta\left[\omega_{i_1}\omega_{i_2}\omega_{i_3}\left(\omega_{i_4}h_\phi^{i_4}(\mathbf{x}_\mu) - y_\mu\right)\nabla_{\mathbf{x},\phi}h_\phi^{i_1}(\mathbf{x})\nabla_\phi h_\phi^{i_2}(\mathbf{x}_\mu)\nabla_{\mathbf{x}}h_\phi^{i_3}(\mathbf{x})^\top\right]\right).$$

Note that the terms related to $y_\mu$ in (7) contain an odd number of $\omega_i$s. Considering that $\omega$ is i.i.d. and zero-mean Gaussian, its odd moments are equal to zero, which means these terms will be eliminated. This completes our proof. □

As we can observe from Proposition A.1, $\Delta_\mathcal{F}$ is provably independent of the task and only depends on inputs. In order to show that this behavior remains consistent at the initial stages of training, we formulate an experiment similar to Section 3. Specifically, we try to show that even if there's nothing to learn from the data, and the task is fully about the "memorization" of the data, $\Delta_\mathcal{F}$ still follows the behavior outline in Conjecture 1. So we relabel the CIFAR-2 dataset introduced in Section 3 using a uniform random distribution over $\{-1, 1\}$. We then report the correlation between $\mathbf{G}_\mathcal{F}^t$ and $\mathbf{G}_\mathcal{F}\mathbf{S}\mathbf{G}_\mathcal{F}$ along with the training accuracy for each epoch. The results are visualized in Figure 8. We point out that these results clearly indicate the universality of our theoretical motivations, meaning that the behavior outlined in the theorems in Section 3 culminating in the introduction of Conjecture 1 are independent of the task.

## A.5 THE STRUCTURE OF $\mathbf{G}_\mathcal{F}$

In this section we try to investigate how the architecture affects $\mathbf{G}_\mathcal{F}$. First, we will present the following theorem for the case where $\mathcal{F}$ corresponds to an MLP or CNN with ReLU non-linearity and without pooling layers or skip-connections:

**Theorem A.2.** *Let $\mathcal{F}$ be the family of MLPs or CNNs with ReLU non-linearity, without pooling layers or skip-connections. Then we have $\mathbf{G}_\mathcal{F}(\mathbf{x}) = c \cdot \mathbf{I}$ for some constant $c$. For the proof, please refer to App. A.11.4.*

As we can observe in Theorem A.2, in the case where $\mathcal{F}$ does not contain any pooling, self-attention, or skip-connections, the matrix $\mathbf{G}_\mathcal{F}$ will correspond to the identity matrix, and therefore, is a full rank matrix.

However, this is not the case in other types of architecture components. For instance, the following theorem will show that the introduction of an average pooling layer will introduce a structure to $\mathbf{G}_\mathcal{F}$:

**Theorem A.3.** *Let $\mathcal{F}$ be the family of CNNs with a linear convolution layer and a single kernel, followed by a batch normalization layer and a global average pooling layer. Also, let $\mathbf{M}_j$ correspond to a binary matrix that simulates the convolutional operation for the $j^{th}$ filter. Assuming we use SSE loss, we have:*

$$\mathbf{G}_\mathcal{F} \propto \sum_{j_1}\sum_{j_2}\mathbf{I}_{j_1,j_2},$$

where $\mathbf{I}_{j_1,j_2}$ is a diagonal binary matrix corresponding to the overlap of the $j_1^{th}$ and $j_2^{th}$ patch of the data. For the proof, please refer to App. A.11.5.

From Theorem A.3 we observe that $\mathbf{G}_{\mathcal{F}}$ will have a diagonal form, with each diagonal element proportional to how many times the corresponding input element has been in the receptive field of the convolution filter. In the case of batch normalization, the structure of $\mathbf{G}_{\mathcal{F}}$ can be potentially even more complex:

**Corollary A.4.** *Let $\mathcal{F}$ be the family of CNNs with a linear convolution layer and a single kernel, followed by a batch normalization layer and a global average pooling layer. Also, let $\mathbf{M}_j$ correspond to a binary matrix that simulates the convolutional operation for the $j^{th}$ filter. Assuming that the train data is zero-mean and we use SSE loss, then we have:*

$$\mathbf{G}_{\mathcal{F}} \propto \sum_{j_1}\sum_{j_2}\mathbf{M}_{j_1}\mathbb{E}_{\theta}\left[\frac{\theta\theta^{\top}}{\sigma_{j_1}(\theta)\cdot\sigma_{j_2}(\theta)}\right]\mathbf{M}_{j_2}^{\top},$$

*where $\sigma_j(\theta)=\mathbb{V}\mathrm{ar}_{\mathbf{x}_{\mu}\in\mathcal{D}_T}(\theta^{\top}\mathbf{M}_j^{\top}x_{\mu})$. Therefore, the terms in $\mathbf{G}_{\mathcal{F}}$ corresponding to each pair of indices $j_1,j_2$ is bounded as:*

$$\frac{1}{\sqrt{\lambda_{min}^{j_1}\lambda_{min}^{j_2}}}\mathbf{I}_{j_1,j_2}\succeq\mathbf{M}_{j_1}\mathbb{E}_{\theta}\left[\frac{\theta\theta^{\top}}{\sigma_{j_1}(\theta)\cdot\sigma_{j_2}(\theta)}\right]\mathbf{M}_{j_2}^{\top}\succeq\frac{1}{\sqrt{\lambda_{max}^{j_1}\lambda_{max}^{j_2}}}\mathbf{I}_{j_1,j_2},$$

*where $\lambda_{min}^j,\lambda_{max}^j$ are the smallest and largest eigenvalues of the covariance of the $j^{th}$ patch of the data. For the proof, please refer to App. A.11.6.*

The Corollary A.4 indicates that introducing normalization to the model will result in a re-weighting of the influence of the input elements based on their variance onto $\mathbf{G}_{\mathcal{F}}$, atop the receptive field-related structure introduced by Theorem A.3. In our experience, in cases where the covariance of the training data has a structure, this will result in the normalization of $\mathbf{G}_{\mathcal{F}}$ w.r.t. each patch of data which depends on the kernel size in CNNs and the patch size in ViT.

Therefore, we can see that certain architecture components can introduce structure to $\mathbf{G}_{\mathcal{F}}$, which can reduce the ranking of $\mathbf{G}_{\mathcal{F}}$. This indicates a significant narrowing of the geometric inductive biases of the model on the input space, as considered by this paper, which as we observed in Figure 4, results in the model being unable to generalize in tasks wherein the decision boundary lies in directions with low correlation with $\mathbf{G}_{\mathcal{F}}$.

## A.6 VISUALIZATION OF THE AVERAGE GEOMETRY

In this section, we provide a visualization of the average geometry at initialization, i.e., $\mathbf{G}_{\mathcal{F}}$ for various architectures. As we can observe in Figure 9, an MLP has a full-rank average geometry at initialization, with the matrix $\mathbf{G}_{\mathcal{F}}$ resembling the identity matrix. On the other hand, the eigenvalues of the other architecture have a much wider range, clearly indicating a presence of structure in the average geometry.

## A.7 GIH AND GENERALIZATION GAP WITH ISOTROPIC DATA

In this section, we try to isolate the effect of GIH over the generalization gap by making the data covariance as close to "isotropic" as possible. Specifically, we design the following experiment involving synthetic data formulated as follows:

$$\mathbf{x}=\epsilon\alpha\mathbf{u}+\omega,\quad\omega\sim\mathcal{N}\big(0,\sigma^2\big(\mathbf{I}-\mathbf{u}\mathbf{u}^{\top}\big)\big),\tag{8}$$

where we have $\alpha\sim\mathcal{N}(0,1)$, and we consider $\mathbf{u}$ to be the discriminant feature, similar to Section 4.2. Note that in this setting, for $\epsilon\approx\sigma$ we can roughly say $\mathbf{x}\sim\mathcal{N}\big(0,\sigma^2\mathbf{I}\big)$, which eliminates the effect of data over the changes in geometry. Similar to the experiment in Section 4.2, we assess the generalization ability of the model for varying complexity of tasks: linear as $y=\mathbf{sgn}((\mathbf{x}^{\top}\mathbf{u})+\mathbf{b})$, quadratic as $y=\mathbf{sgn}((\mathbf{x}^{\top}\mathbf{u})^2+\mathbf{b})$, and sinusoidal as $y=\mathbf{sgn}(\sin(\bar{\mathbf{x}}^{\top}\mathbf{u})+\mathbf{b})$. Note that we can view the ratio $\epsilon/\sigma$ as a measure of the margin between the two classes, which controls the difficulty of the problem. In Figure 10, you can observe the test accuracy of the models for these tasks, with $\mathbf{u}$ selected from the eigenvectors of $\mathbf{G}_{\mathcal{F}}$ in descending order of the eigenvalues.

As we can observe, in the absence of a structure in the data covariance, the MLP model will be completely devoid of any geometric inductive bias, achieving generalization in all directions with similar equality.

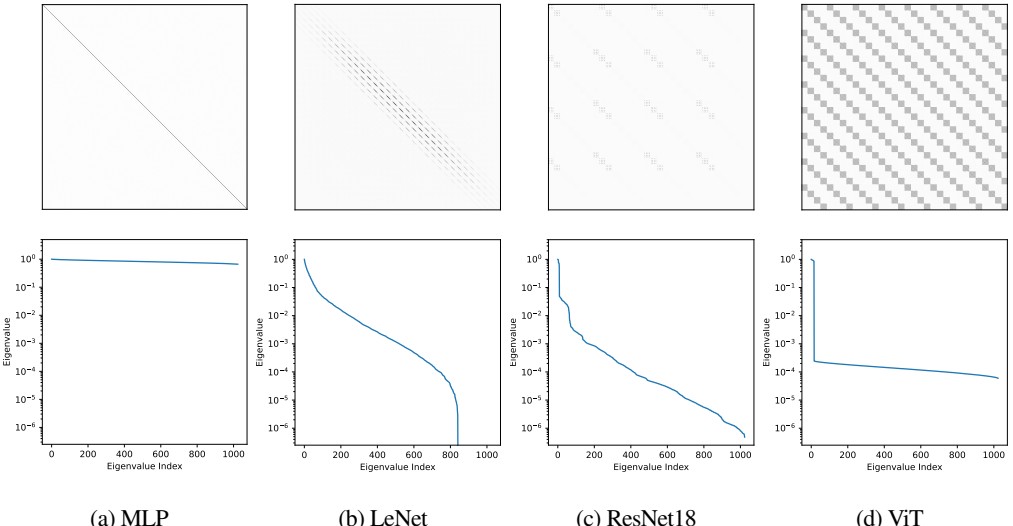

(a) MLP      (b) LeNet      (c) ResNet18      (d) ViT

Figure 9: The visualization of the average geometry at initialization ($\mathbf{G}_{\mathcal{F}}$) for **(a)** MLP, **(b)** LeNet, **(c)** ResNet18, and **(d)** ViT in the upper row, along with their eigenvalues in the lower row. The input space corresponds to a single-channel $32 \times 32$ image. Note that the visualized NAD components in (Ortiz-Jiménez et al., 2020) corresponds to the eigenvectors of this matrix. Considering Conjectures 1 and 2, we can interpret this matrix as how the variation in the data impacts the geometry during training. For instance, in MLP we observe that the average geometry is a scaled identity matrix. As a result, the data variation impacts the geometry of each input element separately. On the other hand, in LeNet the average geometry is a matrix with non-zero elements around the diagonal. As a result, we expect the data variation to impact the geometry of input elements in proximity to each other. Similarly, in ResNet and ViT we observe a periodic structure in the average geometry. As a result, we expect the data variation to impact the geometry of input elements in a cyclic manner.

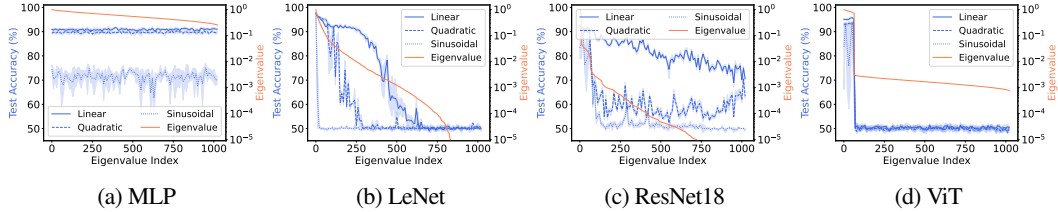

(a) MLP      (b) LeNet      (c) ResNet18      (d) ViT

Figure 10: The test accuracy for the **(a)** MLP, **(b)** LeNet, **(c)** ResNet18, and **(d)** ViT on the synthetic data. We set $\sigma = 1$ for all experiments and $\epsilon$ to 0.9 and 2.0 for the linear and quadratic decision boundaries, respectively. For the sinusoidal decision boundary, we set $\epsilon$ to 2.1 for MLP and LeNet, and 5.0 for ResNet18 and ViT.

On the other hand, in the case of the models with structured average geometry, the performance of the model in all tasks is correlated with the eigenvalue of the corresponding discriminant feature. These results provide further support for the GIH, while also confirming that the observation in Section 4.2 is due to the existence of a structure in both the data covariance and the initial geometry.

## A.8 THE EFFECT OF DATASIZE ON GIH

GIH is implicitly based on the assumption that the average geometry evolution still follows the pattern introduced in Conjecture 1 in the latter stages of training. Therefore, a natural question that may arise from this assumption and our experimental setting is that: *Can GIH be solely attributed to the over-parameterized regime?* In order to answer this question definitely, we designed several experiments to investigate the effect of datasize on GIH fully.

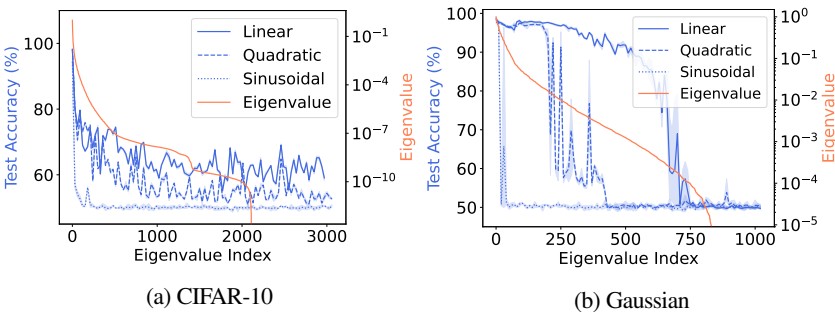

(a) CIFAR-10

(b) Gaussian

Figure 11: The test accuracy of LeNet for the **(a)** CIFAR-10 and the **(b)** synthetic Gaussian datasets introduced in Section 4.2 and App. A.7, respectively. We used the exact same training setting but with the same amount of training samples as model parameters (50k samples).

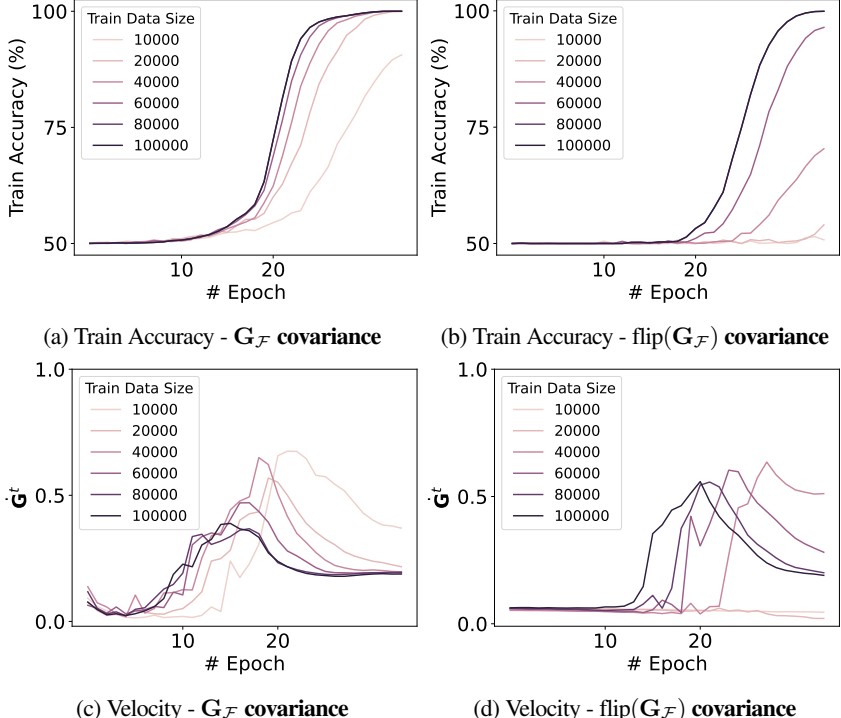

(a) Train Accuracy - $\mathbf{G}_{\mathcal{F}}$ **covariance**

(b) Train Accuracy - flip($\mathbf{G}_{\mathcal{F}}$) **covariance**

(c) Velocity - $\mathbf{G}_{\mathcal{F}}$ **covariance**

(d) Velocity - flip($\mathbf{G}_{\mathcal{F}}$) **covariance**

Figure 12: The train accuracy and velocity $\dot{\mathbf{G}}^{t}(\cdot,\cdot)$ of the ResNet18 without batch normalization on two synthetic datasets: $\mathbf{G}_{\mathcal{F}}$ **covariance** $\mathbf{x} \sim \mathcal{N}(0, \mathbf{G}_{\mathcal{F}}/\|\mathbf{G}_{\mathcal{F}}\|_{2})$ and flip($\mathbf{G}_{\mathcal{F}}$) **covariance** $\mathbf{x} \sim \mathcal{N}(0, \text{flip}(\mathbf{G}_{\mathcal{F}})/\|\text{flip}(\mathbf{G}_{\mathcal{F}})\|_{2})$ with random labels. We perform the experiments for datasizes in $\{10000, 20000, 40000, 60000, 80000, 100000\}$.

In the first step, we start by looking at the test accuracy and the effects of GIH on generalization as we observed in Section 4.2 and App. A.7 *on the verge of under-parameterization* in LeNet. You can see the results of this experiment in Figure 11. Note that in order to make the under-parameterized regime possible, we had to slightly decrease the size of LeNet by making the fully connected layers slightly smaller. Despite the larger dataset and the fact that the model is no longer in the over-parameterized regime, we can observe that the performance of the model on the test sample is still exactly as the GIH would predict. Specifically, we can still observe a "preference" to learn the features more aligned with the average geometry during training. This observation rules out over-parameterization as a condition for GIH.

We can also look at the train accuracy and how it is affected by datasize. In order to do so, we focus on ResNet18 and use the experimental settings of Section 4 for Figure 3. So we will look at the train accuracy and the velocity of the average geometry for datasizes in $\{10000, 20000, 40000, 60000, 80000, 100000\}$ for data sampled from $\mathcal{N}(0, \mathbf{G}_{\mathcal{F}}/\|\mathbf{G}_{\mathcal{F}}\|_{2})$ and $\mathcal{N}(0, \text{flip}(\mathbf{G}_{\mathcal{F}})/\|\text{flip}(\mathbf{G}_{\mathcal{F}})\|_{2})$. You can see the experiment

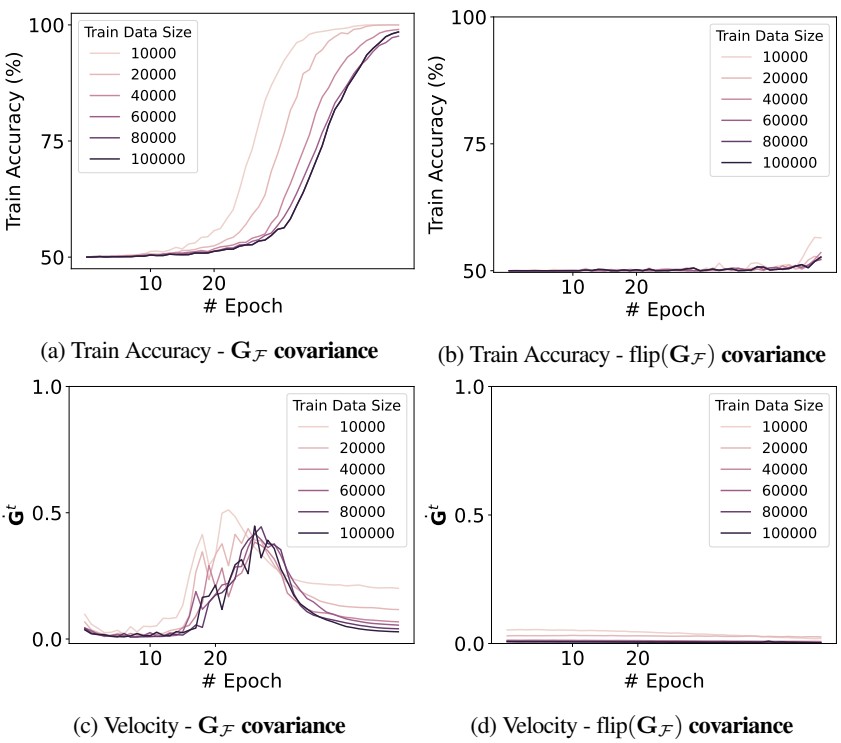

Figure 13: The train accuracy and velocity $\dot{\mathbf{G}}^t(\cdot, \cdot)$ of the ResNet18 without batch normalization on two synthetic datasets: $\mathbf{G}_{\mathcal{F}}$ **covariance** $\mathbf{x} \sim \mathcal{N}(0, \mathbf{G}_{\mathcal{F}}/\|\mathbf{G}_{\mathcal{F}}\|_2)$ and flip $(\mathbf{G}_{\mathcal{F}})$ **covariance** $\mathbf{x} \sim \mathcal{N}(0, \text{flip}(\mathbf{G}_{\mathcal{F}})/\|\text{flip}(\mathbf{G}_{\mathcal{F}})\|_2)$ with random labels. We perform the experiments for datasizes in $\{10000, 20000, 40000, 60000, 80000, 100000\}$. Furthermore, we increase the batchsize for each datasize so that the number of mini-batches would remain the same.

results in Figure 12. At first glance, the results may seem to indicate that the effects of GIH are vanishing on the **train accuracy** as the number of datapoints increases. However, we were able to refute this point. Specifically, based on these results we hypothesize that the reason the model is able to generalize on the flip($\mathbf{G}_{\mathcal{F}}$) **covariance** data is that the number of iterations are also increasing, resulting in the parameter ending up very far away from the initialization point. Therefore, it will effectively eliminate the geometric inductive biases of the model caused by initialization, resulting in the model behaving according to the geometric inductive biases of some $\mathbf{G}_{\mathcal{F}}^t$ for some very large $t > 0$[11].

In order to support our claim, we designed another similar experiment this time with a dynamic batchsize that increases with the datasize to make sure the number of iterations of SGD on the parameters remains constant. This will ensure that we are moving a similar distance in the parameter space during training for all train data sizes. You can see the experiment results in Figure 13. As evident by the results, in this experiment, the model is reverting to adhering to the behavior predicted by GIH, which supports our hypothesis for the reason behind the observation in Figure 12. In App. A.9, we will see that while the distance from initialization seems to be affecting the predictions of GIH for train accuracy, it is not the case for test accuracy.

## A.9 THE EFFECT OF LEARNING RATE ON GIH

Given that GIH is concerned with the training dynamics of neural networks, an important relationship to explore is how learning rate impacts GIH. Specifically, in this section, we are trying to understand *whether the learning rate amplifies or reduces the effect of GIH*, similar to the results of Appendix A.8. In order to answer this question, we designed the following experiment for the ResNet18 model. We follow the experimental settings of App. A.7 for the experiments presented in Figure 10, but for learning rate in $\{1.0, 0.1, 0.01, 0.001\}$. You see the experiment results in Figure 14. Note that in all these experiments, the model converges on the train data and reaches a $100\%$ train accuracy.

---

[11]Note that we are abusing our notation to make the point that the effects of GIH are still present, but with some average geometry computed at a different point during training than at initialization

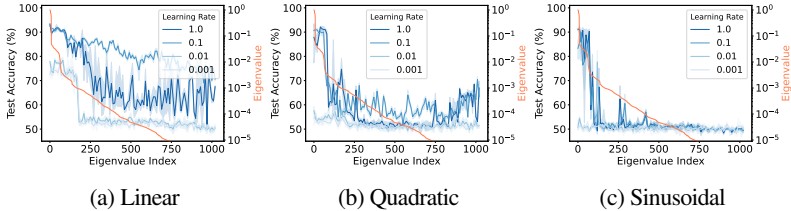

(a) Linear         (b) Quadratic         (c) Sinusoidal

Figure 14: The test accuracy of ResNet18 for the **(a)** Linear, **(b)** Quadratic, and **(c)** Sinusoidal decision boundaries on the synthetic data. We follow the same experimental setting as in App. A.7, except for the learning rate, which is chosen from the following set: $\{1.0, 0.1, 0.01, 0.001\}$.

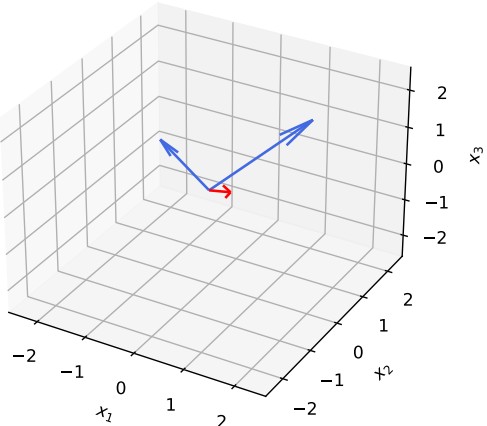

Figure 15: A visualization of the eigenvectors for the model introduced in App. A.10. The blue vectors have non-zero eigenvalues, while the red vector has zero eigenvalue.

As we can observe, a very small learning rate has a detrimental effect on the performance of the model, while larger learning rates generally generalize better. However, in all these cases we can observe that the model is following the predictions of GIH, and we can observe a "preference" to learn the features more aligned with the average geometry during training. Furthermore, these observations also support our hypothesis for the results of the experiment performed in App. A.8, as larger learning rates, and thus longer optimization paths, do not cause the effects of GIH over the **test data** to vanish.

## A.10 AN EXAMPLE

For the sake of clarity, let us consider an example of a 3-dimensional input space, a point on which we denote as:

$$\mathbf{x} = \begin{bmatrix} x_1 \\ x_2 \\ x_3 \end{bmatrix}.$$

For our model, let us assume we have a convolution layer with a single kernel in the form of $\theta = \begin{bmatrix} \theta_1 \\ \theta_2 \end{bmatrix}$, followed by an average pooling layer of size 2. Then, we can write the model in the vector form as:

$$f_\theta(\mathbf{x}) = [1/2 \quad 1/2] \begin{bmatrix} \theta_1 & \theta_2 & 0 \\ 0 & \theta_1 & \theta_2 \end{bmatrix} \begin{bmatrix} x_1 \\ x_2 \\ x_3 \end{bmatrix}$$

$$= \frac{1}{2} \cdot [\theta_1 \quad \theta_1 + \theta_2 \quad \theta_2] \begin{bmatrix} x_1 \\ x_2 \\ x_3 \end{bmatrix}.$$

So the gradient of the model w.r.t. the input will be:

$$\nabla_\mathbf{x} f_\theta(\mathbf{x}) = \frac{1}{2} \cdot \begin{bmatrix} \theta_1 \\ \theta_1 + \theta_2 \\ \theta_2 \end{bmatrix}.$$

So the outer product of the gradient with itself will be:

$$\nabla_\mathbf{x} f_\theta(\mathbf{x}) \nabla_\mathbf{x} f_\theta(\mathbf{x})^\top = \frac{1}{4} \cdot \begin{bmatrix} \theta_1^2 & \theta_1^2 + \theta_1\theta_2 & \theta_1\theta_2 \\ \theta_1^2 + \theta_1\theta_2 & (\theta_1 + \theta_2)^2 & \theta_1\theta_2 + \theta_2^2 \\ \theta_1\theta_2 & \theta_1\theta_2 + \theta_2^2 & \theta_2^2 \end{bmatrix}.$$

Now let us assume the parameters are initialized as i.i.d. zero-mean Gaussian random variables, i.e., $\theta_i \sim \mathcal{N}(0, \sigma_\theta^2)$. Then we can write:

$$\mathbf{G}_\mathcal{F} = \frac{\sigma_\theta^2}{2} \begin{bmatrix} 1 & 1 & 0 \\ 1 & 2 & 1 \\ 0 & 1 & 1 \end{bmatrix}.$$

Therefore, the average geometry at initialization is a rank-2 matrix with the following eigenvalues and eigenvectors:

- Eigenvector: $\mathbf{v}_1 = \begin{bmatrix} 1 \\ 2 \\ 1 \end{bmatrix}$, eigenvalue: $\lambda_1 = 3$

- Eigenvector: $\mathbf{v}_2 = \begin{bmatrix} -1 \\ 0 \\ 1 \end{bmatrix}$, eigenvalue: $\lambda_2 = 1$

- Eigenvector: $\mathbf{v}_3 = \begin{bmatrix} 1 \\ -1 \\ 1 \end{bmatrix}$, eigenvalue: $\lambda_3 = 0$

You can see a visualization of these eigenvectors in Figure 15. According to GIH the geometry of the model in the input space can only change in the direction of the blue vectors, which corresponds to a 2-dimensional plane. Therefore, the model is not capable of forming decision boundaries in the direction of the red vectors, thus having a geometric inductive bias towards the features residing on the plane defined by the blue vectors.

### A.11    PROOF OF THEOREMS

### A.11.1    LEMMAS

We will start the proofs by first proving four lemmas that we will be using in the rest of this section.

**Lemma A.5.** *Let $\delta \sim \mathcal{N}(0, \mathbf{I})$ be a standard Gaussian variable, and $\mathbf{a}, \mathbf{b} \in \mathbb{R}^D$ be two unit vectors. Then, as $D$ becomes larger ($D \to \infty$), and assuming $\mathbf{a}, \mathbf{b}$ are non-sparse (i.e., their magnitude is not concentrated on a few elements), we have:*

$$\mathbb{E}_\delta \left[ \mathbf{1}_{\delta^\top \mathbf{a} > 0} \mathbf{1}_{\delta^\top \mathbf{b} > 0} \delta\delta^\top \right] \approx \mathbb{E}_\delta [\mathbf{1}_{\delta^\top \mathbf{a} > 0} \mathbf{1}_{\delta^\top \mathbf{b} > 0}] \cdot \mathbf{I}_D,$$

*which becomes accurate at the rate of $\sqrt{D}$.*

**Proof.** We will start by partitioning the variability into three factors: $\delta = \alpha \cdot \mathbf{a} + \beta \cdot \mathbf{b}_\perp + \delta_\perp$, where $\mathbf{b}_\perp$ is the unit vector in the direction of the orthogonal projection of $\mathbf{b}$ on $\mathbf{a}$, and $\delta_\perp \sim \mathcal{N}(0, \mathbf{I}_D - \mathbf{a}\mathbf{a}^\top - \mathbf{b}_\perp \mathbf{b}_\perp^\top)$. Then we can write:

$$
\begin{aligned}
\mathbb{E}_\delta\left[\mathbf{1}_{\delta^\top \mathbf{a} > 0}\mathbf{1}_{\delta^\top \mathbf{b} > 0}\delta\delta^\top\right] = &\left(\mathbb{E}\left[\mathbf{1}_{\alpha > 0}\mathbf{1}_{\alpha \cdot \mathbf{a}^\top \mathbf{b}_\perp + \beta \cdot \mathbf{b}^\top \mathbf{b}_\perp > 0}\alpha^2\right] - \mathbb{E}\left[\mathbf{1}_{\alpha > 0}\mathbf{1}_{\alpha \cdot \mathbf{a}^\top \mathbf{b}_\perp + \beta \cdot \mathbf{b}^\top b_\perp > 0}\right]\right)\mathbf{a}\mathbf{a}^\top \\
&+ \left(\mathbb{E}\left[\mathbf{1}_{\alpha > 0}\mathbf{1}_{\alpha \cdot \mathbf{a}^\top \mathbf{b}_\perp + \beta \cdot \mathbf{b}^\top \mathbf{b}_\perp > 0}\beta^2\right] - \mathbb{E}\left[\mathbf{1}_{\alpha > 0}\mathbf{1}_{\alpha \cdot \mathbf{a}^\top \mathbf{b}_\perp + \beta \cdot \mathbf{b}^\top \mathbf{b}_\perp > 0}\right]\right)\mathbf{b}_\perp \mathbf{b}_\perp^\top \\
&+ \mathbb{E}\left[\mathbf{1}_{\alpha > 0}\mathbf{1}_{\alpha \cdot \mathbf{a}^\top \mathbf{b}_\perp + \beta \cdot \mathbf{b}^\top \mathbf{b}_\perp > 0}\alpha\beta\right]\left(\mathbf{b}_\perp \mathbf{a}^\top + \mathbf{a}\mathbf{b}_\perp^\top\right) \\
&+ \mathbb{E}\left[\mathbf{1}_{\alpha > 0}\mathbf{1}_{\alpha \cdot \mathbf{a}^\top \mathbf{b}_\perp + \beta \cdot \mathbf{b}^\top \mathbf{b}_\perp > 0}\right]\mathbf{I}_D.
\end{aligned}
$$

Now, we will upper-bound the coefficients of the first three terms. For the coefficient of the first term, we can write:

$$
= \mathbb{E}\left[\mathbf{1}_{\alpha > 0}\mathbf{1}_{\alpha \cdot c + \beta > 0}\alpha^2\right] - \mathbb{E}[\mathbf{1}_{\alpha > 0}\mathbf{1}_{\alpha \cdot c + \beta > 0}] \tag{9}
$$

$$
= \frac{1}{2} \cdot \left(\mathbb{E}\left[\mathbf{1}_{\alpha > 0}\left(\mathrm{erf}\left(\frac{1}{\sqrt{2}}\alpha \cdot c\right) + 1\right)\alpha^2\right] - \mathbb{E}\left[\mathbf{1}_{\alpha > 0}\left(\mathrm{erf}\left(\frac{1}{\sqrt{2}}\alpha \cdot c\right) + 1\right)\right]\right) \tag{10}
$$

$$
= \frac{1}{2} \cdot \left(\mathbb{E}\left[\mathbf{1}_{\alpha > 0}\mathrm{erf}\left(\frac{1}{\sqrt{2}}\alpha \cdot c\right)\alpha^2\right] - \mathbb{E}\left[\mathbf{1}_{\alpha > 0}\mathrm{erf}\left(\frac{1}{\sqrt{2}}\alpha \cdot c\right)\right]\right) \tag{11}
$$

$$
= \frac{1}{\pi}\mathbb{E}\left[\mathbf{1}_{\alpha > 0}\alpha\exp\left(-\frac{1}{\sqrt{2}}\alpha^2 \cdot c^2\right)\right], \tag{12}
$$

where $c = \frac{\mathbf{a}^\top \mathbf{b}_\perp}{\mathbf{b}^\top \mathbf{b}_\perp}$, (10) comes from the definition of indicator function and the error function $\mathrm{erf}(\cdot)$, (11) comes from $\alpha$ being a standard Gaussian variable which gives us $\mathbb{E}\left[\mathbf{1}_{\alpha > 0}\alpha^2\right] = \mathbb{E}[\mathbf{1}_{\alpha > 0}] = \frac{1}{2}$, and (12) comes from the following:

$$
\mathbb{E}\left[\mathbf{1}_{\alpha > 0}\mathrm{erf}\left(\frac{\alpha \cdot c}{\sqrt{2}}\right)\alpha^2\right] = \frac{1}{\sqrt{2\pi}}\int_0^{+\infty}\mathrm{erf}\left(\frac{\alpha \cdot c}{\sqrt{2}}\right)\alpha^2\exp\left(-\frac{\alpha^2}{2}\right)\,d\alpha \tag{13}
$$

$$
= -\frac{1}{\sqrt{2\pi}}\int_0^{+\infty}\mathrm{erf}\left(\frac{\alpha \cdot c}{\sqrt{2}}\right)\alpha\,d\exp\left(-\frac{\alpha^2}{2}\right) \tag{14}
$$

$$
= \frac{1}{\sqrt{2\pi}}\int_0^{+\infty}\left(\mathrm{erf}\left(\frac{\alpha \cdot c}{\sqrt{2}}\right) + \frac{2}{\sqrt{\pi}}\alpha\exp\left(-\frac{\alpha^2 \cdot c^2}{\sqrt{2}}\right)\right)\exp\left(-\frac{\alpha^2}{2}\right)\,d\alpha \tag{15}
$$

$$
= \mathbb{E}\left[\mathbf{1}_{\alpha > 0}\mathrm{erf}\left(\frac{\alpha \cdot c}{\sqrt{2}}\right)\right] + \frac{2}{\sqrt{2\pi}}\mathbb{E}\left[\mathbf{1}_{\alpha > 0}\alpha\exp\left(-\frac{\alpha^2 \cdot c^2}{\sqrt{2}}\right)\right], \tag{16}
$$

with (14) resulting from $\alpha\exp\left(-\frac{\alpha^2}{2}\right)\,d\alpha = -d\exp\left(-\frac{\alpha^2}{2}\right)$, and (15) resulting from integration by parts.

Similarly, for the coefficient of the second term we can write:

$$
= \mathbb{E}\left[\mathbf{1}_{\alpha > 0}\mathbf{1}_{\alpha \cdot c + \beta > 0}\beta^2\right] - \mathbb{E}[\mathbf{1}_{\alpha > 0}\mathbf{1}_{\alpha \cdot c + \beta > 0}] \tag{17}
$$

$$
= \frac{1}{\sqrt{2\pi}}\mathbb{E}\left[\mathbf{1}_{\alpha > 0}\alpha \cdot c \cdot \exp\left(-\frac{\alpha^2 \cdot c^2}{2}\right)\right] + \mathbb{E}[\mathbf{1}_{\alpha > 0}\mathbf{1}_{\alpha \cdot c + \beta > 0}] - \mathbb{E}[\mathbf{1}_{\alpha > 0}\mathbf{1}_{\alpha \cdot c + \beta > 0}] \tag{18}
$$

$$
= \frac{1}{\sqrt{2\pi}}\mathbb{E}\left[\mathbf{1}_{\alpha > 0}\alpha \cdot c \cdot \exp\left(-\frac{\alpha^2 \cdot c^2}{2}\right)\right], \tag{19}
$$

where (18) comes from:

$$
\mathbb{E}_\beta\left[\mathbf{1}_{\alpha \cdot c + \beta > 0}\beta^2\right] = \frac{1}{\sqrt{2\pi}}\int_{-\alpha \cdot c}^{+\infty}\beta^2\exp\left(-\frac{\beta^2}{2}\right)\,d\beta \tag{20}
$$

$$
= -\frac{1}{\sqrt{2\pi}}\int_{-\alpha \cdot c}^{+\infty}\beta\,d\exp\left(-\frac{\beta^2}{2}\right) \tag{21}
$$

$$
= -\frac{1}{\sqrt{2\pi}}\left(-\alpha \cdot c \cdot \exp\left(-\frac{\beta^2 \cdot c^2}{2}\right) - \int_{-\alpha \cdot c}^{+\infty}\exp\left(-\frac{\beta^2}{2}\right)\,d\beta\right) \tag{22}
$$

$$
= \frac{1}{\sqrt{2\pi}}\alpha \cdot c \cdot \exp\left(-\frac{\beta^2 \cdot c^2}{2}\right) + \mathbb{E}_\beta[\mathbf{1}_{\alpha \cdot c + \beta > 0}]. \tag{23}
$$

And for the third term, we have:

$$\mathbb{E}[\mathbf{1}_{\alpha>0}\mathbf{1}_{\alpha\cdot c+\beta>0}\alpha\beta] = \frac{1}{\sqrt{2\pi}}\mathbb{E}\left[\mathbf{1}_{\alpha>0}\alpha\cdot\exp\left(-\frac{\alpha^2\cdot c^2}{2}\right)\right],$$

which comes from:

$$\begin{aligned}
\mathbb{E}_\beta[\mathbf{1}_{\alpha\cdot c+\beta>0}\beta] &= \frac{1}{\sqrt{2\pi}}\int_{-\alpha\cdot c}^{+\infty}\beta\cdot\exp\left(-\frac{\beta^2}{2}\right)\,\mathrm{d}\beta \\
&= -\frac{1}{\sqrt{2\pi}}\int_{-\alpha\cdot c}^{+\infty}\mathrm{d}\exp\left(-\frac{\beta^2}{2}\right) \\
&= \frac{1}{\sqrt{2\pi}}\exp\left(-\frac{\alpha^2\cdot c^2}{2}\right).
\end{aligned}$$

For (12), we have:

$$\frac{1}{\pi}\mathbb{E}\left[\mathbf{1}_{\alpha>0}\alpha\exp\left(-\frac{1}{\sqrt{2}}\alpha^2\cdot c^2\right)\right] \leq \frac{1}{\pi}\mathbb{E}[\mathbf{1}_{\alpha>0}\alpha]$$
$$= \frac{1}{\pi\sqrt{2\pi}},$$

which comes from $\exp\left(-\alpha^2\cdot c^2\right)\leq 1$. For (19), note that the maximum value of $c\cdot\exp\left(-\frac{\alpha^2\cdot c^2}{2}\right)$ happens at $\alpha^2\cdot c^2=1$, which gives us:

$$\frac{1}{\sqrt{2\pi}}\mathbb{E}\left[\mathbf{1}_{\alpha>0}\alpha\cdot c\cdot\exp\left(-\frac{\alpha^2\cdot c^2}{2}\right)\right] \leq \frac{1}{\sqrt{2\pi}}\mathbb{E}\left[\mathbf{1}_{\alpha>0}\exp\left(-\frac{1}{2}\right)\right]$$
$$= \frac{\exp\left(-\frac{1}{2}\right)}{2\sqrt{2\pi}}.$$

And for (A.11.1), we have:

$$\frac{1}{\sqrt{2\pi}}\mathbb{E}\left[\mathbf{1}_{\alpha>0}\alpha\cdot\exp\left(-\frac{\alpha^2\cdot c^2}{2}\right)\right] \leq \frac{1}{\sqrt{2\pi}}\mathbb{E}[\mathbf{1}_{\alpha>0}\alpha]$$
$$= \frac{1}{2\pi},$$

which also comes from $\exp\left(-\frac{\alpha^2\cdot c^2}{2}\right)\leq 1$.

So we can see that the coefficients of the first three terms are bounded. Furthermore, we can assume $\mathbf{a}\neq c'\cdot\mathbf{b}$ for some constant $c'\in\{+1,-1\}$, since otherwise the result would be trivial:

$$\begin{aligned}
\mathbb{E}_\delta\left[\mathbf{1}_{\delta^\top\mathbf{a}>0}\mathbf{1}_{\delta^\top\mathbf{b}>0}\delta\delta^\top\right] &= \mathbb{E}_\delta\left[\mathbf{1}_{\delta^\top\mathbf{a}>0}\delta\delta^\top\right] \\
&= \frac{1}{2}\mathbb{E}_\delta\left[\delta\delta^\top\right] \\
&= \frac{1}{2}\mathbf{I}_D,
\end{aligned}$$

for $c'=1$, and:

$$\mathbb{E}_\delta\left[\mathbf{1}_{\delta^\top\mathbf{a}>0}\mathbf{1}_{\delta^\top\mathbf{b}>0}\delta\delta^\top\right] = 0,$$

otherwise. So as $D\to\infty$, given that $\mathbf{a},\mathbf{b}$ are non-sparse and unit vectors, their magnitude will be stretched over the elements, resulting in each element of $\mathbf{a}$ and $\mathbf{b}$ approaching $0$ at the rate of $\sqrt{D}$. Therefore, the last term will be the dominant term, which completes our proof. $\qquad\square$

**Lemma A.6.** *Let $\delta\sim\mathcal{N}(0,\mathbf{I})$ be a standard Gaussian variable, and $\mathbf{a},\mathbf{b}\in\mathbb{R}^D$ be two vectors. Then, assuming $a\sim\mathcal{N}(0,\mathbf{I}_D)$, the term $\mathbb{E}_\delta[\mathbf{1}_{\delta^\top\mathbf{a}>0}\mathbf{1}_{\delta^\top\mathbf{b}>0}]$ will be a uniform random variable w.r.t. $\mathbf{a}$, and we have:*

$$\mathbb{E}_{\delta,\mathbf{a}}[\mathbf{1}_{\delta^\top\mathbf{a}>0}\mathbf{1}_{\delta^\top\mathbf{b}>0}] = \frac{1}{2}, \quad \mathbb{E}_\mathbf{a}\left[\mathbb{E}_\delta[\mathbf{1}_{\delta^\top\mathbf{a}>0}\mathbf{1}_{\delta^\top\mathbf{b}>0}]^2\right] = \frac{1}{3}. \tag{24}$$

**Proof.** Note that since $\mathbf{a}$ is a standard Gaussian variable, its angle with a given vector $\mathbf{b}$ is a uniformly distributed variable. Furthermore, $\mathbb{E}_\delta[\mathbf{1}_{\delta^\top \mathbf{a}>0}\mathbf{1}_{\delta^\top \mathbf{b}>0}]$ corresponds to the overlap of the two halfspaces defined by $\delta^\top \mathbf{a}>0$ and $\delta^\top \mathbf{b}>0$, which is equal to the ratio of the angle between $\mathbf{a}$ and $\mathbf{b}$ in Radian, divided by $2\pi$. As a result, it is a uniform variable $\sim \mathrm{U}[0,1]$ (i.e., uniformly distributed in $[0,1]$ range). So its expected value and second moment are, respectively, $\frac{1}{2}$ and $\frac{1}{3}$. $\qquad\square$

**Lemma A.7.** *Let $\delta \sim \mathcal{N}(0, \mathbf{I}_D)$ be a standard Gaussian random variable of size $D$. Then we have:*

$$\mathbb{E}_\delta\left[\frac{\delta\delta^\top}{\|\delta\|_2^2}\right] = \frac{1}{D}\mathbf{I}_D. \tag{25}$$

**Proof.** First, note that due to symmetry, the diagonal elements and the off-diagonal elements are equal in (25). Therefore, we can get the diagonal elements by evaluating the trace:

$$\mathbf{Tr}\left(\mathbb{E}_\delta\left[\frac{\delta\delta^\top}{\|\delta\|_2^2}\right]\right) = \mathbb{E}_\delta\left[\frac{\mathbf{Tr}(\delta\delta^\top)}{\|\delta\|_2^2}\right]$$
$$= \mathbb{E}_\delta[1]$$
$$= 1.$$

This gives us diagonal elements equal to $1/D$. On the other hand, note that we can write an off-diagonal element in (25) as:

$$\mathbb{E}_{\delta_i,\delta_j}\left[\frac{\delta_i\delta_j}{\delta_i^2+\delta_j^2+D-2}\right],$$

which is equal to $0$ due to the fact that $\delta$ is zero-mean and symmetric in distribution. This completes the proof. $\qquad\square$

### A.11.2 PROOF FOR THEOREM 3.1

Now that we have proved the two lemmas, we can prove Theorem 3.1.

**Proof.** Let $f_\theta(\mathbf{x})$ be the model described in the theorem:

$$f_\theta(\mathbf{x}) = \sum_{i=1}^n \omega_i \cdot \mathbf{1}_{\phi_i^\top \mathbf{x}>0}\phi_i^\top x,$$

where $\omega_i$ and $\phi_i$ are i.i.d. random Gaussian variables with zero mean and variance $\sigma_\omega^2$ and $\sigma_\phi^2$, respectively. As such, we have $\theta^\top = [\omega_i; \phi_i^\top]_{i=1}^n$, i.e., the concatenation of $\omega_i$s and $\phi_i$s. The derivatives of $f_\theta(\mathbf{x})$ can be written as:

$$\nabla_\mathbf{x} f_\theta(\mathbf{x}) = \sum_{i=1}^n \omega_i \cdot \mathbf{1}_{\phi_i^\top \mathbf{x}}\phi_i,$$

$$\frac{\partial f_\theta}{\partial \omega_i}(\mathbf{x}) = \mathbf{1}_{\phi_i^\top \mathbf{x}>0}\phi_i^\top \mathbf{x},$$

$$\frac{\partial f_\theta}{\partial \phi_i}(\mathbf{x}) = \omega_i \cdot \mathbf{1}_{\phi_i^\top \mathbf{x}>0}\mathbf{x},$$

$$\frac{\partial \nabla_\mathbf{x} f_\theta}{\partial \omega_i}(\mathbf{x}) = \mathbf{1}_{\phi_i^\top \mathbf{x}>0}\phi_i,$$

$$\frac{\partial \nabla_\mathbf{x} f_\theta(\mathbf{x})}{\partial \phi_i}(\mathbf{x}) = \omega_i \cdot \mathbf{1}_{\phi_i^\top \mathbf{x}>0}\mathbf{I}.$$

Following Proposition A.1, we can write $\Delta_\mathcal{F} = \mathbf{A}_\mathcal{F} + \mathbf{A}_\mathcal{F}^\top$ from (3) with:

$$\mathbf{A}_\mathcal{F} = -\sum_{\mu=1}^m \mathbb{E}_{\mathbf{x},\theta}\left[\left(\sum_{i=1}^n \mathbf{1}_{\phi_i^\top \mathbf{x}>0}\mathbf{1}_{\phi_i^\top \mathbf{x}_\mu>0}\left(\left(\phi_i^\top \mathbf{x}_\mu\right)\phi_i + \omega_i^2\mathbf{x}_\mu\right)\right)\right.$$
$$\left.\left(\sum_{j=1}^n \omega_j\mathbf{1}_{\phi_j^\top \mathbf{x}>0}\phi_j^\top\right)\left(\sum_{k=1}^n \omega_k\mathbf{1}_{\phi_k^\top \mathbf{x}_\mu>0}\phi_k^\top \mathbf{x}_\mu - y\right)\right]. \tag{26}$$

Note that since $\omega$ is zero-mean, the terms related to $y$ are eliminated from (26). Similarly, the terms involving $j \neq k$ are also eliminated. So we have:

$$\mathbf{A}_{\mathcal{F}} = -\sum_{\mu=1}^{m}\sum_{i=1}^{n}\sum_{j=1}^{n}\sigma_\omega^2\Big[\mathbb{E}_{\theta,\mathbf{x}}\Big[\mathbf{1}_{\phi_i^\top\mathbf{x}>0}\mathbf{1}_{\phi_i^\top\mathbf{x}_\mu>0}\mathbf{1}_{\phi_j^\top\mathbf{x}>0}\mathbf{1}_{\phi_j^\top\mathbf{x}_\mu>0}\big(\phi_i^\top\mathbf{x}_\mu\big)\big(\phi_j^\top\mathbf{x}_\mu\big)\phi_i\phi_j^\top\Big] \quad (27)$$
$$+ \sigma_\omega^4\mathbb{E}_{\theta,\mathbf{x}}\Big[\mathbf{1}_{\phi_i^\top\mathbf{x}>0}\mathbf{1}_{\phi_i^\top\mathbf{x}_\mu>0}\mathbf{1}_{\phi_j^\top\mathbf{x}>0}\mathbf{1}_{\phi_j^\top\mathbf{x}_\mu>0}\big(\phi_j^\top\mathbf{x}_\mu\big)\mathbf{x}_\mu\phi_j^\top\Big]\Big].$$

For the first term in (27), we have $n$ terms with $i = j$, which are equal to:

$$\mathbb{E}_{\theta,\mathbf{x}}\Big[\mathbf{1}_{\phi_i^\top\mathbf{x}>0}\mathbf{1}_{\phi_i^\top\mathbf{x}_\mu>0}\big(\phi_i^\top\mathbf{x}_\mu\big)^2\phi_i\phi_i^\top\Big] = \mathbb{E}_\theta\Big[\mathbb{E}_\mathbf{x}\Big[\mathbf{1}_{\phi_i^\top\mathbf{x}>0}\Big]\mathbf{1}_{\phi_i^\top\mathbf{x}_\mu>0}\big(\phi_i^\top\mathbf{x}_\mu\big)^2\phi_i\phi_i^\top\Big]$$
$$= \frac{1}{2}\mathbb{E}_\theta\Big[\mathbf{1}_{\phi_i^\top\mathbf{x}_\mu>0}\big(\phi_i^\top\mathbf{x}_\mu\big)^2\phi_i\phi_i^\top\Big]$$
$$= \frac{1}{4}\mathbb{E}_\theta\Big[\big(\phi_i^\top\mathbf{x}_\mu\big)^2\phi_i\phi_i^\top\Big]$$
$$= \frac{\sigma_\phi^4}{4}\mathbf{x}_\mu\mathbf{x}_\mu^\top + \frac{\sigma_\phi^4}{4}\mathbf{x}_\mu^\top\mathbf{x}_\mu\mathbf{I},$$

And in the case when $i \neq j$, we have $n^2 - n$ terms each of which are equal to:

$$\mathbb{E}_\mathbf{x}\Big[\mathbb{E}_{\phi_i}\Big[\mathbf{1}_{\phi_i^\top\mathbf{x}>0}\mathbf{1}_{\phi_i^\top\mathbf{x}_\mu>0}\phi_i\phi_i^\top\Big]\mathbf{x}_\mu\mathbf{x}_\mu^\top\mathbb{E}_{\phi_j}\Big[\mathbf{1}_{\phi_j^\top\mathbf{x}>0}\mathbf{1}_{\phi_j^\top\mathbf{x}_\mu>0}\phi_j\phi_j^\top\Big]\Big] \quad (28)$$
$$\approx \sigma_\phi^4\mathbb{E}_\mathbf{x}\Big[\mathbb{E}_{\phi_i}\Big[\mathbf{1}_{\phi_i^\top\mathbf{x}>0}\mathbf{1}_{\phi_i^\top\mathbf{x}_\mu>0}\Big]\cdot\mathbb{E}_{\phi_j}\Big[\mathbf{1}_{\phi_j^\top\mathbf{x}>0}\mathbf{1}_{\phi_j^\top\mathbf{x}_\mu>0}\Big]\Big]\mathbf{x}_\mu\mathbf{x}_\mu^\top \quad (29)$$
$$= \frac{\sigma_\phi^4}{3}\mathbf{x}_\mu\mathbf{x}_\mu^\top, \quad (30)$$

where (29) comes from Lemma A.5, and (30) comes from Lemma A.6. For the second term in (27), when $i = j$ we have $n$ terms in the following form:

$$\mathbb{E}_{\theta,\mathbf{x}}\Big[\mathbf{1}_{\phi_i^\top\mathbf{x}>0}\mathbf{1}_{\phi_i^\top\mathbf{x}_\mu>0}\big(\phi_i^\top\mathbf{x}_\mu\big)\mathbf{x}_\mu\phi_i^\top\Big] = \mathbf{x}_\mu\mathbf{x}_\mu^\top\mathbb{E}_{\theta,\mathbf{x}}\Big[\mathbf{1}_{\phi_i^\top\mathbf{x}>0}\mathbf{1}_{\phi_i^\top\mathbf{x}_\mu}\phi_i\phi_i^\top\Big] \quad (31)$$
$$= \sigma_\phi^2\cdot\mathbf{x}_\mu\mathbf{x}_\mu^\top\mathbb{E}_{\mathbf{x},\phi_i}\Big[\mathbf{1}_{\phi_i^\top\mathbf{x}>0}\mathbf{1}_{\phi_i^\top\mathbf{x}_\mu>0}\Big] \quad (32)$$
$$= \frac{\sigma_\phi^2}{2}\mathbf{x}_\mu\mathbf{x}_\mu^\top, \quad (33)$$

where (32) follows from Lemma A.5 and (33) follows from Lemma A.6. And finally, the second term in (27) has $n^2 - n$ terms with $i \neq j$, which are equal to:

$$\mathbb{E}_\mathbf{x}\Big[\mathbb{E}_{\phi_i}\Big[\mathbf{1}_{\phi_i^\top\mathbf{x}>0}\mathbf{1}_{\phi_i^\top\mathbf{x}_\mu>0}\Big]\mathbb{E}_{\phi_j}\Big[\mathbf{1}_{\phi_j^\top\mathbf{x}>0}\mathbf{1}_{\phi_j^\top\mathbf{x}_\mu>0}\big(\phi_j^\top\mathbf{x}_\mu\big)\mathbf{x}_\mu\phi_j^\top\Big]\Big] \quad (34)$$
$$= \sigma_\phi^2\mathbb{E}_\mathbf{x}\Big[\mathbb{E}_{\phi_i}\Big[\mathbf{1}_{\phi_i^\top\mathbf{x}>0}\mathbf{1}_{\phi_i^\top\mathbf{x}_\mu>0}\Big]\mathbb{E}_{\phi_j}\Big[\mathbf{1}_{\phi_j^\top\mathbf{x}>0}\mathbf{1}_{\phi_j^\top\mathbf{x}_\mu>0}\Big]\Big]\mathbf{x}_\mu\mathbf{x}_\mu^\top \quad (35)$$
$$= \frac{\sigma_\phi^2}{3}\mathbf{x}_\mu\mathbf{x}_\mu^\top, \quad (36)$$

where (35) comes from Lemma A.5 and (36) comes from Lemma A.6.

Finally, given the results so far, we can write:

$$\Delta_{\mathcal{F}} = -2\cdot\sum_{\mu=1}^{m}\Big(\frac{n\sigma_\phi^4\sigma_\omega^2}{4} + \big(n^2-n\big)\frac{\sigma_\phi^4\sigma_\omega^2}{3} - \frac{n\sigma_\phi^2\sigma_\omega^4}{2} - \big(n^2-n\big)\frac{\sigma_\phi^2\sigma_\omega^4}{3}\Big)\mathbf{x}_\mu\mathbf{x}_\mu^\top - 2\cdot\frac{n\sigma_\phi^4\sigma_\omega^2}{4}\|\mathbf{x}_\mu\|_2^2\mathbf{I} - 2\cdot\mathcal{O}\big(\frac{1}{\sqrt{D}}\big)\mathcal{E}_1$$

$$= -2n\Big(\frac{\sigma_\phi^4\sigma_\omega^2}{4} + (n-1)\frac{\sigma_\phi^4\sigma_\omega^2}{3} + \frac{\sigma_\phi^2\sigma_\omega^4}{2} + (n-1)\frac{\sigma_\phi^2\sigma_\omega^4}{3}\Big)\sum_{\mu=1}^{m}\mathbf{x}_\mu\mathbf{x}_\mu^\top - \frac{2n\sigma_\phi^4\sigma_\omega^2}{4}\sum_{\mu=1}^{m}\|\mathbf{x}_\mu\|_2^2\mathbf{I} - \mathcal{O}\big(\frac{2}{\sqrt{D}}\big)\mathcal{E}_1$$

$$= -\mathcal{O}(n^2)\cdot\mathbf{S} - \mathcal{O}\big(\frac{1}{\sqrt{D}}\big)\cdot\mathcal{E}_1 - \mathcal{O}(n)\cdot\mathcal{E}_2,$$

where $\mathcal{E}_1$ is the error corresponding to our approximation from Lemma A.5, and $\mathcal{E}_2 = \sum_{x_\mu=1}^{m}\|x_\mu\|_2^2\mathbf{I}$.

So we have $\displaystyle\lim_{n,D\to\infty}\left|\frac{\mathbf{Tr}(\Delta_{\mathcal{F}}^\top\mathbf{S})}{\|\Delta_{\mathcal{F}}\|_F\cdot\|\mathbf{S}\|_F}\right| = \lim_{n,D\to\infty}\left|\frac{\mathbf{Tr}\Big(\big(\mathcal{O}(n^2)\mathbf{S}+\mathcal{O}\big(\frac{1}{\sqrt{D}}\big)\mathcal{E}_1+\mathcal{O}(n)\mathcal{E}_2\big)^\top\mathbf{S}\Big)}{\left\|\big(\mathcal{O}(n^2)\mathbf{S}+\mathcal{O}\big(\frac{1}{\sqrt{D}}\big)\mathcal{E}_1+\mathcal{O}(n)\mathcal{E}_2\big)^\top\mathbf{S}\right\|_F\cdot\|\mathbf{S}\|_F}\right| = 1$ $\qquad\square$

### A.11.3 PROOF FOR THEOREM 3.2

**Proof.** Let $f_\theta(\mathbf{x})$ be the model described in the theorem:

$$f_\theta(\mathbf{x}) = \frac{1}{k}\sum_{i=1}^n \omega_i \left(\sum_{j=1}^k \phi_i^\top \mathbf{M}_j^\top \mathbf{x}\right),$$

where $\omega_i$ and $\phi_i$ are i.i.d. random Gaussian variables with zero mean and variances $\sigma_\omega^2$ and $\sigma_\phi^2$, respectively, and $\mathbf{M}_j$ is a binary matrix mimicking the convolution operation. Note that we set $\mathbf{I}_{j_1,j_2} = \mathbf{M}_{j_1}\mathbf{M}_{j_2}^T$, which is a binary diagonal matrix with elements in the receptive field of both $j_1$ and $j_2$ set to 1 and 0 otherwise. Furthermore, we have $\theta^\top = [\omega_i; \phi_i^\top]_{i=1}^n$, i.e., the concatenation of $\omega_i$s and $\phi_i$s. The derivatives of $f_\theta(\mathbf{x})$ can be written as:

$$\nabla_\mathbf{x} f_\theta(\mathbf{x}) = \frac{1}{k}\sum_{i=1}^n \omega_i \left(\sum_{j=1}^k \mathbf{M}_j \phi_i\right),$$

$$\frac{\partial f_\theta(\mathbf{x})}{\partial \omega_i}(\mathbf{x}) = \frac{1}{k}\sum_{j=1}^k \phi_i^\top \mathbf{M}_j^\top \mathbf{x},$$

$$\frac{\partial \nabla_\mathbf{x} f_\theta}{\partial \omega_i}(\mathbf{x}) = \frac{1}{k}\sum_{j=1}^k \mathbf{M}_j \phi_i,$$

$$\frac{\partial f_\theta}{\partial \phi_i}(\mathbf{x}) = \frac{1}{k}\omega_i \sum_{j=1}^k \mathbf{M}_j^\top \mathbf{x},$$

$$\frac{\partial \nabla_\mathbf{x} f_\theta}{\partial \phi_i}(\mathbf{x}) = \frac{1}{k}\omega_i \sum_{j=1}^k \mathbf{M}_j.$$

We first evaluate $\mathbf{G}_\mathcal{F}(\mathbf{x})$:

$$\mathbf{G}_\mathcal{F}(\mathbf{x}) = \frac{1}{k^2}\mathbb{E}_\theta\left[\left(\sum_{i_1=1}^n \omega_{i_1}\left(\sum_{j_1=1}^k \mathbf{M}_{j_1}\phi_{i_1}\right)\right)\left(\sum_{i_2=1}^n \omega_{i_2}\left(\sum_{j_2=1}^k \mathbf{M}_{j_2}\phi_{i_2}\right)\right)^\top\right]$$

$$= \frac{\sigma_\omega^2}{k^2}\sum_{i=1}^n\sum_{j_1=1}^k\sum_{j_2=1}^k \mathbb{E}_\theta\left[\mathbf{M}_{j_1}\phi_i\phi_i^T\mathbf{M}_{j_2}^\top\right]$$

$$= \frac{n\sigma_\omega^2\sigma_\phi^2}{k^2}\sum_{j_1=1}^k\sum_{j_2=1}^k \mathbf{I}_{j_1,j_2}.$$

Now we will try to show that $\Delta_\mathcal{F}(\mathbf{x}) = \mathbf{A}_\mathcal{F}(\mathbf{x}) + \mathbf{A}_\mathcal{F}(\mathbf{x})^\top = \mathbf{G}_\mathcal{F}(\mathbf{x})\mathbf{S}\mathbf{G}_\mathcal{F}(\mathbf{x})$. Following Proposition A.1, we can write $\Delta_\mathcal{F}$ as:

$$\mathbf{A}_\mathcal{F}(\mathbf{x}) = -\frac{1}{k^4}\sum_{\mu=1}^m\sum_{i_1=1}^n\sum_{i_2=1}^n\sum_{j_1=1}^k\sum_{j_2=1}^k\sum_{j_3=1}^k\sum_{j_4=1}^k \left(\sigma_\omega^2\mathbb{E}_\theta\left[\left(\phi_{i_1}^\top\mathbf{M}_{j_1}^\top\mathbf{x}_\mu\right)\left(\phi_{i_1}^\top\mathbf{M}_{j_4}^\top\mathbf{x}_\mu\right)\mathbf{M}_{j_2}\phi_{i_1}\phi_{i_2}^\top\mathbf{M}_{j_3}^\top\right]\right. \quad (37)$$
$$\left. + \sigma_\omega^4\mathbb{E}_\theta\left[\left(\phi_{i_2}^\top\mathbf{M}_{j_4}^\top\mathbf{x}_\mu\right)\mathbf{M}_{j_1}\mathbf{M}_{j_2}^\top\mathbf{x}_\mu\phi_{i_2}^\top\mathbf{M}_{j_3}^\top\right]\right).$$

The first term in (37) has $n$ terms with $i_1 = i_2$ (substitute $i_1, i_2$ with $i$), which can be written as:

$$n\mathbf{M}_{j_2}\mathbb{E}_{\phi_i}\left[\left(\phi_i^\top\mathbf{M}_{j_1}^\top\mathbf{x}_\mu\right)\phi_i\phi_i^\top\mathbf{x}_\mu\mathbf{x}_\mu^\top\mathbf{M}_{j_4}\phi_i\phi_i^\top\right] = n\mathbf{M}_{j_2}\mathbb{E}_{\phi_i}\left[\phi_i\phi_i^\top\mathbf{M}_{j_1}^\top\mathbf{x}_\mu\mathbf{x}_\mu^\top\mathbf{M}_{j_4}\phi_i\phi_i^\top\right]$$

So using Stein's lemma, we have:

$$n\mathbf{M}_{j_2}\mathbb{E}_{\phi_i}\left[\phi_i\phi_i^\top\mathbf{M}_{j_1}^\top\mathbf{x}_\mu\mathbf{x}_\mu^\top\mathbf{M}_{j_4}\phi_i\phi_i^\top\right] = n\sigma_\phi^2\mathbf{M}_{j_2}\left(\mathbb{E}_{\phi_i}\left[\phi_i^\top\mathbf{M}_{j_1}^\top\mathbf{x}_\mu\mathbf{x}_\mu^\top\mathbf{M}_{j_4}\phi_i\right] + \mathbf{M}_{j_1}^\top\mathbf{x}_\mu\mathbf{x}_\mu^\top\mathbf{M}_{j_4}\mathbb{E}_{\phi_i}\left[\phi_i\phi_i^\top\right]\right).$$

The first term is equal to $\sigma_\phi^2 \mathbf{Tr}\big(\mathbf{M}_{j_1}^\top x_\mu x_\mu^\top \mathbf{M}_{j_4}\big)$, and the second term is equal to $\sigma_\phi^2 \mathbf{M}_{j_1} \mathbf{x}_\mu \mathbf{x}_\mu^\top \mathbf{M}_{j_4}$. So the $n$ terms in (37) with $i_1 = i_2$ are equivalent to:

$$n\sigma_\phi^4 \mathbf{I}_{j_2,j_1} \mathbf{x}_\mu \mathbf{x}_\mu^\top \mathbf{I}_{j_3,j_4} + n\sigma_\phi^4 \mathbf{Tr}\big(\mathbf{M}_{j_1}^\top \mathbf{x}_\mu \mathbf{x}_\mu^\top \mathbf{M}_{j_4}\big)\mathbf{I}_{j_2,j_3}.$$

The first term in (37) also has $n^2 - n$ terms with $i_1 \neq i_2$, which can be written as:

$$\big(n^2 - n\big)\mathbf{M}_{j_2}\mathbb{E}_{\phi_{i_1}}\big[\phi_{i_1}\phi_{i_1}^\top\big]\mathbf{M}_{j_1}^\top \mathbf{x}_\mu \mathbf{x}_\mu^\top \mathbf{M}_{j_4}\mathbb{E}_{\phi_{i_2}}\big[\phi_{i_2}\phi_{i_2}^\top\big]\mathbf{M}_{j_3} = \big(n^2 - n\big)\sigma_\phi^4 \mathbf{I}_{j_2,j_1}\mathbf{x}_\mu \mathbf{x}_\mu^T \mathbf{I}_{j_3,j_4}.$$

The second term in (37) has $n^2$ which can be written as:

$$n^2\mathbf{I}_{j_1,j_2}\mathbf{x}_\mu \mathbf{x}_\mu^\top \mathbf{M}_{j_4}\mathbb{E}_{\phi_{i_2}}\big[\phi_{i_2}\phi_{i_2}^\top\big]\mathbf{M}_{j_3} = \sigma_\phi^2 n^2 \mathbf{I}_{j_1,j_2}\mathbf{x}_\mu \mathbf{x}_\mu^T \mathbf{I}_{j_3,j_4}.$$

So we can write:

$$\Delta_{\mathcal{F}}(\mathbf{x}) = -2\cdot\sum_{j_1=1}^{k}\sum_{j_2=1}^{k}\sum_{j_3=1}^{k}\sum_{j_4=1}^{k}\frac{n^2}{k^4}\big(\sigma_\omega^2\sigma_\phi^4 + \sigma_\omega^4\sigma_\phi^2\big)\mathbf{I}_{j_1,j_2}\mathbf{S}\mathbf{I}_{j_3,j_4} - 2\cdot\frac{n}{k^4}\sigma_\omega^2\sigma_\phi^4 \mathbf{Tr}\big(\mathbf{M}_{j_1}^\top \mathbf{S}\mathbf{M}_{j_4}\big)\mathbf{I}_{j_2,j_3}.$$

So we have:

$$\Delta_{\mathcal{F}}(\mathbf{x}) = -\mathcal{O}\big(n^2\big)\cdot\mathbf{G}_{\mathcal{F}}(\mathbf{x})\mathbf{S}\mathbf{G}_{\mathcal{F}}(\mathbf{x})^\top - \mathcal{O}(n)\cdot\mathcal{E},$$

where $\mathcal{E}$ corresponds to the error term $\sum_{j_1=1}^{k}\sum_{j_2=1}^{k}\sum_{j_3=1}^{k}\sum_{j_4=1}^{k}\mathbf{Tr}\,\big(\mathbf{M}_{j_1}^\top \mathbf{S}\mathbf{M}_{j_4}\big)\,\mathbf{I}_{j_2,j_3}$.

Therefore, we can write: $\displaystyle\lim_{n\to\infty}\left|\frac{\mathbf{Tr}\big(\Delta_{\mathcal{F}}(\mathbf{x})^\top \mathbf{G}_{\mathcal{F}}(\mathbf{x})\mathbf{S}\mathbf{G}_{\mathcal{F}}(\mathbf{x})\big)}{\|\Delta_{\mathcal{F}}(\mathbf{x})\|_F \cdot \|\mathbf{G}_{\mathcal{F}}(\mathbf{x})\mathbf{S}\mathbf{G}_{\mathcal{F}}(\mathbf{x})\|_F}\right| = $

$\displaystyle\lim_{n\to\infty}\left|\frac{\mathbf{Tr}\Big(\big(\mathcal{O}\big(n^2\big)\cdot\mathbf{G}_{\mathcal{F}}(\mathbf{x})\mathbf{S}\mathbf{G}_{\mathcal{F}}(\mathbf{x})^\top + \mathcal{O}(n)\cdot\mathcal{E}\big)^\top \mathbf{G}_{\mathcal{F}}(\mathbf{x})\mathbf{S}\mathbf{G}_{\mathcal{F}}(\mathbf{x})\Big)}{\|\big(\mathcal{O}\big(n^2\big)\cdot\mathbf{G}_{\mathcal{F}}(\mathbf{x})\mathbf{S}\mathbf{G}_{\mathcal{F}}(\mathbf{x})^\top + \mathcal{O}(n)\cdot\mathcal{E}\big)\|_F \cdot \|\mathbf{G}_{\mathcal{F}}(\mathbf{x})\mathbf{S}\mathbf{G}_{\mathcal{F}}(\mathbf{x})\|_F}\right| = 1$ which completes our proof. $\qquad\square$

### A.11.4 PROOF FOR THEOREM A.2

**Proof.** Let $f_\theta^\ell(\mathbf{x})$ be an MLP or CNN with $\ell$ layers, no pooling or normalization layers, and a scalar output:

$$f_\theta^\ell(\mathbf{x}) = \sum_{i=1}^{n}\omega_i \mathbf{1}_{f_{\phi_i}^{\ell-1}(\mathbf{x})>0}f_{\phi_i}^{\ell-1}(\mathbf{x}),$$

where $\theta$ corresponds to the union of $\omega_i$s and $\phi_i$s, and $f_{\phi_i}(\cdot)$ and $f_{\phi_j}(\cdot)$ for $i \neq j$ share the same parameters up to the last layer. The derivative of $f_\theta^\ell(\mathbf{x})$ w.r.t. $\mathbf{x}$ is:

$$\nabla_\mathbf{x}f_\theta^\ell(\mathbf{x}) = \sum_{i=1}^{n}\omega_i \mathbf{1}_{f_{\phi_i}^{\ell-1}(\mathbf{x})>0}\nabla_\mathbf{x}f_{\phi_i}^{\ell-1}(\mathbf{x}),$$

which means we have:

$$\mathbb{E}_\theta\big[\nabla_\mathbf{x}f_\theta^\ell(\mathbf{x})\nabla_\mathbf{x}f_\theta^\ell(\mathbf{x})^\top\big] = \sum_{i_1=1}^{n}\sum_{i_2=1}^{n}\mathbb{E}_\theta\bigg[\omega_i\omega_j \mathbf{1}_{f_{\phi_i}^{\ell-1}(\mathbf{x})>0}\mathbf{1}_{f_{\phi_j}^{\ell-1}(\mathbf{x})>0}\nabla_\mathbf{x}f_{\phi_{i_1}}^{\ell-1}(\mathbf{x})\nabla_x f_{\phi_{i_2}}^{\ell-1}(\mathbf{x})^\top\bigg]$$

$$= \sigma_\omega^2\sum_{i=1}^{n}\mathbb{E}_{\phi_i}\bigg[\mathbf{1}_{f_{\phi_i}^{\ell-1}(\mathbf{x})>0}\nabla_\mathbf{x}f_{\phi_i}^{\ell-1}(\mathbf{x})\nabla_\mathbf{x}f_{\phi_i}^{\ell-1}(\mathbf{x})^\top\bigg].$$

So we need to prove that:

$$\mathbb{E}_{\phi_i}\bigg[\mathbf{1}_{f_{\phi_i}^{\ell-1}(\mathbf{x})>0}\nabla_\mathbf{x}f_{\phi_i}^{\ell-1}(\mathbf{x})\nabla_x f_{\phi_i}^{\ell-1}(\mathbf{x})^\top\bigg] = c\cdot\mathbf{I}.$$

First note that since the weights of the last layer (i.e., the classification layer) are zero-mean and Gaussian, the model w.r.t. these weights is symmetric at 0, which means we have:

$$\mathbb{E}_{\phi_i}\bigg[\mathbf{1}_{f_{\phi_i}^{\ell-1}(\mathbf{x})>0}\nabla_\mathbf{x}f_{\phi_i}^{\ell-1}(\mathbf{x})\nabla_\mathbf{x}f_{\phi_i}^{\ell-1}(\mathbf{x})^\top\bigg] = \frac{1}{2}\mathbb{E}_{\phi_i}\bigg[\nabla_\mathbf{x}f_{\phi_i}^{\ell-1}(\mathbf{x})\nabla_\mathbf{x}f_{\phi_i}^{\ell-1}(\mathbf{x})^\top\bigg]$$

So in order to prove this theorem, we need to use induction. For the case with $\ell = 2$, we can write the model as:

$$f_\theta^2(\mathbf{x}) = \sum_{i=1}^{n}\omega_i \mathbf{1}_{\phi_i^\top \mathbf{x}>0}\phi_i^\top \mathbf{x},$$

where both $\omega_i$s and $\phi_i$s are i.i.d. Gaussian variables with variances equal to $\sigma_\omega^2$ and $\sigma_\phi^2$, respectively. This gives us the following gradient w.r.t. the input:

$$\nabla_x f_\theta^2(\mathbf{x}) = \sum_{i=1}^{n} \omega_i \mathbf{1}_{\phi_i^\top \mathbf{x} > 0} \phi_i.$$

So we have:

$$\mathbb{E}_\theta \left[ \nabla_\mathbf{x} f_\theta^2(\mathbf{x}) \nabla_\mathbf{x} f_\theta^2(\mathbf{x})^\top \right] = \sigma_\omega^2 \sum_{i=1}^{n} \mathbb{E}_\theta \left[ \mathbf{1}_{\phi_i^\top \mathbf{x} > 0} \phi_i \phi_i^\top \right] \tag{38}$$

$$= \frac{1}{2} \sum_{i=1}^{n} \mathbb{E}_{\phi_i} \left[ \phi_i \phi_i^\top \right] \tag{39}$$

$$= \frac{n \sigma_\phi^2}{2} \mathbf{I}, \tag{40}$$

where (39) follows from $\phi_i$s being zero-mean i.i.d. Gaussian variables. The rest of the proof follows from induction. $\square$

### A.11.5 Proof for Theorem A.3

**Proof.** Let $f_\theta(\mathbf{x})$ be the model described:

$$f_\theta(\mathbf{x}) = \frac{1}{k} \sum_{j=1}^{k} \theta^\top \mathbf{M}_j^\top \mathbf{x},$$

where $\theta$ is a i.i.d. random Gaussian variable with zero mean and variance $\sigma_\theta^2$, and $\mathbf{M}_j$ is a binary matrix mimicking the convolution operation. The derivative of $f_\theta(x)$ w.r.t. $x$ is:

$$\nabla_\mathbf{x} f_\theta(\mathbf{x}) = \frac{1}{k} \sum_{j=1}^{k} \frac{1}{\sigma_j(\theta)} \mathbf{M}_j \theta.$$

So we can write:

$$\mathbf{G}_\mathcal{F} = \frac{1}{k^2} \sum_{j_1=1}^{k} \sum_{j_2=1}^{k} \mathbf{M}_{j_1} \mathbb{E}_\theta \left[ \theta \theta^\top \right] \mathbf{M}_{j_2}^\top$$

$$= \frac{\sigma_\theta^2}{k^2} \sum_{j_1=1}^{k} \sum_{j_2=1}^{k} \mathbf{M}_{j_1} \mathbf{M}_{j_2}^\top$$

$$= \frac{\sigma_\theta^2}{k^2} \sum_{j_1=1}^{k} \sum_{j_2=1}^{k} \mathbf{I}_{j_1, j_2},$$

which completes the proof. $\square$

### A.11.6 Proof for Corollary A.4

**Proof.** Let $f_\theta(\mathbf{x})$ be the model described:

$$f_\theta(\mathbf{x}) = \frac{1}{k} \sum_{j=1}^{k} \frac{1}{\sigma_j(\theta)} \theta^\top \mathbf{M}_j^\top \mathbf{x},$$

where $\theta$ is a i.i.d. random Gaussian variable with zero mean and variance $\sigma_\theta^2$, and $\mathbf{M}_j$ is a binary matrix mimicking the convolution operation. The derivative of $f_\theta(x)$ w.r.t. $x$ is:

$$\nabla_\mathbf{x} f_\theta(\mathbf{x}) = \frac{1}{k} \sum_{j=1}^{k} \frac{1}{\sigma_j(\theta)} \mathbf{M}_j \theta.$$

So we can write:

$$\mathbf{G}_{\mathcal{F}} = \frac{1}{k^2} \sum_{j_1=1}^{k} \sum_{j_2=1}^{k} \mathbf{M}_{j_1} \mathbb{E}_{\theta} \left[ \frac{\theta \theta^{\top}}{\sigma_{j_1}(\theta) \sigma_{j_2}(\theta)} \right] \mathbf{M}_{j_2}^{\top}.$$

Note that we can write $\sigma_j(\theta)$ for some $j$ as:

$$\sigma_j(\theta) = \frac{1}{D} \theta^{\top} \mathbf{M}_j^{\top} \mathbf{S} \mathbf{M}_j \theta$$
$$= \frac{1}{D} \theta^{\top} \mathbf{S}_j \theta,$$

which can be bounded as:

$$\lambda_{min}^j \|\theta\|_2^2 \leq D \cdot \sigma_j(\theta) \leq \lambda_{max}^j \|\theta\|_2^2.$$

This gives us the following bounds:

$$\frac{1}{\sqrt{\lambda_{min}^{j_1} \lambda_{min}^{j_2}}} \mathbb{E}_{\theta} \left[ \frac{\theta \theta^{\top}}{\|\theta\|_2^2} \right] \succeq \mathbb{E}_{\theta} \left[ \frac{\theta \theta^{\top}}{\sigma_{j_1}(\theta) \sigma_{j_2}(\theta)} \right] \succeq \frac{1}{\sqrt{\lambda_{max}^{j_1} \lambda_{max}^{j_2}}} \mathbb{E}_{\theta} \left[ \frac{\theta \theta^{\top}}{\|\theta\|_2^2} \right].$$

And from Lemma A.7, we have:

$$\frac{1}{\sqrt{\lambda_{min}^{j_1} \lambda_{min}^{j_2}}} \mathbf{I} \succeq \mathbb{E}_{\theta} \left[ \frac{\theta \theta^{\top}}{\sigma_{j_1}(\theta) \sigma_{j_2}(\theta)} \right] \succeq \frac{1}{\sqrt{\lambda_{max}^{j_1} \lambda_{max}^{j_2}}} \mathbf{I},$$

which completes our proof. □

## A.12 Experimental setting

### A.12.1 Hyperparameters

All of our experiments are performed on the following models:

- **MLP:** A multi-layer perception with 2 hidden layers of size 100 and ReLU non-linearity.
- **LeNet:** We use the model introduced in (LeCun et al., 1998).
- **ResNet18:** We use the model introduced in (He et al., 2016).
- **VGG11:** We use the model introduced in (Simonyan & Zisserman, 2015).
- **ViT.** We use the architecture introduced in (Dosovitskiy et al., 2021) with the settings provided by (Lee et al., 2021) for small-scale data.

Similar to (Ortiz-Jiménez et al., 2021), we observed much better and accurate value for $\mathbf{G}_{\mathcal{F}}$ when using GELU non-linearity (Hendrycks & Gimpel, 2016) compared to ReLU, which we also attribute to numerical problems caused by ReLU. In this case, we observed lower variance when computing $\mathbf{G}_{\mathcal{F}}$ with a probing function with a smaller standard deviation, which we attribute to the saturation of the GELU function for standard Gaussian input. Specifically, in our experiments we set $\mathcal{P} = \mathcal{N}\left(0, \sigma_{\mathcal{P}}^2 \mathbf{I}\right)$ with $\sigma_{\mathcal{P}}$ working best in range of $10^{-3}$ and $10^{-5}$ in terms of correlation between generalization gap and eigenvalues of $\mathbf{G}_{\mathcal{F}}$ in the experiment presented in App. A.7. However, similar to (Ortiz-Jiménez et al., 2021), we observe that a first-order approximation of $\mathbf{G}_{\mathcal{F}}$ in the form of $\mathbf{G}_{\mathcal{F}}(0)$ is also a relatively good approximation.

The experiments provided in Figure 2 are performed using gradient-descent with the learning rate equal to 0.1 and momentum set to 0.9, and for 100 epochs. The experiments provided in Figure 8 are performed using the Adam variant of gradient descent with default parameters due to slower convergence and for 100 epochs. The weight decay in all these experiments is set to 0.

For the experiments in Figure 4 and Figure 5, we use the experiment settings of (Ortiz-Jiménez et al., 2020) for the synthetic data, except for the experiments in Figure 4 performed on a sinusoidal decision boundary, for which we train all models with 50 epochs.

The experiments in Figure 3 are performed using stochastic gradient-descent with a batch size of 128 and learning rate of 0.1. We train the model for 100 epochs for the experiment involving LeNet and 50 epochs for the experiments involving ResNet18 and ViT. We set the momentum to 0.9 and the weight decay to 0.

The experiments in Figure 6 are performed using stochastic gradient-descent with a batch size of $128$, learning rate of $0.1$, and momentum set to $0.9$ and $50$ epochs. The weight decay in all these experiments is set to $0$. The experiments in Table 1 and Table 2 are performed using stochastic gradient-descent with a batch size of $128$, learning rate of $0.2$, and momentum set to $0.9$ and $50$ epochs. The weight decay in all these experiments is set to $5 \times 10^{-4}$.

In all experiments, we estimate $\mathbf{G}_{\mathcal{F}}$ on $10000$ models and $\mathbf{G}_{\mathcal{F}}^{t}$ for $t > 0$ on $25$ models.

### A.12.2 STATISTICAL INFORMATION

For the experiments in Figure 2, Figure 8, and Figure 3 we only report the average of test/train accuracy since we mainly aim to show the trend of the changes in these values. However, the experiments in Figure 4, Figure 5, Figure 6, Table 1, and Table 2 are all performed $5$ times. We report the average along with the $68\%$ confidence interval for all these experiments.

### A.12.3 HARDWARE AND FRAMEWORK

All experiments are performed on a single Nvidia GTX 1080 Ti, and implemented on Python using PyTorch 2.1.1. We base our implementation on the code provided by (Ortiz-Jiménez et al., 2020).

### A.13 LIMITATIONS

We consider two main limitations to our work: theoretical limitations and experimental limitations. In the case of the theoretical limitation, we note that our theoretical results are mainly focused on simple models (i.e., models with one or two hidden layers, and ReLU or linear activation functions) and based on approximation (i.e., correct for large input space size and model width). The reason for this approach is that in theoretical results, we're mainly concerned with insight as opposed to completeness. Consequently, we omit providing proofs with minimum assumptions and more realistic models in order to keep the message of the paper focused and the content concise.

In the case of experimental limitation, we note that our experiments are mainly concerned with over-parameterized settings and small-scale image classification datasets (i.e., models with a few million parameters and datasets with about 5000 samples for each class). This decision is mainly due to our lack of computational resources. As a result, we do not make any claims about more complex settings involving, for instance, foundation models or large-scale datasets. Furthermore, we do not extend our results to other aspects of machine learning, such as regression or reinforcement learning.

