# OpenReview forum: "Geometric Inductive Biases of Deep Networks: The Role of Data and Architecture"
_ICLR.cc/2025/Conference — ICLR 2025 Spotlight_

### Official Review · Reviewer_sttW · 2024-10-20

**Soundness:** 4
**Presentation:** 2
**Contribution:** 3
**Rating:** 8
**Confidence:** 3

**Summary:**

The authors study the input-space geometry of neural networks, i.e., the decision boundaries and loss surface at each point in the input space. They give mathematical characterizations of the expected geometry and dynamics of the geometry under gradient flow at initialization. They make some intuitive conjectures about how the geometry evolves under training, one of which is the Geometric Invariance Hypothesis, which says that the geometry of the model remains invariant in "un-important" directions of the input space while training. They then experimentally verify parts of their conjectures using some representative architectures on both synthetic data and smaller-scale real data.

**Strengths:**

- The examination of input-space geometry of neural networks is a more novel approach: other more mainstream approaches such as NTK and AGOP consider the parameter-space geometry.
- The conclusions are interesting, and to some degree, sensible: if there are directions in the input space which do not support the training data distribution, then the training dynamics may not move the input-space geometry in these directions.
- The paper does a good job motivating a very theoretical research direction with plausible practical implications in Section 5.
- The paper is reasonably well-written; despite the material being technically dense, it mostly makes sense after a thorough read.

**Weaknesses:**

- The definitions of geometry in this work are based on expected second moments of the data. This captures the smallest supporting subspace of the data. But data can have very large linear dimension while having small intrinsic dimension, cf "manifold hypothesis". The theory here does not seem to capture this more involved geometric aspect, but this assumption has been hypothesized to be more realistic for real data. The lack of understanding here makes the experiments less likely to track with the theory and may preclude more prescriptive insights, beyond those generated by the preliminary experiments in Section 5.
- More minor: There is not much intuition provided about why the predictions made at initialization should transfer reasonably well to training, and thus (in my opinion) not too much motivation for the Conjectures (which are about training dynamics).

**Questions:**

- The experiments remove normalization from the neural networks. Is this crucial? Does the trend totally deteriorate with normalization? Is there any hypothesis made by the theory as to why?
- Regarding the Conjectures 1 and 2 about dynamics: Is there an infinite- or large-width assumption being implicitly made, for instance? Should readers interpret the Conjectures as holding in appropriate scaling limits or should they hold at reasonable scales?

---

> ### Author Response · Authors · 2024-11-24
> **Official Response to Reviewer sttW (1/2)**
>
> We thank you for the helpful comments and questions. We would like to note that we have uploaded a revised version of the paper based on the comments, and we are currently waiting on experiment results for ResNet9 and Tiny ImageNet, which we hope to be able to upload as soon as possible.
>
> About your comment on the definitions of geometry in this paper, we thank you for this very interesting point and perspective. Answering this question is of course challenging given the nonlinear nature of generic data manifolds, we however like to point out that the dataset in Figure 1 is of a “donut” shape, and hence with nontrivial geometry. We note that almost all existing results characterizing dynamics during training do not rely on a precise characterization of the data manifold. Actually, most times, such a characterization is simply not necessary. Take the example of linear regression with data matrix $\mathbf{X}$ (datapoints $\mathbf{x}$ feature dimensions). One could characterize the manifold shape of the data, but it is well known from optimization theory that in this setting the only quantities characterizing convergence are the eigenvalues of $\mathbf{X}^\top \mathbf{X}$. Note that this is tight and rolled in the invariance property of linear dynamical systems with respect to rotations. In other words, characterizing the data manifold beyond linear arguments~(i.e. Eigendecomposition of covariance) is simply not providing any additional information to understand the macroscopic rotation-invariant aspects of SGD trajectory (e.g. convergence speed).
> Conceptually, our analysis relies on a similar observation: local changes in the quantities we analyze can be described by local linearizable gradient updates (i.e. stepsize not big enough). As such, only the linear properties of the data distribution are sufficient to derive our theoretical results, which we confirm extensively in our experiments.
>
> With respect to your comment on intuitions on why the predictions made at initialization should transfer reasonably, we agree that a fundamental limitation of most deep learning research is a lack of tools for the analysis of landscapes and trajectories beyond random initialization. Note however that **(1)** understanding initialization here provided us with useful intuition confirmed by our experimental results. We do not overclaim, but use theory to guide our intuition and provide a satisfactory level of formality. **(2)** many (if not all) milestone results in optimization for deep learning, concerning gradient dynamics, are inspired by (and only proved in) the initialization setting. A notable example is Glorot/He initialization, as well as all the literature on rank collapse in transformers [1]. Other examples include initialization schemes such as ReZero [2] and SkipInit [3]. The analysis of PreLN is also similar [4]. These insights – specific to initialization and random weights, unlocked training of neural networks at scale also beyond initialization.
>
> Regarding your first question about removing normalization from the neural networks in some of the experiments, we explain the reason behind the removal of normalization in certain experiments in footnote 2. In our preliminary experiments, we noticed that when different patches of the data have very different variances (such as the CIFAR-10 data), it is much better to compute $\mathbf{G}\_{\mathcal{F}}$ with the normalization layers set to the training mode (i.e., the normalization statistics being computed). The reason behind this observation is explained in Corollary A.4, where we observe that $\mathbf{G}\_{\mathcal{F}}$ is affected by the variance of each patch of the data due to the existence of normalization. While on its own this does not seem like an important issue, it would go against our aim at separating the effect of data and architecture over the input-space geometry, since in this case, $\mathbf{G}\_{\mathcal{F}}$ would become input-dependent as well. Therefore, based on this principle we decided to exclude the normalization layer from the models that utilize them when dealing with datasets that are not isotropic enough for $\mathbf{G}\_{\mathcal{F}}$ to be computed in an input-independent manner.
>
> Regarding your second question about Conjectures 1 and 2, we purposefully excluded any assumptions about the model in the description of these conjectures, because the models used to confirm them in our experiments do not adhere to any of these limitations. For instance, the LeNet model is extremely small (with about 60k parameters), and thus does not support the idea that we need the assumptions used in Theorems 3.1 and 3.2 for the claims in the Conjectures 1 and 2 to hold true.

---

> ### Author Response · Authors · 2024-11-24
> **Official Response to Reviewer sttW (2/2)**
>
> --------
> **References**
>
>
> [1] Noci, L., Anagnostidis, S., Biggio, L., Orvieto, A., Singh, S.P. and Lucchi, A., 2022. Signal propagation in transformers: Theoretical perspectives and the role of rank collapse. Advances in Neural Information Processing Systems, 35, pp.27198-27211.
>
> [2] Bachlechner, T., Majumder, B.P., Mao, H., Cottrell, G. and McAuley, J., 2021, December. Rezero is all you need: Fast convergence at large depth. In Uncertainty in Artificial Intelligence (pp. 1352-1361). PMLR.
>
> [3] De, S. and Smith, S., 2020. Batch normalization biases residual blocks towards the identity function in deep networks. Advances in Neural Information Processing Systems, 33, pp.19964-19975.
>
> [4] Xiong, R., Yang, Y., He, D., Zheng, K., Zheng, S., Xing, C., Zhang, H., Lan, Y., Wang, L. and Liu, T., 2020, November. On layer normalization in the transformer architecture. In International Conference on Machine Learning (pp. 10524-10533). PMLR.

---

> ### Comment · Reviewer_sttW · 2024-11-24
> **Response to Rebuttal**
>
> Thanks for the detailed response. It is true that you only need the covariance structure at initialization for results, and not any more fine-grained characterization of the local geometry. And I do acknowledge that obtaining results about training dynamics is very difficult. Perhaps the nonlinearities may play a more sophisticated role in the training dynamics: a natural hypothesis is that they govern the radius up to which the local linear approximation is valid. A characterization of this interplay may be extremely interesting, but of course also extremely challenging, and out of scope of this work.
>
> Your point about previous work only examining initialization and using it to derive conclusion is also valid. Nonetheless, my point was more about the exposition rather than technical work --- in my opinion it is not super clear to see how your analysis at initialization can give you intuitions about training throughout the trajectory. If that were improved the paper would make more sense.
>
> Responding to the normalization point: Thank you for the explanation of this. I wonder if it's possible to show in a very toy/simple experimental setup that if the assumptions are followed exactly then the results of the theorems should follow exactly.
>
> I will maintain my score.

---

> > ### Author Response · Authors · 2024-11-25
> > **Official Response to Rebuttal of Reviewer sttW**
> >
> > Thank you for your further comments.
> >
> > About characterizing training dynamics beyond initialization, we agree it would be highly desirable and informative. Yet we believe such a careful analysis can only be possible in the asymptotic setting of NTKs, $\mu$P, or DMFT. In all these cases, we need to make assumptions about parametrization and training setup. In the spirit of trying to keep the analytical part mainly as a source of inspiration and use experimentation with non-trivial settings to validate our results, we restricted attention to a more classical analysis. We would also like to point out that the paper is already dense with information; yet, we believe a more thorough investigation on the interplay between, e.g., GIH and feature learning scaling limits, would be an excellent direction for future research.
> >
> > Regarding your question on the normalization point, in case you are referring to Theorems 3.1 and 3.2 in the body of the paper, we do in fact have experimental results confirming the theorems. You can see those results in the Appendix in Figure 7. In case you instead are referring to the Theorems on normalization (Corollary A.4 in the Appendix), then we would like to point out that the results of the Corollary are more difficult to validate since this involves upper-bounds and lower-bounds on the function $\mathbf{G}\_{\mathcal{F}}$. However, we would like to emphasize that the theory does indeed capture how normalization reacts with GIH (Corollary A.4) - and our experimental results comprise both normalized and unnormalized models to amplify our spectrum of investigation.

---

### Official Review · Reviewer_tWDw · 2024-11-03

**Soundness:** 4
**Presentation:** 4
**Contribution:** 4
**Rating:** 8
**Confidence:** 2

**Summary:**

This paper introduces the geometric invariance hypothesis (GIH), which suggests that during the training of neural networks, the curvature of the input space remains invariant under transformations along specific directions determined by the network’s architecture. The authors begin by examining a non-linear binary classification problem situated on a hyperplane in a high-dimensional space. They observe that a multilayer perceptron (MLP) can solve this problem regardless of the hyperplane’s orientation, whereas a residual network (ResNet) cannot. Motivated by this finding, they define two architecture-dependent maps: the average geometry, which provides a compact summary of the network’s input-space geometry, and the average geometry evolution, which describes how this geometry changes during training. Through both theoretical analysis and empirical investigations, the paper demonstrates that GIH arises because the average geometry evolution closely approximates the projection of the data covariance onto the average geometry. This results in an invariance property, especially when the average geometry is low-rank. Finally, the authors present extensive experimental results to explore the consequences of GIH and its relationship to generalization in neural networks.

**Strengths:**

The introduction of the geometric invariance hypothesis (GIH) offers a fresh and innovative perspective on the interplay between neural network architectures and the geometry of the input space during training. By proposing the concepts of average geometry and average geometry evolution, the authors provide novel tools for quantifying how different architectures influence learning dynamics. This approach moves beyond traditional analyses by directly linking architectural properties to geometric transformations in the input space, which is a significant conceptual advancement in understanding deep learning models.

**Weaknesses:**

While the paper makes significant contributions, there are areas that could be improved:

- Lack of Intuitive Explanation: It is challenging to develop an intuition for why ResNets behave differently from MLPs. Providing more intuitive explanations or illustrative examples before introducing the mathematical formalism would help readers grasp the core concepts and follow the subsequent analysis more effectively.
- Limited Architectural Comparison: The focus on ResNets without discussing other architectures like AlexNet leaves some gaps. Clarifying whether the observed behaviors are due to specific features like skip connections or are common across different architectures would strengthen the generality of the findings.

**Questions:**

I am interested in how your observations are affected by different training dynamics, specifically when using stochastic gradient descent (SGD) with large learning rates—a common practice for achieving high prediction accuracy. Have you explored how the geometric invariance hypothesis (GIH) and the behavior of average geometry and its evolution manifest under such training conditions?

---

> ### Author Response · Authors · 2024-11-24
> **Official Response to Reviewer tWDw**
>
> We thank you for your helpful comments, questions, and suggestions. We would like to note that we have uploaded a revised version of the paper based on the comments, and we plan on providing further experiments on ResNet9 and the Tiny ImageNet dataset on OpenReview as soon as possible.
>
> Regarding your first point about an illustrative example, we have added Appendix A.10 to the paper providing an intuitive example of a model with low-rank average geometry and how GIH predicts its learning dynamics. You can find the new section in our revision of the paper.
>
> About your second point concerned with the limited architecture comparison, we would like to first note that the majority of the work presented in this paper was done as independent research and with very limited resources. Despite these conditions, we tried to make sure that all of the popular architecture choices were present in our experimental setup. Namely, we have simple convolutional neural networks (LeNet), CNNs with skip-connections (ResNet18), and self-attention-based models (ViT). Furthermore, the selection of models we used also covers a wide range of parameter size, from 60k (LeNet), to 4M (ViT), and 11M (ResNet18). However, we will make sure to also provide experiment results for medium-size neural networks without skip-connection. Specifically, we are currently running experiments with ResNet9 without skip-connections [1], the result of which will be presented here upon completion.
>
> Regarding your question about the training dynamics of SGD, we think this is a very important and engaging point that you’ve brought up. In order to answer this question, we performed the experiment from Appendix A.7 (Figure 10) for the learning rates 1.0, 0.1, 0.01, and 0.001 for the ResNet18 model. For a detailed discussion and the experiment results, please refer to Appendix A.9 in the revised version of the paper. As evident by the results, we can see that both very large and very small learning rates seem to hurt the generalization capabilities of the model. However, it seems to be the case that in all these settings, the results are in line with GIH.
>
> -------------
> **References**
>
> [1] Yao, Z., Gholami, A., Keutzer, K. and Mahoney, M.W., 2020, December. Pyhessian: Neural networks through the lens of the hessian. In 2020 IEEE international conference on big data (Big data) (pp. 581-590). IEEE.

---

> ### Author Response · Authors · 2024-11-29
> **Official Response to Reviewer tWDw: Further Experimental Results**
>
> Once again, we would like to thank the reviewer for the interesting comments and ideas. We are looking forward to discussing any remaining concerns that you may have in the remainder of the rebuttal period.
>
> We would like to also inform you that we have been able to perform the experiments addressing your concerns regarding further diversity in the experimental setup. Specifically, we have performed the experiments from Figure 4 and Figure 6 on the ResNet9 architecture without skip-connection. You can find the experiment results for the generalization gap (Figure 4) [here](https://drive.google.com/file/d/1WFMtjpsjC5GyppGE5HGFfn9FnMBXO_L8/view?usp=sharing), and the experiment results for the orthogonalization experiment (Figure 6) [here](https://drive.google.com/file/d/1mk5xVjGwYNlGPGLOSyKEHuOWRMwAK10v/view?usp=sharing). We decided to use the ResNet9 architecture without skip-connection since you specifically pointed this architecture choice in your concerns. We would also point to the other architecture used in our experiment, namely LeNet, which also lacks this feature and still behaves similar to a ResNet18 with respect to our claims in the paper.
>
> As evident by the results, we can see that the observations are still in line with the predictions of GIH, which indicates the architecture independent nature of our claims in the paper. We hope that these results combined with the newly added sections focused on learning rate dynamics and the dataset size completely addresses your concerns.

---

### Official Review · Reviewer_xFyb · 2024-11-04

**Soundness:** 3
**Presentation:** 4
**Contribution:** 3
**Rating:** 8
**Confidence:** 3

**Summary:**

In their paper, the authors argue for the link between input-space geometry and the ability of models to generalize. They propose the geometric invariance hypothesis (GIH) which states that, depending on the architecture of the model, the curvature of the input-space remains invariant under transformations in certain directions. They then attempt to prove and study this phenomenon by separating the changes in the input-space geometry during training into architecture-dependent changes and data-dependent changes. This resulted in finding the special cases where the invariance occurs, and they link this to the generalization ability of the model and where it collapses to reliance on noise and memory.

**Strengths:**

The paper is able to gradually build up to the main hypothesis being proposed while maintaining a clear chain of reasoning. The authors also provide extensive mathematical proofs for each step in the build-up and mention what assumptions are made and any limitations on what can be shown or derived. Finally, they are able to provide some insight into the effect of this hypothesis on an architecture's generalization ability while addressing any possible ideas with empirical results.

**Weaknesses:**

While the "performance" gains of the paper do seem marginal, I see these experiments as more of a proof of concept of the ideas and the proposed hypothesis. However, it would have been nice to see these experiments on multiple datasets to verify if the claims still hold, especially given the simplicity of the current model choices as well.

**Questions:**

Just as a small note, in line 211 you mention the interaction between two factors, specifically the data-dependent factor and another. I'm guessing this is the model's geometric inductive bias? If so, I think it could be written a bit clearer.

Additionally, would you think this approach would be useful for architecture-based optimal dataset subsampling methodology?

---

> ### Author Response · Authors · 2024-11-24
> **Official Response to Reviewer xFyb (1/2)**
>
> We thank you for your insightful and interesting comments. We would like to note that we are currently working on further experiments on the Tiny ImageNet dataset and the ResNet9 model, the result of which we plan on uploading on OpenReview as soon as possible.
>
> Regarding the performance gains, we agree that, as you noted, the main purpose of our experiments is to provide insight and verification for the ideas and the hypothesis proposed in the paper. However, while modest, we would like to point out that the results are statistically significant.
>
> Concerning our experimental setting and our use of small models and datasets, we would like to note that most of the research presented in this paper was done with limited resources. However, in designing the experiments and selecting the models and datasets, we tried to make sure that all types of architectures commonly used in practice are present in the results, and we also covered a range of model sizes (from LeNet with ~60K parameters, to ViT with ~4M parameters, and  RestNet18 with ~11M parameters). However, following Reviewer #1’s suggestions we were able to investigate the effect of datasize on GIH, the results of which you can read in Appendix A.8. Specifically, we performed the experiments of Figure 3 for a ResNet18 with larger datasizes and the experiment of Figure 4 and Figure 10 with larger datasets for LeNet. In the first experiment, the LeNet model is on the verge of under-parameterization with exactly the same number of train data points as parameters. However, we observe that the model adheres to GIH despite the larger datasize. However, in the second experiment involving ResNet18, we observe that the effect of GIH on train accuracy starts to disappear with larger train data sizes. But we were able to show that the main cause of this disappearance is not the datasize itself, but rather the increase in the number of mini-batches. Specifically, by increasing the batch size along with the datasize, so that the number of mini-batches in each epoch remains the same, we can observe that the behavior of the model again becomes very similar to the observations provided in Figure 3. Therefore, we hypothesize that the effect of GIH on the train data is at least partly caused by how far away the parameters are from the initial point. So our hypothesis is that the behavior in Figure 3 happens when the optimization path is not long enough for the changes in the average geometry to become large enough that the initialization point becomes completely irrelevant. However, we were able to confirm that this effect does not transfer to test accuracy (and thus, generalization), since we observed that increasing the learning rate for the experiments in Appendix A.7 does not change the behavior of the model. For more explanation, please refer to Appendix A.9 in the revised version of the paper.
>
> For the ImageNet data, we would like to point out that given the image size in this dataset, computing and storing the $\mathbf{G}\_{\mathcal{F}}$ function would be extremely difficult and expensive, both in terms of memory and required computation. As a result, we are instead going to focus on the Tiny ImageNet dataset, which is a subsample of ImageNet with much smaller images (in the 64x64 range) and data size. We will be adding the results of these experiments here when they're available.

---

> ### Author Response · Authors · 2024-11-24
> **Official Response to Reviewer xFyb (2/2)**
>
> Regarding your second question on the usefulness of GIH for architecture-based optimal dataset subsampling, we absolutely think the findings of the paper can be helpful both to architecture search where one tries to tailor an architecture to a specific data, and dataset subsampling where one tries to tailor the data to a specific architecture. Similarly, the results can also be helpful when we in general try to condition a dataset to train/generalize well on an architecture. Given the results of the paper, in all of these cases, one can use the average geometry function to make sure the features of the dataset match the “geometric inductive biases” of the model. However, we would like to point out two caveats:
>
> 1. We do not claim to capture all of the possible inductive biases of a neural network. For instance, there are results pointing out architecture-dependent inductive biases in the frequency domain [1-3], while also results that show similar effects caused by activation functions. However, we are hoping to provide a more tangible and interpretable way to capture the inductive biases of a neural network that relate to the directions in which the features reside.
>
> 2. Our results in Conjecture 1 and 2 are based on approximations for the average geometry evolution. While they seem to be good approximations as the predictions of Conjectures 1 and 2 are confirmed by empirical results, they can still in theory be arbitrarily inaccurate. In this case, we would suggest computing Delta directly and perhaps accumulating it through the training, which in our experiments provided much better margins for the experiments in Section 5.
>
> -------------
> **References**
>
> [1] Wen, Y. and Jacot, A., 2024. Which Frequencies do CNNs Need? Emergent Bottleneck Structure in Feature Learning. arXiv preprint arXiv:2402.08010.
>
> [2] Ronen, B., Jacobs, D., Kasten, Y. and Kritchman, S., 2019. The convergence rate of neural networks for learned functions of different frequencies. Advances in Neural Information Processing Systems, 32.
>
> [3] Teney, D., Nicolicioiu, A.M., Hartmann, V. and Abbasnejad, E., 2024. Neural Redshift: Random Networks are not Random Functions. In Proceedings of the IEEE/CVF Conference on Computer Vision and Pattern Recognition (pp. 4786-4796).

---

> ### Author Response · Authors · 2024-11-29
> **Official Response to Reviewer xFyb: Further Experimental Results**
>
> Once again, we would like to thank the reviewer for the helpful comments and ideas to improve the paper. We would be happy to discuss the paper further with you in case you have any remaining concerns that need addressing.
>
> We have been able to perform the extra experiments we referred to in our initial response. Specifically, we have performed the experiments from Figure 4 and Figure 6 on the Tiny ImageNet dataset to further diversify our experimental settings and address your concern regarding the effect of datasize on GIH. You can find the experiment results for the generalization gap (Figure 4) [here](https://drive.google.com/file/d/1RZ8gKZS_TbsWZKU6RU6jg1qWcS7uU__4/view?usp=sharing), and the experiment results for the orthogonalization experiment (Figure 6) [here](https://drive.google.com/file/d/1VrIo82MsbZe1B2UJsUB7QB1M2S9XBFDz/view?usp=sharing). We would like to note that the reason we chose Tiny ImageNet instead of the original ImageNet for our experiments was the size of the data and the images themselves, which would make working with the average geometry function $\mathbf{G}\_{\mathcal{F}}$ very difficult and expensive.
>
> As evident by the results, the datasize (both in terms of image size and the number of datapoints) does not negate the results of GIH. This is in line with our findings from the newly added Appendix A.8, which we hope completely answers your concerns regarding the verification of the claims in the paper for more complex datasets.

---

### Official Review · Reviewer_wQbe · 2024-11-04

**Soundness:** 2
**Presentation:** 2
**Contribution:** 2
**Rating:** 6
**Confidence:** 4

**Summary:**

The paper studies how the geometry of neural network predictors evolves in the input space during training. It proposes the Geometric Invariance Hypothesis (GIH), which posits that this evolution occurs within a constrained subspace of the input space determined by the network architecture. The GIH is supported through experiments with MLPs, CNNs, and ViTs on a subset of CIFAR with binarized labels. The paper also studies the link between the GIH and generalization.

**Strengths:**

1. The introduction of the Geometric Invariance Hypothesis appears novel and extends findings of Neural Anisotropy Directions (Ortiz-Jimenez et al., 2021) to non-linear decision boundaries. This hypothesis has the potential to provide insights into the relationship between neural network architecture and the structure of data, contributing to our understanding of inductive biases in deep learning.

2. The experiments and the theoretical analysis are generally fair, although several imprecisions are present, and some clarifications are needed (see detailed points below).

3. The theoretical findings are tested using real networks (e.g., ResNets, ViTs) and datasets (CIFAR). These experiments appear to align with and support the GIH. Nevertheless, it is worth noting that the considered settings are highly overparameterized and relatively simple, which raises concerns regarding the broader validity and general significance of the results (see detailed points below).

**Weaknesses:**

The paper is quite dense. There are multiple points of confusion and imprecisions affecting both clarity and soundness. Specifically:

4. The introduction and main text lack a comprehensive overview of the field and references to related work. Only a few broad papers are cited, despite the extended page limit of this year's edition. I strongly encourage the authors to move much of the discussion from Appendix A.1 into the main text to better place the work in context. In particular, NADs introduced in Ortiz-Jimenez et al. (2021) should be explicitly discussed in the main text, given how strongly related they are to this work.

5. Lines L058+ seem to confuse expressivity with generalization, i.e., approximation vs. statistical properties. The fact that an MLP is universal (as some deep and wide CNNs are) does not imply that it can generalize effectively.

6. Assuming i.i.d. parameters is not enough to conclude that $G_F \propto I$. A Gaussian distribution with a non-zero mean is a counterexample.

7. The claim that MLPs have a lack of inductive bias (L210, L224, L249) is wrong. Even in the NTK regime, MLPs display a strong spectral bias for low modes. It may be more accurate to state that MLPs at initialization have no preferred directions in the input space, i.e., they have an isotropic prior in the input space.

8. Multiple terms are left undefined or used imprecisely: e.g., “SSE loss” and $y_{\mu}$. Some phrases are vague: What is the “input-space curvature”, and how can it depend on the training process (L011)? What is the “input geometry” of a neural network (L076, L521)? What does it mean for a dimension to be noise (L053)? What are “all the possible values of $\theta$” ( L145)? How are they distributed? Given that you consider gradient-flow in your analysis, how do you define the “first step” in L286?

**Questions:**

9. From Figure 1, it seems that simple (and standard) early stopping would solve the generalization gap problem and actually result in a lower test loss for dataset $D_B$ compared to dataset $D_A$. Can you comment on this? How general is this observation?

10. What’s the underlying distribution over which you are taking the expectation at L227? It should be made explicit.

11. When you write “momentums” (L244, L246) do you mean “moments” instead?

12. Is the fact that $\Delta_F$ becomes label-independent purely due to the fact that you assume $\mathbb{E}_\theta [\theta]=0$ at initialization?

13. Could the phenomena observed in the paper, particularly when testing conjecture 2 about the GIH, be due to the fact that networks in all experiments are strongly overparameterized (even with CIFAR, the paper considers only a small fraction of the full dataset) and training in the “lazy” or NTK regime where weights stay close to their initial values? Did the authors measure the weight evolution throughout training? Would you expect the same results when training with more data, e.g., the full CIFAR10 dataset, and/or with tasks requiring learning latent features of the data that might be less dependent on the exact geometry and statistics in input space, e.g., more complex image datasets such as ImageNet?

14. Could the authors comment on Conjecture 2 in the case of linear regression? Intuitively, it seems it should hold in that simpler setting.

15. What is the motivation behind adding label noise (L406)? Would the same results or trends be observed in its absence?

---

> ### Author Response · Authors · 2024-11-24
> **Official Response to Reviewer wQbe (1/4)**
>
> We thank the reviewer for their response and helpful suggestions. Hereby we try to provide an initial response to your points. We have uploaded a revised version of the paper along with these responses. However, we would like to note that we are also waiting on additional experiments, specifically on the Tiny ImageNet dataset, to further diversify our experimental results.
>
> While your questions gave us the opportunity to improve our work, we did not fully identify the reasons behind your rejection score, as most of your points are addressable with further experiments or text editing. We believe points 6, 10, and 11 deserve the most attention and we thank the reviewer for the insightful questions. We took all your concerns seriously and that led us to further experiments that we think greatly aid the robustness of our reasoning.
>
> First, regarding the size of the paper, we would like to note that our aim was first and foremost clarity. Given that the idea is relatively novel and there are not many related works (aside from the NADs paper [1]) we felt the need to be as detail-oriented and meticulous as possible in our definition, theoretical results, and experimentation. However, we absolutely welcome your suggestions about further improving the content and making the paper more concise and accessible.
>
> Now, regarding your listed points:
>
> 4. We thank you for the helpful suggestion. For the sake of space, we would suggest that perhaps a compromise can be made by just mentioning the most relevant papers to our work in the introduction in a small related works subsection, and then referring the reader to the much more dense related works section in the appendix. In the revision of our paper, we have added a small related works section discussing the NADs paper and its follow up [2] in more detail.
>
> 5. Thank you for pointing out this clarity issue. By “solving”, we are here explicitly referring to the ability of MLPs to fit the decision boundary in the training data. However, what we discuss and comment about later is of course generalization. We have clarified this in our revised version of the paper and edited the part you mentioned.
>
> 6. We are assuming you are referring to the second paragraph of Section 3.1 (lines 222-232). In that case, you are correct. The condition for the parameters should be i.i.d. and zero mean. We have fixed this issue in our revised version of the paper.
>
> 7. Thank you for this comment. While you are correct about line 249, we would like to note that we purposefully used the word “geometric” as a way to make the distinction that you are referring to in lines 210 and 249. So by “geometric inductive biases,” we are referring to the findings of our paper which attributes an isometric prior in an MLP. However, if you prefer another term, we are open to suggestions. For instance, we also had another candidate for this term, namely “directional inductive biases.” Furthermore, in the revised version of the paper we have specifically made a distinction in a footnote inside the introduction, stating that we are focusing on the geometric inductive biases at initialization.

---

> ### Author Response · Authors · 2024-11-24
> **Official Response to Reviewer wQbe (2/4)**
>
> 8. Thank you for these points, which helped us improve clarity: **a)** Regarding the SSE loss and $y_\mu$, we have clarified these two notations in Section 2.1. Furthermore, we would like to note that we have an extra section in Appendix A.2 explaining the SSE loss and its dynamics in detail. **b)** Regarding the term “input-space curvature,” we used it as a placeholder for “input-space geometry” in the abstract since curvature in the optimization community has a more diverse meaning that is not exclusive to the second-order derivative of the function and can be generally seen as the characteristics of the surface of the function. However, the term “geometry” can be more confusing as it can also refer to the type of geometry (e.g., Euclidean vs. non-Euclidean), the characteristics of the surface of the function, or many other things. This would merit a clarification, which would not be possible in the abstract considering the word count limit. We are, of course, open to suggestions. **c)** Regarding “input geometry” in line 521, that is a typo which we have fixed. We meant to write “input-space geometry,” which is also used in line 76. By the term “input-space geometry,” we mean to encapsulate all the information about the geometric characteristics of the surface of the model as an input-output function. However, this term can admittedly be vague. So we are open to suggestions about ways to make sure this term does not cause any confusion. **d)** By the other dimensions being “uncorrelated noise” in line 53, we mean the variations in the data in those dimensions are coming from an i.i.d. and zero-mean Gaussian distribution. We have clarified this term in the paper. **e)** By all the possible values of $\theta$ in line 145, we mean its distribution. For instance, at initialization, $\theta$ is usually sampled from i.i.d. zero-mean Gaussian or uniform distributions. On the other hand, we could also compute the expectation over a specific point at training, which means the distribution will be induced by the stochasticity introduced from the training procedure as well as initialization. This section of the paper is purposefully written in a way that makes it possible to avoid making any assumptions about the model and its parameters in the definitions. We have replaced this line with “over a distribution of $\theta$” in the revised version. **f)** Regarding the gradient-flow model vs. gradient descent in line 286, that was a typo that we fixed in the revised version of the paper.
>
> 9. Thank you for your comment. We note that this misunderstanding is caused by our use of test loss instead of test accuracy, which is sometimes not a good metric to assess how well a model generalizes. Instead, we believe accuracy would be a better metric for assessing the model in terms of generalization. So we have provided the plots of Figure 1 with test accuracy instead of test loss in the revision of the paper. As you noted, in the first few epochs of the training the test loss dips down in both datasets but later stabilizes at a larger point in $\mathcal{D}_B$ compared to $\mathcal{D}_A$. While it seems like there might be a point in the first few epochs with higher accuracy in $\mathcal{D}_B$, this is not the case in practice. In fact, as we can see in the new plots, in both datasets there’s an almost monotonous increase in the accuracy during training, with the test accuracy of $\mathcal{D}_B$ stabilizing at around 70% while in $\mathcal{D}_A$, this number is above 90%. So the decrease of the test loss at the first few epochs seems to be due to the model starting to learn the decision boundaries around $u_1$ in $\mathcal{D}_B$ while the increase that happens later seems to be due to the model starting to “memorize” the noise patterns in the other dimensions. This behavior is present in all our models, corroborating our results and claims in the paper. We would like to also point out that this example is just a motivating experiment and should be considered along with the other results and experiments in the paper.
>
> 10. Thank you for pointing out the missing information in line 227. The expectation is computed over $\theta_0$, which is assumed to be i.i.d. Gaussian as is the common practice in modern machine learning. We have explicitly pointed this out in the revised version of the paper.
>
> 11. Yes, that is a typo. Thank you for pointing it out. We have fixed it in the revised version.
>
> 12. Yes. As you know, it is common practice in deep learning to initialize models with i.i.d. zero-mean Gaussian or uniform distributions. Furthermore, we do not consider the case of pre-training, as it will introduce arbitrary distributions over the parameters at initialization.

---

> ### Author Response · Authors · 2024-11-24
> **Official Response to Reviewer wQbe (3/4)**
>
> 13. We thank you for the interesting question. We believe our experimental setting does not align with the NTK regime, as this requires a specific reparameterization of the model which we do not follow [3]. On the other hand, we are confident that we do not reside in the lazy training regime [4] either, since residing in the lazy training regime without NTKs requires a particular training regime and parameterization that we do not try to satisfy [5]. However, for the sake of completeness, we looked at the normalized distance between the initialization point of $\theta$ and further iterates to make sure that our training regime does not satisfy the requirements for linearization [4]. In this experiment, we observed that this distance actually increases significantly in the case of the data sampled from $\mathcal{N}(0, \mathbf{G}\_{\mathcal{F}})$ in Figure 3, with the distance being around 0.2 for ResNet18 and 1.8 for LeNet at convergence (you can see the relevant plots for LeNet [here](https://drive.google.com/file/d/1jEq9L_lqUPU6NHkD4Rx42L9EagaVCMWk/view?usp=sharing) and for ResNet18 [here](https://drive.google.com/file/d/1REFAk6EJ5y-ubJ_7EfYOlwGfcJvtxFWV/view?usp=sharing)). Therefore, it is safe to say that our model cannot be equivalent to its first-order Taylor expansion. However, your curiosity regarding the main source of GIH and the effect of over-parameterization led us on an interesting further path. For a detailed explanation, please refer to the newly added Appendix A.8 of the revised paper. In the case of over-parameterization, we can refute that GIH is caused by over-parameterization by just looking at LeNet. LeNet (or as it used to be called, LeNet 5, for having five layers) is a small network by today’s standards, with just above 60k parameters. So we designed the following experiment where we reduced the number of parameters of LeNet to around 50k by making the fully connected layers slightly smaller, and then used the same number of samples from CIFAR-10 as parameters (i.e., around 50k samples) and performed the experiments from Figure 4 (you can see the relevant plot in Figure 11a of the revised paper). In this case, one can see that the model will reside on the verge of under-parameterization. Despite this quality, the model is still performing almost identically to the results of Figure 4. In order to make sure this observation cannot be solely attributed to the characteristics of CIFAR-10, we also performed the same experiment but on the synthetic data introduced in Appendix A.7, where the data is almost isotropic and sampled from a Gaussian distribution (you can see the relevant plot in Figure 11b of the revised paper). Still, the results were identical, which rules out over-parameterization. However, since for a ResNet18, it would be very difficult to perform a similar experiment, we instead tried to look at the training dynamics we observed in the experiment from Figure 3. Specifically, for both of the datasets sampled from $\mathcal{N}(0, \mathbf{G}\_{\mathcal{F}})$ and $\mathcal{N}(0, \text{flip}(\mathbf{G}\_{\mathcal{F}}))$, we increase the train data size from 10000 to 100000, while keeping the batch size at 128 (you can see the relevant plots in Figure 12 of the revised paper). In this case, we actually observe that the effects of GIH over the **train accuracy** start to vanish with larger datasets. So at first glance, it may seem like over-parameterization may have some effect on GIH when it comes to train accuracy. However, we were able to refute this explanation by increasing the batch size along with the datasize, so that the number of mini-batches in each epoch remains the same (you can see the relevant plots in Figure 13 of the revised paper). In this case, we observe a very interesting phenomenon: the model again reverts back to its original behavior observed in Figure 3, regardless of data size. This indicates that the main factor causing the behavior in Figure 3 may not be the data size, but rather how far away the parameters are from the initial point. So we hypothesize that the behavior in Figure 3 happens when the optimization path is not long enough for the changes in the average geometry to become large enough that the initialization point becomes completely irrelevant. A good example of this condition not being satisfied is our experiment with ResNet18 with larger datasets (but constant batch size), wherein the changes in the geometry are monotonically increasing with the train data size. On the other hand, in the case of having larger batch sizes, the changes in the geometry remain about the same despite the increase in data size. However, this does not mean we are in the lazy training regime, as the linearization behavior (i.e., the model is equivalent to its first-order Taylor expansion) is not present.

---

> ### Author Response · Authors · 2024-11-24
> **Official Response to Reviewer wQbe (4/4)**
>
> 13. *(continued)* However, borrowing the idea of seeing the effect of learning rate from Reviewer #3, we tried to answer this question: does elongating the optimization path artificially by increasing the learning rate have a similar effect on the test accuracy? In order to answer this question, we performed the experiment from Appendix A.7 with the synthetic data on a ResNet18 with learning rates 1.0, 0.1, 0.01, and 0.001 (you can see the relevant plots in Figure 14 of the revised paper). Remarkably, we observe that this is not the case. The “preference” towards certain directions in all cases seems to exist, regardless of the learning rate size. However, interestingly we see that smaller learning rates actually perform worse, which can be attributed more to overfitting than GIH. We note that in all these cases, we were able to confirm that the distance between the initialization point of the parameters and the final point monotonically increases with the learning rate size. On a final note, we would like to emphasize that we are currently performing some of the experiments in the paper on the Tiny ImageNet dataset, the result of which will be available on OpenReview after completion.
>
> 14. Thank you for the interesting question. A linear regression model does indeed fit the description of GIH, wherein $G\_{\mathcal{F}}$ is equal to identity (i.e., we do not have a geometric inductive bias). However, we would like to **caution about mistaking expressivity for learnability**. GIH is strictly about learnability with the underlying assumption that we have expressivity. In other words, the model needs to be expressive enough in the first place to be able to solve the optimization problem associated with a task, for the results of GIH to be applicable to it. In the case of a linear regression model, it is expressive enough for solving a linear classification problem, and per the results of the NADs paper [1], it is completely capable of generalizing in all directions in the input space. However, this is not the case for more complex decision boundaries.
>
> 15. Thank you for asking for clarification about this issue. The reason behind adding label noise to the tasks in Figure 4 is to make sure that the optimal Bayes classifier for these tasks cannot achieve perfect accuracy. Given the distribution of the eigenvalues of the  CIFAR-10 data, the noise introduced by the orthogonal components in the data for most directions in the input space is usually very small. As a result, without the label noise, we would observe a smaller margin of performance for different directions of the discriminant feature. So this decision was made purely in the interest of better visualization. If you find it necessary, we could also perform the experiment without label noise and add the results to the revised version of the paper.
>
> -------------------
> **References**
>
> [1] Ortiz-Jiménez, G., Modas, A., Moosavi, S.M. and Frossard, P., 2020. Neural anisotropy directions. Advances in Neural Information Processing Systems, 33, pp.17896-17906.
>
> [2] Ortiz-Jiménez, G., Moosavi-Dezfooli, S.M. and Frossard, P., 2021. What can linearized neural networks actually say about generalization?. Advances in Neural Information Processing Systems, 34, pp.8998-9010.
>
> [3] Jacot, A., Gabriel, F. and Hongler, C., 2018. Neural tangent kernel: Convergence and generalization in neural networks. Advances in neural information processing systems, 31.
>
> [4] ​​Chizat, L., Oyallon, E. and Bach, F., 2019. On lazy training in differentiable programming. Advances in neural information processing systems, 32.
>
> [5] Fort, S., Dziugaite, G.K., Paul, M., Kharaghani, S., Roy, D.M. and Ganguli, S., 2020. Deep learning versus kernel learning: an empirical study of loss landscape geometry and the time evolution of the neural tangent kernel. Advances in Neural Information Processing Systems, 33, pp.5850-5861.

---

> ### Author Response · Authors · 2024-11-29
> **Official Response to Reviewer wQbe: Further Experimental Results**
>
> Once again, we thank the reviewer for the helpful suggestions and comments. We would be happy to further discuss your remaining concerns.
>
> We have been able to perform the extra experiments we referred to in our initial response. Specifically, we have performed the experiments from Figure 4 and Figure 6 on the Tiny ImageNet dataset to further diversify our experimental settings and address your concern regarding the effect of datasize on GIH. You can find the experiment results for the generalization gap (Figure 4) [here](https://drive.google.com/file/d/1RZ8gKZS_TbsWZKU6RU6jg1qWcS7uU__4/view?usp=sharing), and the experiment results for the orthogonalization experiment (Figure 6) [here](https://drive.google.com/file/d/1VrIo82MsbZe1B2UJsUB7QB1M2S9XBFDz/view?usp=sharing). We would like to note that the reason we chose Tiny ImageNet instead of the original ImageNet for our experiments was the size of the data and the images themselves, which would make working with the average geometry function $\mathbf{G}\_{\mathcal{F}}$ very difficult and expensive.
>
> As we can observe, the claims of the paper still hold in this more complex setting, and the size of the data (both in terms of image size, and in terms of the number of training samples) does not negate the conditions introduced by GIH. This observation is also in line with our experiments introduced in the newly added Appendix A.8, which we hope completely resolves your concerns in your point #13.

---

> ### Author Response · Authors · 2024-11-30
> **Gentle Reminder about Rebuttal**
>
> We would like to once again thank you for your helpful comments and ideas. In our responses, we tried to address all of your comments and concerns in our paper. However, we have not received a response from you, yet. Therefore, we assume all of your concerns are addressed. If that is the case, we would appreciate it if you could adjust your score accordingly, especially given that the rebuttal period is drawing to a close. If not, we would be happy to discuss the paper with you more.

---

> > ### Comment · Reviewer_wQbe · 2024-12-01
> >
> > I thank the authors for their detailed rebuttal.
> >
> > The original submission fell short of publication standards in several ways, including wrong mathematical statements, limited experimental evaluation, imprecise language, and undefined quantities. The revised version has effectively addressed many of these issues. However, the **language** used for presenting the new results still requires refinement, e.g., avoid constructions such as “you can see” in the paper.
> >
> > My primary remaining concern is the **scope of the experimental validation**. Both the original and novel results test the main paper’s conjecture in highly simplified contexts, such as small subsets of CIFAR10 and Tiny ImageNet with binarized labels, as well as small architectures like ResNet9 with removed residual connections or normalization layers. In fact, the reported test accuracies are quite low.
> >
> > Given that this is not a theory paper, I would have expected stronger empirical verifications of the proposed hypothesis in a more realistic and meaningful setting. Without such validations, the broader validity and impact of the findings is unclear.
> >
> > Taking all aspects into account, I find the paper to be **borderline between a rating of 5 and 6**. After some consideration, I have decided to increase my rating to 6.

---

> > > ### Author Response · Authors · 2024-12-02
> > > **Official Response to Reviewer wQbe**
> > >
> > > Thank you for your further feedback about the style of the writing. We will take care of any remaining issues in the camer-ready version of the paper.
> > > Regarding the experiments: we performed them at the maximum scale allowed by our compute constraints. Our experimental results span many architectures and data regimes, but if you have further directions in mind (at same scale) we would be happy to hear them.

---

### Author Response · Authors · 2024-11-25
**Global Response by the Authors**

We thank all the reviewers for their helpful suggestions and interesting and engaging comments. We have provided our responses to the reviewers, along with a revised version of the paper, in which we addressed their concerns with additional experiments and discussions. All of the edited or newly added sections have been distinguished from the rest of the paper with the color blue. Specifically, aside from fixing the typos and some of the text requiring clarification, we have also added a small related works section to the introduction following Reviewer #1’s demand. On the other hand, following Reviewer #1’s comment on the plots in Figure 1, we have replaced the test loss plot with the test accuracy plot, which would prevent any misunderstanding regarding the results of the experiment.

Also, given the comments of Reviewer #1 and Reviewer #2 about the effect of datasize on the results of the paper, we addressed this concern in a newly added appendix section (App. A.8). Furthermore, following Reviewer #3’s question regarding the effect of learning rate on the results of the paper, we have added another appendix section (App. A.9) to the paper addressing this question fully. Also, we have added another section to address Reviewer #3’s concern regarding an intuitive example in the appendix (App. A.10), where we provide a simple example of a model with a low-rank average geometry and how GIH predicts its behavior during training.

On a final note, we would like to point out that we are currently waiting on experiment results on ResNet9 without skip-connection and Tiny ImageNet in order to further diversify our experiment results. We will add the experiment results here as soon as they are available.

---

> ### Author Response · Authors · 2024-11-29
> **Global Response by the Authors: Further Experimental Results**
>
> We again thank all of the reviewers for the discussions so far, and also their patience waiting for the experiment results. We believe the newly added section resulting from the suggestions provided by the reviewers have without a doubt improved the quality of our work. We are looking forward to further discussing the paper with the reviewers, especially Reviewer #1, #2, and #3 who have not provided any feedbacks yet.
>
> Even though the paper is dense of experimental results, comments by reviewers made us curious to further explore a few more angles at scale. We have now managed to complete these experiments: and they confirm and further solidify our claims in the paper. Specifically, following Reviewer #1’s and Reviewer #2’s interest in performing the experiments on larger datasets, specifically ImageNet-sized datasets, we have performed the experiments from Figure 4 and Figure 6 on the Tiny-ImageNet dataset. The reason behind our use of Tiny-ImageNet instead of the original ImageNet was saving time and computation at the interest of the remaining time for the rebuttal period and our own computational resources, as the size of the images in ImageNet would make working with $\mathbf{G}\_{\mathcal{F}}$ extremely difficult and expensive. You can find the results for the Figure 4 and Figure 6 experiments [here](https://drive.google.com/file/d/1RZ8gKZS_TbsWZKU6RU6jg1qWcS7uU__4/view?usp=sharing) and [here](https://drive.google.com/file/d/1VrIo82MsbZe1B2UJsUB7QB1M2S9XBFDz/view?usp=sharing), respectively. As expected, the experiments are in line with the claims in the paper, and follow the idea introduced in GIH.
>
> Furthermore, following Reviewer #3’s interest in seeing further experiment results on other architecture, specially without skip-connection, we have also performed the experiments of Figure 4 and Figure 6 on a ResNet9 architecture without an skip-connection [1], the results of which you can observe [here](https://drive.google.com/file/d/1WFMtjpsjC5GyppGE5HGFfn9FnMBXO_L8/view?usp=sharing) and [here](https://drive.google.com/file/d/1mk5xVjGwYNlGPGLOSyKEHuOWRMwAK10v/view?usp=sharing). In this experiment, we again observe results that are in line with the claims in the paper and GIH.
>
> We hope that these experiment results along with the newly added sections and discussions in the paper have completely resolves the concerns and curiosities of the reviewers.
>
> [1] Yao, Z., Gholami, A., Keutzer, K. and Mahoney, M.W., 2020, December. Pyhessian: Neural networks through the lens of the hessian. In 2020 IEEE international conference on big data (Big data) (pp. 581-590). IEEE.

---

### Meta-Review · Area_Chair_tvym · 2024-12-17

**Metareview:**

This paper offers an innovative approach to understanding generalization in machine learning through geometry. The introduction of the Geometric Invariance Hypothesis could inspire further research into input-space properties and their role in model performance. While some aspects require further elaboration and validation, the work provides a solid foundation for exploring new directions in theoretical and empirical studies of generalization.

**Additional Comments On Reviewer Discussion:**

All reviewers agree that this is an interesting result and provide deep insights into understanding deep learning

---

### Decision · Program_Chairs · 2025-01-22

Accept (Spotlight)